# SCALING LAWS AND SPECTRA OF SHALLOW NEURAL NETWORKS IN THE FEATURE LEARNING REGIME

**Leonardo Defilippis**[*1]**, Yizhou Xu**[*2,3]**, Julius Girardin**[2]**, Emanuele Troiani**[2]**, Vittorio Erba**[2]**,
Lenka Zdeborová**[2]**, Bruno Loureiro**[1]**, and Florent Krzakala**[3]

[1]Departement d'Informatique, École Normale Supérieure, PSL & CNRS
[2]Statistical Physics of Computation Laboratory, École Polytechnique Fédérale de Lausanne (EPFL)
[3]Information, Learning and Physics Laboratory, École Polytechnique Fédérale de Lausanne (EPFL)

## ABSTRACT

Neural scaling laws underlie many of the recent advances in deep learning, yet their theoretical understanding remains largely confined to linear models. In this work, we present a systematic analysis of scaling laws for quadratic and diagonal neural networks in the feature learning regime. Leveraging connections with matrix compressed sensing and LASSO, we derive a detailed phase diagram for the scaling exponents of the excess risk as a function of sample complexity and weight decay. This analysis uncovers crossovers between distinct scaling regimes and plateau behaviors, mirroring phenomena widely reported in the empirical neural scaling literature. Furthermore, we establish a precise link between these regimes and the spectral properties of the trained network weights, which we characterize in detail. As a consequence, we provide a theoretical validation of recent empirical observations connecting the emergence of power-law tails in the weight spectrum with network generalization performance, yielding an interpretation from first principles.

## 1 INTRODUCTION

A central development in modern deep learning has been the recognition that neural network generalization does not improve unboundedly when training data, model size, or compute are scaled in isolation. Instead, extensive empirical evidence reveals the presence of performance bottlenecks unless these resources are increased together (Kaplan et al., 2020; Brown et al., 2020; Hoffmann et al., 2022). Characterizing these trade-offs, and in particular predicting the resulting *neural scaling laws*, has emerged as a fundamental challenge for deep learning research, with significant implications for the design of efficient and resource-conscious models.

Our goal in this work is to investigate this question in the context of shallow neural networks. More precisely, consider the following supervised empirical risk minimization (ERM) problem for the class of two-layer neural networks $f(\boldsymbol{x}; \boldsymbol{W}, \boldsymbol{a}) = \boldsymbol{a}^\top \sigma(\boldsymbol{W}\boldsymbol{x} + \boldsymbol{b})$:

$$\min_{\boldsymbol{W}, \boldsymbol{a}} \sum_{\mu=1}^{n} (y_\mu - f(\boldsymbol{x}_\mu; \boldsymbol{W}, \boldsymbol{a}))^2 + \lambda \left( ||\boldsymbol{W}||_{\mathrm{F}}^2 + ||\boldsymbol{a}||_2^2 \right) \tag{1}$$

where $\boldsymbol{W} \in \mathbb{R}^{p \times d}$ and $\boldsymbol{a} \in \mathbb{R}^p$ are the first- and second-layer weights, respectively. Although substantial progress has been achieved in recent years, our current understanding of scaling laws for the generalization performance of the ERM minimizer in eq. (1) remains largely confined to the random features regime (Bahri et al., 2024; Maloney et al., 2022; Paquette et al., 2024; Atanasov et al., 2024; Bordelon et al., 2024; Kunstner & Bach, 2025). In this setting, the problem reduces to a kernel method, where scaling behavior has been classically studied, and is known as *source and capacity conditions* (Caponnetto & De Vito, 2007; Cui et al., 2021; Defilippis et al., 2024).

In this work, we move beyond the random features regime and investigate neural scaling laws for the ERM minimizer in eq. (1) in the teacher-student setting. That is, we assume that the target task

---

[*]These authors contributed equally to this work

is generated by a teacher network of the same architecture

$$y_\mu = f(\boldsymbol{x}_\mu; \boldsymbol{W}^\star, \boldsymbol{a}^\star) + \sqrt{\Delta}\xi_\mu, \tag{2}$$

where $\{\boldsymbol{x}_\mu\}_{\mu=1}^n \overset{i.i.d.}{\sim} \mathcal{N}(0, I_d)$ denotes the dataset and $\{\xi_\mu\}_{\mu=1}^n \overset{i.i.d.}{\sim} \mathcal{N}(0, 1)$ is an additive Gaussian label noise with variance $\Delta \geq 0$. The statistics of the target weights $\boldsymbol{W}^\star, \boldsymbol{a}^\star$ will be specified later. Our main goal in this paper is to characterize the scaling-law and bottleneck behaviors of the excess risk

$$R(\boldsymbol{W}, \boldsymbol{a}) = \mathbb{E}_{\boldsymbol{x} \sim \mathcal{N}(0, I_d)} \left[ (f(\boldsymbol{x}; \boldsymbol{W}^\star, \boldsymbol{a}^\star) - f(\boldsymbol{x}; \boldsymbol{W}, \boldsymbol{a}))^2 \right] \tag{3}$$

associated with the minimizers $\hat{\boldsymbol{W}}, \hat{\boldsymbol{a}}$ of eq. (1). We will consider two specific classes of shallow neural networks. Thanks to exact mappings to classical problems in signal processing, these models admit a mathematical characterization, enabling an end-to-end analysis of the optimization problem in the feature learning regime.

**Diagonal networks and LASSO.** The first architecture is a diagonal neural network with $p = d$, diagonal first-layer weights $\boldsymbol{W} = \text{diag}(\boldsymbol{w})$, linear activation and no bias ($\boldsymbol{b} = 0$):

$$f(\boldsymbol{x}; \boldsymbol{W}, \boldsymbol{a}) = \boldsymbol{a}^\top \frac{(\boldsymbol{w} \odot \boldsymbol{x})}{\sqrt{d}}. \tag{4}$$

While the expressivity of this architecture is the same as that of a linear model, the reparameterization creates an effective implicit regularization that allows for feature selection and has made this setting popular among theoreticians. Indeed, adapting an argument by Neyshabur et al. (2015); Soudry et al. (2018); Pesme & Flammarion (2023) (see Appendix A), the resulting empirical minimization problem is equivalent to the LASSO problem with parameters $\theta_i = a_i w_i/\sqrt{d}$ and objective

$$\hat{\boldsymbol{\theta}} = \arg\min_{\boldsymbol{\theta} \in \mathbb{R}^d} \frac{1}{2} \sum_{\mu=1}^n \left(y_\mu - \boldsymbol{\theta}^\top \boldsymbol{x}_\mu\right)^2 + \lambda \|\boldsymbol{\theta}\|_1. \tag{5}$$

In other words, the ERM problem for a diagonal two-layer linear network trained with $\ell_2$ weight-decay can be understood through the performance of LASSO.

**Quadratic neural network and matrix compressed sensing.** The second architecture is that of an over-parameterized two-layer network with a (centered) quadratic activation,

$$f(\boldsymbol{x}; \boldsymbol{W}, \boldsymbol{a}) = \frac{1}{\sqrt{p}} \sum_{j=1}^p \left( \left(\frac{\boldsymbol{w}_j^\top \boldsymbol{x}}{\sqrt{d}}\right)^2 - \frac{\|\boldsymbol{w}_j\|_2^2}{d} \right) = \text{Tr}\left[\boldsymbol{S} \frac{\boldsymbol{x}\boldsymbol{x}^\top - \boldsymbol{I}_d}{\sqrt{d}}\right], \tag{6}$$

where $\boldsymbol{S} := \frac{\boldsymbol{W}^\top \boldsymbol{W}}{\sqrt{pd}} \in \mathbb{R}^{d \times d}$ and the normalization is taken for convenience. In this case, we fix the second-layer weight $\boldsymbol{a}$ of the model to be an all-one vector, but the target network may have arbitrary second layer weights. This class of quadratic neural networks have recently gained in popularity as simple models for non-convex tasks (Sarao Mannelli et al., 2020; Arnaboldi et al., 2023; Martin et al., 2024; Ben Arous et al., 2025). The ERM problem in eq. (1) for this architecture can be mapped to a sparse estimation setting (Gunasekar et al., 2017; Maillard & Kunisky, 2024; Bandeira & Maillard, 2025; Xu et al., 2025; Erba et al., 2025), namely matrix compressed sensing (or low-rank matrix estimation):

$$\hat{\boldsymbol{S}} = \arg\min_{\boldsymbol{S} \succeq 0} \sum_{\mu=1}^n (y_\mu - \text{Tr}[\boldsymbol{S}\boldsymbol{Z}_\mu])^2 + \lambda \|\boldsymbol{S}\|_*, \tag{7}$$

where $\boldsymbol{Z}_\mu := \frac{\boldsymbol{x}_\mu \boldsymbol{x}_\mu^\top - \boldsymbol{I}_d}{\sqrt{d}}$, $\|\cdot\|_*$ denotes the nuclear norm and we rescaled $\lambda \to \lambda/\sqrt{pd}$. We refer again to Appendix A for the explicit mapping. Thus, the performance of a quadratic network trained with weight decay can be analyzed via low-rank matrix estimation with nuclear norm regularization.

These two equivalences underline the central theme of this work: by mapping neural network training problems to sparse vector and matrix estimation tasks, we can leverage the rich theoretical toolbox developed for LASSO and compressed sensing, and in particular approximate message passing and its high-dimensional state evolution (Donoho et al., 2009; 2013; Javanmard & Montanari, 2013; Berthier et al., 2020b; Erba et al., 2025). This bridge not only enables precise predictions for generalization error and scaling exponents, but also provides a principled understanding of the weight spectral distribution in neural networks.

**Power-law/quasi-sparse targets.** To study scaling behavior, we adopt the classical assumption of a target with a power-law spectrum, as considered for instance in (Caponnetto & De Vito, 2007; Steinwart et al., 2009; Spigler et al., 2020; Cui et al., 2021; Bordelon et al., 2024; Ben Arous et al., 2025). In the language of compressed sensing, this corresponds to the notion of *quasi-sparsity* (Negahban & Wainwright, 2011; Raskutti et al., 2011), where the signal is not exactly sparse but its coefficients decay according to a heavy-tailed distribution. This makes the setting natural and relevant to both the machine learning and signal processing communities. Concretely, in the case of diagonal linear networks we assume effective weights

$$\theta_i^\star \overset{i.i.d.}{\sim} \mathcal{N}(0,\, d\, i^{-2\gamma}), \qquad \boldsymbol{\theta}^\star := \boldsymbol{W}^\star \boldsymbol{a}^\star, \tag{8}$$

while for quadratic neural networks we assume that

$$\boldsymbol{S}^\star := \frac{1}{\sqrt{pd}} \sum_{j=1}^{p} a_j^\star \, \boldsymbol{w}_j^\star (\boldsymbol{w}_j^\star)^T \tag{9}$$

is rotationally invariant with eigenvalues $\{\sqrt{d}\, i^{-\gamma}\}_{i=1}^{d}$. This setting was recently studied as well in Ben Arous et al. (2025) (who considered noiseless target ($\Delta = 0$) and obtain one of our scaling exponents). In both cases we fix $\gamma > 1/2$ to ensure square-summability of $\boldsymbol{\theta}^\star$ and $\boldsymbol{S}^\star$.

## 1.1 MAIN RESULTS

1. **Phase diagram and complete characterization of excess risk rates for power-law targets.** We provide a sharp characterization of the excess risk achieved by empirical risk minimization (1) for both diagonal linear networks and quadratic networks in the regime $n, d \gg 1$ with $p \geq d$, under a power-law design for the target function and varying regularization strength $\lambda$, summarized in Figure 1. Our results uncover a striking universality between the two settings, including a transition from *benign* to *harmful overfitting*. Exploring the extent of this universality beyond the setting here is an interesting avenue for future work. We also derive the risk rates under optimal regularization $\lambda$, and show that optimally-regularized ERM achieves the Bayes-optimal rates — previously known only for the diagonal case (Raskutti et al., 2011). These findings are of independent interest for sparse vector and low-rank matrix estimation.

2. **Spectral behavior of the learned weights.** We characterize, across all phases, the spectral properties of the trained network weights. The learned spectrum reflects the implicit trade-off between signal, noise, and regularization, and exhibits phenomena directly connected to feature learning. Remarkably, the resulting spectral behavior mirrors observations in modern deep learning practice (Martin & Mahoney, 2021a; Thamm et al., 2024).

3. **First-principles explanation of spectra–generalization connection.** We provide a clear interpretation of the spectrum and its relation to generalization. Building on Result 3, which decomposes the error into *underfitting, overfitting, and approximation* terms, we show that each of these components is directly connected to the spectral statistics of the weights. In doing so, we provide a mathematical theory for the empirical observations of Martin et al. (2021) and Wang et al. (2023) for the spectral statistics of weights in large-scale trained networks.

4. **Non-asymptotic validity of state evolution.** Our derivations rely on approximate message passing (AMP) and its state evolution equations, which are rigorously valid only in the proportional asymptotic regime with fixed ratios $n/d$ (or $n/d^2$) and fixed $\lambda$. We extend these equations heuristically beyond their proven setting, covering arbitrary scalings of $n, d, \lambda$. Through extensive numerical experiments, we demonstrate that state evolution remains accurate down to constants across the whole parameter space, including far beyond proven guarantees. This surprising robustness, already established in ridge regression (Cui et al., 2021; Cheng & Montanari, 2024; Misiakiewicz & Saeed, 2024; Defilippis et al., 2024), suggests a broader conjecture: the AMP framework, and related tools from spin glass theory, may provide predictive power well outside their standard asymptotic assumptions. We hope this will further motivate work on non-asymptotic control of AMP (Rush & Venkataramanan, 2018; Miolane & Montanari, 2021; Li & Wei, 2022; Reeves, 2025).

Together, these results provide a comprehensive theoretical and empirical understanding of scaling laws for feature learning in simple network models.

## 1.2 FURTHER RELEVANT WORK

**Scaling laws —** A large body of work has studied scaling laws in the lazy regime, where the features remain fixed. This includes kernel methods (Caponnetto & De Vito, 2007; Spigler et al., 2020;

Cui et al., 2021), random features (Defilippis et al., 2024; Atanasov et al., 2024; Bahri et al., 2024; Maloney et al., 2022; Paquette et al., 2024; Kunstner & Bach, 2025), and neural tangent kernels (NTK) (Bordelon et al., 2020). Bordelon et al. (2024; 2025) analyzed how scaling laws change for linear networks when both weights are trained, and Worschech & Rosenow (2024) explicitly solves the dynamics of a linear network to obtain the scalings. Our goal in this work is to go beyond linear networks and the lazy regime and analyze scaling laws in the presence of genuine feature learning. Two recent works (Ren et al., 2025; Ben Arous et al., 2025) analyze settings related to ours, but with important differences. Both consider two-layer networks with sublinear width, orthogonal first-layer weights, and power-law decaying second-layer weights. Ren et al. (2025) study activation functions with large information exponents, which is orthogonal to our setting, while Ben Arous et al. (2025) focus on quadratic activations with a specific SGD dynamics. Both works, additionally, consider noiseless targets and unregularized training ($\lambda = \Delta = 0$). Here, we study empirical risk minimization with weight decay $\lambda > 0$ and noise $\Delta \geq 0$.

**Spectral properties of learned weights —** A growing literature investigates the distribution of weight spectra in trained neural networks, with particular attention to the emergence of heavy-tailed eigenvalue distributions in both weights and activations (Mahoney & Martin, 2019; Martin et al., 2021; Martin & Mahoney, 2021b;a; Thamm et al., 2024; Wang et al., 2023; Zhou et al., 2023; Hodgkinson et al., 2025). Despite these empirical observations, a precise theoretical characterization of the learned spectra and their relation to generalization remains elusive. Recent progress includes analyses of the spectrum after a single or a few gradient steps (Dandi et al., 2024; Moniri et al., 2023; Cui et al., 2024; Dandi et al., 2025; Kothapalli et al., 2025), as well as results showing convergence of SGD in mean-field models to spectral distributions reminiscent of those we obtain (Olsen et al., 2025). Our description of the spectrum of the trained weight provides an analytic characterization of this phenomenon, and provides an interpretation of these properties from first principles.

**AMP and State Evolution —** Our analysis relies on approximate message passing (AMP) and its state evolution (SE), which has become a central tool for studying high-dimensional inference problems with structure (Donoho et al., 2013; Bayati & Montanari, 2011; Javanmard & Montanari, 2013; Berthier et al., 2020b; Zou & Yang, 2022; Feng et al., 2022; Gerbelot & Berthier, 2023; Dudeja et al., 2023; Erba et al., 2025). It has also been applied to learning problems beyond sparse recovery, such as kernel methods and learning rates (Cui et al., 2021; Loureiro et al., 2021). In this work, we use the state evolution equations of AMP heuristically, to analyze quasi-sparse models *beyond* their rigorously proven asymptotic regimes (typically assuming a fixed ratio $n/d$). While recent advances in non-asymptotic control (Rush & Venkataramanan, 2018; Miolane & Montanari, 2021; Li & Wei, 2022; Reeves, 2025) provide reassurance, a finer control of the limit is still required for a fully rigorous justification. Our experiments nevertheless show excellent agreement between SE predictions and numerical results across regimes, suggesting that AMP may be predictive well beyond its standard assumptions.

**Compressed sensing —** Quasi-sparse settings, where coefficients decay with a power law in Fourier or wavelet bases, have long been studied in statistics and signal processing. This is natural since most real-world signals are not exactly sparse but have heavy-tailed coefficient distributions (Mallat, 1999). Classical work on LASSO and matrix compressed sensing analyzed $\ell_p$-controlled targets, deriving minimax bounds on error and sample complexity (Raskutti et al., 2011; Negahban & Wainwright, 2011). Our results extend this line of work by providing the full phase diagram across all regularization strengths and data scales. For instance, the optimal LASSO rate of Raskutti et al. (2011) arises from setting $\lambda = \tilde{\Theta}(\sqrt{n/d})$ (here $\tilde{\Theta}$ is up to logarithmic factors).

## 2 MAIN RESULTS

### 2.1 UNIVERSAL ERROR RATES

In this section we discuss the excess risk rates associated to the two problems introduced above. Our analysis is based on a deterministic characterization of the risk $\hat{R}(\hat{\boldsymbol{W}}, \hat{\boldsymbol{a}}) \simeq \mathsf{R}_{n,d}$ at large $n, d \gg 1$, which is discussed in Section 2.4. We only consider the noisy case ($\Delta > 0$), and refer to Appendix B for the noiseless case. In order to highlight the correspondence between the two neural network models, we express the results in terms of the effective sample size $n_{\text{eff}}$ as follows:

$$n_{\text{eff}} \equiv \begin{cases} n & \text{for diagonal network} \\ n/d & \text{for quadratic network} \end{cases}. \tag{10}$$

Surprisingly, this definition will be enough to present both cases in a unified manner.

**Result 1** (Excess risk rates). *Under the setting of Sec. 1 for $\Delta > 0$ and $n_{\text{eff}} \gg 1$, the excess risk satisfies*

$$
\mathsf{R}_{n_{\text{eff}},d}(\lambda) = \begin{cases}
\Theta\left(n_{\text{eff}}^{-1+1/(2\gamma)} + \rho\left(n_{\text{eff}}/d\right)\right) & \text{if} \quad 1 \ll n_{\text{eff}} \ll d \quad \text{and} \quad \lambda \ll \sqrt{\frac{n_{\text{eff}}}{d}} \\
\Theta\left(\lambda^{-2/3}\right) & \text{if} \quad n_{\text{eff}} \sim d \quad \text{and} \quad \lambda \ll 1 \\
\Theta(d/n_{\text{eff}}) & \text{if} \quad n_{\text{eff}} \gg d \quad \text{and} \quad \lambda \ll \sqrt{\frac{n_{\text{eff}}}{d}} \\
\Theta\left(\left(\lambda d^{1/2}/n_{\text{eff}}\right)^{2-1/\gamma}\right) & \text{if} \quad \max\left(\sqrt{\frac{n_{\text{eff}}}{d}}, \frac{n_{\text{eff}}}{d^{\gamma+1/2}}\right) \ll \lambda \ll \frac{n_{\text{eff}}}{d^{1/2}} \\
\Theta\left(\lambda^2 d^2/n_{\text{eff}}^2\right) & \text{if} \quad \sqrt{\frac{n_{\text{eff}}}{d}} \ll \lambda \ll \frac{n_{\text{eff}}}{d^{\gamma+1/2}}
\end{cases}, \quad (11)
$$

*and $\mathsf{R}_{n_{\text{eff}},d} = \Theta(1)$ otherwise, where $\rho(t) = -1/\log(t)$ in the diagonal network case and $\rho(t) = t^{2/5}$ in the quadratic network case. Notice that in both cases the $\rho$ term is monotone increasing with $n_{\text{eff}}/d$, and dominates the error rate when $n_{\text{eff}} \to d$. Additionally, in the diagonal network case, the first rate holds up to logarithmic factors that we specify in eq. (86) in Appendix C.2.*

These rates are summarized in Figure 1. For small (fixed) regularization $\lambda < 1/\sqrt{d}$, with $d$ fixed and $n_{\text{eff}}$ increasing, the excess error moves from an initial plateau (Phase Ia), driven by data scarcity, to a fast–decay (Phase IV), where $\mathsf{R}_{n_{\text{eff}},d} = \Theta(n_{\text{eff}}^{-1+1/(2\gamma)})$, matching the minimax rate in (Raskutti et al., 2011; Donoho et al., 2011). As $n_{\text{eff}}$ approaches $d$, the estimator begins to fit the noise, and we observe a harmful overfitting (Phase V), in which the excess risk is dominated by the non-universal scale $\rho$ (arising from overfitting the noise as in Result 3). This transition, characteristic of the under-regularized and under-sampled regime ($1 \ll n_{\text{eff}} \ll d, \lambda \ll \sqrt{\frac{n_{\text{eff}}}{d}}$), happens at

$$
n_{\text{eff}}^{\text{cross}} = \begin{cases}
(\log d)^{\frac{4\gamma-1}{2\gamma-1}} & \text{for diagonal network} \\
d^{\frac{4\gamma}{14\gamma-5}} & \text{for quadratic network}
\end{cases}. \quad (12)
$$

The excess risk reaches its maximum around $n_{\text{eff}} \sim d$ with $\mathsf{R}_{n_{\text{eff}},d} \sim \lambda^{-2/3}$. This non-monotonicity of the risk at interpolation is reminiscent of the double descent behavior (Belkin et al., 2019; Mei & Montanari, 2022), and extends previous findings (Bartlett et al., 2020; Wang et al., 2024) to non-linear models. For $n_{\text{eff}} \gg d$, the excess risk then enters a second fast–decay Phase VIa, with rate proportional to $d/n_{\text{eff}}$; this is the fastest decay we observe, provided $n_{\text{eff}} \gg d^{2\gamma}$ (Phase VIb). For larger regularization strength $\lambda > 1/\sqrt{d}$, the excess risk decay is described by the upper part of the phase diagram in Figure 1, eventually crossing to the lower part when $n_{\text{eff}} \sim d\lambda^2$. In particular, if $\lambda \gg d^{\gamma-1/2}$, we observe that increasing $n_{\text{eff}}$, the excess risk is initially in a plateau (Phase Ib), induced by the strong regularization with respect to the sample size. For $n \sim \lambda d^{1/2}$, it crosses into a slow rate Phase II with $\mathsf{R}_{n_{\text{eff}},d} = \Theta(\lambda d^{1/2}/n_{\text{eff}})^{2-1/\gamma}$, which transitions to a faster rate (Phase III), still influenced by the large regularization, for $n_{\text{eff}} \sim \lambda d^{\gamma+1/2}$, with $\mathsf{R}_{n_{\text{eff}},d} = \Theta((\lambda d/n_{\text{eff}})^2)$. Phase II recovers the rate in (Negahban & Wainwright, 2011, Corollary 2). The excess risk eventually transitions to the fast-decay Phase VIb for $n_{\text{eff}} \sim \lambda^2 d$, where the effect of the regularization becomes negligible due to the large sample size. These cross-overs are reminiscent of the ones observed for kernel and random feature ridge regression respectively in Cui et al. (2021); Defilippis et al. (2024).

We observe that there are region boundaries along which the rates are discontinuous (red lines in Figure 1). At the boundary $n_{\text{eff}} = \Theta(d)$, we observe the aforementioned crossover between harmful overfitting and fast decay, with an interpolation peak emerging. At the boundary $d \ll n_{\text{eff}} \ll d^{2\gamma}$ and $\lambda = \Theta(\sqrt{n_{\text{eff}}/d})$, the excess risk jumps from $d/n_{\text{eff}}$ to $n_{\text{eff}}^{-1+1/(2\gamma)}$ (which is much lower) when increasing the regularization.

As a corollary of the above results, we can immediately estimate the behavior of the optimal regularization $\lambda_{\text{opt}}$ and the associated optimal ERM rates.

**Corollary 1** (Optimal regularization and optimality of ERM). *The optimal regularization satisfies*

$$
\lambda_{\text{opt}}(n_{\text{eff}}, d) = \begin{cases}
O(\sqrt{n_{\text{eff}}/d}), & \text{if } \Delta > 0 \text{ and } \left(1 \ll n_{\text{eff}} \ll n_{\text{eff}}^{\text{cross}} \text{ or } n_{\text{eff}} \gg d^{2\gamma}\right) \\
\tilde{\Theta}(\sqrt{n_{\text{eff}}/d}), & \text{if } \Delta > 0 \text{ and } n_{\text{eff}}^{\text{cross}} \ll n_{\text{eff}} \ll d^{2\gamma}
\end{cases}. \quad (13)
$$

*where $\tilde{\Theta}$ is up to logarithm factors in the argument. The excess risk rates in the optimally regularized case matches the large $n_{\text{eff}}, d \gg 1$ Bayesian risk $R_{\text{BO}}(\mathcal{D}) = \mathbb{E}[R(\boldsymbol{W}, \boldsymbol{a})|\mathcal{D}] \simeq \mathsf{R}_{n_{\text{eff},d}}^{\text{BO}}$ \** (for the*

---

\**See section 2.4 for a formal statement.

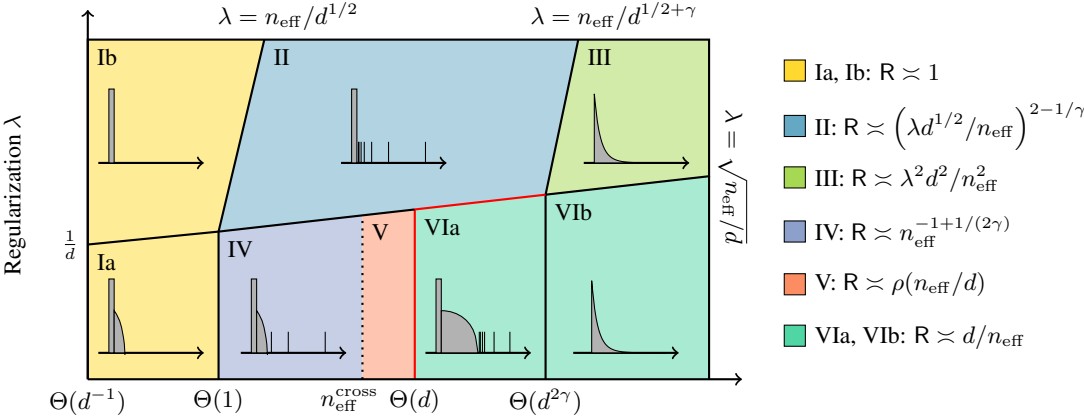

Effective number of samples $n_{\text{eff}}$ ($n$ for diagonal, $n/d$ for quadratic)

Figure 1: Excess risk rates of Result 1 as a function of $n$ and $\lambda(n,d)$, with a sketch of the corresponding spectral properties of the learned weights (Result 2). Red lines represent discontinuous phase boundaries.

*diagonal network, up to logarithmic factors), given by*

$$
\mathrm{R}_{n_{\text{eff}},d}(\lambda_{\text{opt}}) = \Theta(\mathrm{R}^{\text{BO}}_{n_{\text{eff}}}) = \begin{cases} \Theta(n_{\text{eff}}^{-1+1/(2\gamma)}) & \text{if} \quad \Delta > 0 \quad \text{and} \quad 1 \ll n_{\text{eff}} \ll d^{2\gamma} \\ \Theta(d/n_{\text{eff}}) & \text{if} \quad \Delta > 0 \quad \text{and} \quad n_{\text{eff}} \gg d^{2\gamma} \end{cases}, \quad (14)
$$

*and $\mathrm{R}_{n_{\text{eff}},d} = \Theta(1)$ otherwise. Again, in the diagonal network case, in the first regime the rate holds up to logarithmic factors that we specify in eq. (86) in Appendix C.2.*

Corollary 1 shows that by appropriately tuning the regularization allows to avoid the harmful overfitting phase in the noisy case and reach Bayesian optimality. Interestingly, the noisy rate $\Theta(n_{\text{eff}}^{-1+1/(2\gamma)})$ in the regime $1 \ll n_{\text{eff}} \ll d^{2\gamma}$ coincides with the classical minimax rate for high-dimensional linear regression over an $\ell_q$-ball with $q = 1/\gamma$ (Raskutti et al., 2011; Donoho et al., 2011). Corollary 1 not only recovers the well-known result that properly regularized LASSO achieves this minimax rate, but also extends it to additional regimes and to the matrix case, revealing a cross-over between the minimax rate and a faster $\Theta(d/n_{\text{eff}})$ rate.

## 2.2 SPECTRA OF THE LEARNED WEIGHTS

Our second set of results concerns the structural properties of the learned weights, that are given by a soft thresholding function applied to a noisy version of the target's weights. Notice that for diagonal neural networks, the weights $\boldsymbol{\theta}$ can be seen as a diagonal matrix (modulo a sign), hence they coincide with the eigenvalues of $\boldsymbol{W}$.

**Result 2** (Spectrum of the learned weights). *For the diagonal network case, there exists constants $\delta(n,d,\lambda)$ and $\epsilon(n,d,\lambda)$ (specified in Appendix C.1.2) such that the empirical risk estimator (1) satisfies (in distribution)*

$$
\hat{\theta}_i \sim \sigma_{\text{d}}(\theta_i^* + \delta z_i; \epsilon), \quad (15)
$$

*where $z_i \sim \mathcal{N}(0,1)$, and $\sigma_{\text{d}}(x;a) = \max(x-a,0) - \max(-x-a,0)$ is the soft-thresholding function. For the quadratic network case, there exists constants $\delta(n,d,\lambda)$ and $\epsilon(n,d,\lambda)$ (that are obtained from (20)) such that the spectrum $\nu$ of the empirical risk estimator (7) satisfies*

$$
\nu(x) = F_{\mu_\delta}(\lambda\epsilon)\boldsymbol{\delta}_0(x) + \mu_\delta(x + \lambda\epsilon)\mathbf{1}_{x>0}. \quad (16)
$$

$\boldsymbol{\delta}_0$ *represents a Dirac mass at 0, $\mathbf{1}_A$ is the indicator function of the set $A$ and $\mu_\delta$ represents the spectrum of $\boldsymbol{S}^\star + \delta\boldsymbol{Z}$ with its cumulative function $F_{\mu_\delta}$, where $\boldsymbol{Z} \sim \text{GOE}(d)$ (i.e. a symmetric matrix with $\mathcal{N}(0,1/d)$ elements up to symmetry).*

Result 2 characterizes the learned weights in both settings: they are *noisy, soft-thresholded versions of the target spectrum*. The parameter $\delta$ quantifies the noise from the label noise and finite sample estimation of the target weights, while $\lambda\epsilon$ sets the cutoff below which singular values vanish due to regularization. For any $n,d,\lambda$, the spectrum consists of a spike at zero, possibly a bulk near zero, and a few outliers aligned with the top eigenvectors of the target.

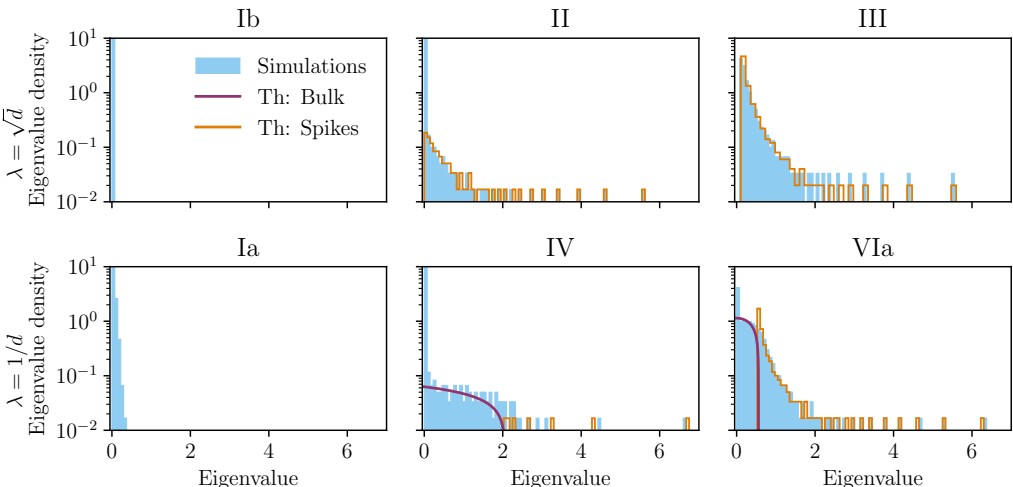

Figure 2: Comparison between spectra from simulations and theory across different training phases. Blue: eigenvalue histograms after training. Purple/orange: theoretical predictions for bulk and spikes, respectively (16) (for clarity, spike histograms are shown separately). Notice that our theory also predicts a spike at zero, which we do not plot for visual clarity. All panels use $d = 800$ except III, for which we have $d = 400$. Bottom row: $\lambda = 1/d$ with $n = 100,\ 1.94 \times 10^4,\ 1.28 \times 10^6$. Top row: $\lambda = \sqrt{d}$ with $n = 800,\ 6.4 \times 10^6,\ 2 \times 10^7$. We discuss the phenomenology in Section 2.3.

## 2.3 INTERPRETABILITY, AND A "UNIVERSAL" ERROR DECOMPOSITION

The spectra depends on $n, d, \lambda$ only through the functions $\delta, \epsilon$, leading to a qualitative structure shared by both models. Our theory predicts eight distinct spectral phases (Figure 1) which are closely connected to the risk rates in Result 1. Focusing now on the quadratic network, the result provides an interpretation of the risk in terms of the weights spectrum.

**Result 3** ("Universal" error decomposition of feature learning). *Let $\{s_i\}_{i=1}^d$ of $\mathbf{S}^\star$ be the eigenvalues of $\mathbf{S}^\star$ in non-increasing order. Consider the following two cases.*

*(i): Under-regularization. Assume that the constants $\delta(n, d, \lambda)$ and $\epsilon(n, d, \lambda)$ in Result 2 satisfy $\lambda\epsilon < 2\delta$ and there exists a cutoff $K(\delta) \ll d$ satisfying $s_{K(\delta)} = \delta$. Then the excess risk reads*

$$
\mathsf{R}_{n,d} = \underbrace{\delta^2 \int_{\lambda\epsilon/\delta}^{2} \mu_{\mathrm{sc}}(\mathrm{d}x)\left(x - \frac{\lambda\epsilon}{\delta}\right)^2 + \frac{1}{d}\delta K'(\delta)(2\delta - \lambda\epsilon)^2}_{\text{overfitting (learned noise)}} + \underbrace{\frac{1}{d}\sum_{i=K(\delta)+1}^{d} s_i^2}_{\substack{\text{underfitting}\\\text{(not learned features)}}} + \underbrace{\frac{1}{d}\sum_{i=1}^{K(\delta)}\left[\left(\frac{\delta^2}{s_i} - \lambda\epsilon\right)^2 + \frac{\delta^2}{s_i}\left(s_i + \frac{\delta^2}{s_i} - \lambda\epsilon\right)\right]}_{\substack{\text{approximation error}\\\text{for learned features}}}.
$$

*where $\mu_{\mathrm{sc}}(\mathrm{d}x) = (2\pi)^{-1}\sqrt{4 - x^2}\mathbf{1}_{x\in[-2,2]}\mathrm{d}x$ denotes the Wigner semi-circle law.* (17)

*(ii): Over-regularization. Assume that the constants $\delta(n, d, \lambda)$ and $\epsilon(n, d, \lambda)$ in Result 2 satisfy $\lambda\epsilon \geq 2\delta$ and there exists a cutoff $K(\delta, \lambda\epsilon) \ll d$ satisfying $s_{K(\delta,\lambda\epsilon)} + \frac{\delta^2}{s_{K(\delta,\lambda\epsilon)}} - \lambda\epsilon = 0$. Then*

$$
\mathsf{R}_{n,d} = \underbrace{\frac{1}{d}\sum_{i=K(\delta,\lambda\epsilon)+1}^{d} s_i^2}_{\substack{\text{underfitting}\\\text{(not learned features)}}} + \underbrace{\frac{1}{d}\sum_{i=1}^{K(\delta,\lambda\epsilon)}\left[\left(\frac{\delta^2}{s_i} - \lambda\epsilon\right)^2 + \frac{\delta^2}{s_i}\left(s_i + \frac{\delta^2}{s_i} - \lambda\epsilon\right)\right]}_{\substack{\text{approximation error}\\\text{for learned features}}}.
$$
(18)

Equations (17) and (18) have clear interpretations. The first component of the overfitting term corresponds to the second moment of the bulk spectrum, representing the power of the learned noise and thus quantifying the degree of overfitting. The second part of the overfitting term is proportional to the square of the bulk size, but always remains subdominant compared to the first term. As shown in Equation (18), the overfitting term diminishes with increasing regularization strength.

The underfitting term measures the mean power of the unlearned spikes, indicating how many features are lost due to the cutoff $K(\delta, \lambda\epsilon)$ (which depends on the noise and regularization). The

approximation error term reflects the average error in the learned spikes, which depends on the effective signal-to-noise ratio $\frac{s_i}{\delta}$ and the effective regularization $\lambda\epsilon$. Notably, when the effective noise $\delta$ is zero, the approximation error increases with regularization; conversely, when the regularization $\lambda$ is zero, the approximation error increases with noise $\frac{\delta}{s_i}$. In general, however, the approximation error arises from a non-trivial interplay between the effective noise and regularization.

This decomposition is "universal" in that it does not depend on the target spectrum, dataset size, or regularization, holds for all $\Delta \geq 0$ and applies across different spectral phases. While derived here for quadratic networks, similar expressions hold as well for diagonal networks. See Appendix C.1.2. Extending it to broader architectures is an interesting direction for future work.

Based on the error decomposition in Result 3, we give an interpretation of the rates in Result 1 in terms of the weight spectral properties (see Figure 1 for illustrations and Figure 2 for experiments): the bulk corresponds to learned noise, the spikes hidden by the bulk are the unlearned features, and the outliers are the learned features. This provides a mathematical theory from first principles for the observations of (Martin & Mahoney, 2021a; Martin et al., 2021), whose terminology (e.g. "bleed-out", "rank collapse", ...) we borrow in the following. We begin by the case of large regularization, considering an increasing number of samples.

- **Ib (Rank collapse)**: All eigenvalues are zero. Data scarcity and strong regularization imply that the ERM estimator is zero. Result 3 then gives a trivial risk $\mathsf{R} = \mathrm{Tr}[(\boldsymbol{S}^\star)^2]/d$.

- **II (Outliers)**: The spectrum contains approximately $N_{\mathrm{out}} = \left(\frac{4n}{\lambda d^{3/2}}\right)^{1/\gamma}$ outliers, while the remaining eigenvalues are zero. Spikes are shifted by $\approx \frac{\lambda d^2}{4n}$. In this regime, some features are learned with noise, while others are lost due to over-regularization. Result 3 implies that the risk is determined by the number and shift of the spikes, yielding $\mathsf{R} = \Theta\left(\frac{1}{d}\sum_{i \geq N_{\mathrm{out}}}(\sqrt{d}i^{-\gamma})^2\right) = \Theta\left((\lambda d^{3/2}/n)^{2-1/\gamma}\right)$, since the error from the shift is of the same order.

- **III (Heavy-tail)**: The spectrum is a perturbed version of the target spectrum with a heavy-tail $\rho(x) \sim x^{-1-1/\gamma}$. Regularization shifts the bulk leftward by $\approx \frac{\lambda d^2}{4n}$, yielding $\mathsf{R} = \Theta\left((\lambda d^2/n)^2\right)$.

As more eigenvalues emerge from the spike at zero, more features are learned and the risk decreases. Strong regularization suppresses any spurious bulk of small eigenvalues, as well as some of the smaller spikes. Consider now the case of small regularization.

- **Ia (Rank collapse)**: The spectrum resembles a small portion of a semi-circle law along with many zero eigenvalues. Perhaps surprisingly, the contribution of the bulk is negligible even for vanishing regularization. Neither features nor noise are learned, so the risk remains $\mathsf{R} = \mathrm{Tr}[(\boldsymbol{S}^\star)^2]/d$.

- **IV (Bulk + Outliers)**: The spectrum exhibits $N_{\mathrm{out}} = (\Delta d/4n)^{-1/2\gamma}$ outliers and a bulk with eigenvalues of order $\Theta\left((\Delta^5 d^2/n)^{1/10}\right)$. As in Phase II, the risk decreases as more spikes emerge from the bulk. Since the bulk contribution is sub-leading, Result 3 implies that the risk scales as the average power of the unlearned spikes: $\mathsf{R} = \frac{1}{d}\sum_{i \geq N_{\mathrm{out}}}(\sqrt{d}i^{-\gamma})^2 = \Theta\left((d/n)^{1-\frac{1}{2\gamma}}\right)$.

- **V (Bulk + Outliers)**: Similar to Phase IV, but the risk is now dominated by the bulk of eigenvalues of order $\Theta\left((\Delta^5 d^2/n)^{1/10}\right)$. The ERM estimator approaches interpolation and begins to fit noise. Although the number of outliers $(N_{\mathrm{out}} = \Delta d/4n)^{-1/2\gamma}$ continues to increase and the bulk range shrinks, the bulk's second moment grows. Altogether, Result 3 implies the risk increases.

- **Interpolation peak (Bulk + Outliers)**: The spectrum is dominated by a large semi-circle bulk of order $\Theta(\Delta^{2/3}\lambda^{-1/3})$. There may still be $\Theta(\Delta^{2/3}\lambda^{-1/3}d^{-1/2})$ spikes if $\lambda \gg d^{-3/2}$, but their contribution is negligible. Even if the model learns features, the noise is overwhelming, and Result 3 implies the risk is dominated by the bulk second moment $\mathsf{R} = \Theta(\lambda^{-2/3})$.

- **VIa (Bulk + Bleed-out + Outliers)**: The spectrum contains $(\Delta d/4n)^{-1/2\gamma}$ outliers and a bulk of small eigenvalues of order $\sqrt{\Delta d^2/4n}$. The smallest outliers merge at the bulk boundary, creating a *bleed-out* effect. The risk decreases as more outliers emerge. The spikes are perturbed by $\Theta(\sqrt{\Delta d^2/n})$, and Result 3 implies the risk scales as $\mathsf{R} = \Theta(d^2/n)$, since this dominates over the unlearned features.

- **VIb (Heavy-tail)**: As in Phase III, the spectrum is a perturbed version of the target (a *heavy-tailed* bulk) with perturbations $\Theta(\sqrt{\Delta d^2/n})$. The risk decays with the perturbation as $\mathsf{R} = \Theta(d^2/n)$.

Therefore, for the under-regularized case, as the number of samples is increased, the bulk keeps shrinking as the spikes pop out. However, as the shape of the bulk and the number of zero eigenvalues changes, the risk changes non-monotonically. In other words, the model learns an increasing number of features, but the influence of noise leads to a non-monotonic behavior in the risk. Furthermore, Result 3 shows that regularization only affects the first term of eq. (17) and may increase the last two terms. Thus the optimal regularization strategy is to truncate the bulk, setting the first term to zero, leaving the second unchanged and minimally increasing the third. If the bulk contribution is negligible, weaker regularization may be chosen (i.e. in phases IV and VIb). This reasoning explains Corollary 1.

Finally, we should note that although phases VIa and VIb exhibit similar risk decay rates, only VIb achieves the Bayes-optimal rate. In the regime $d \ll n_{\text{eff}} \ll d^2$, the optimal performance is reached in phase II with $\lambda = \sqrt{n_{\text{eff}}/d}$. The corresponding spectral density shows a transition from outlier-dominated (zero eigenvalues+spikes) to heavy-tailed behavior, which supports the argument of Martin et al. (2021) that heavy-tailed spectra are associated with superior generalization.

## 2.4 Non-asymptotic state evolution

The results of Sections 2.1, 2.2 and 2.3 build on the theory of state evolution and approximate message passing algorithms (Donoho et al., 2009; Javanmard & Montanari, 2013; Gerbelot & Berthier, 2023), whose formal guarantees hold in the high-dimensional limit $n_{\text{eff}}, d \to \infty$ with fixed ratio $n_{\text{eff}}/d$ and constant strength $\lambda$. In this regime, state evolution allows to characterize the asymptotic risk and the spectrum of the weights in both the neural networks models under consideration, both for the empirical risk minimizer and for the Bayes-optimal estimator (see Appendices C and D).

Non-rigorous analyses in the ridge regression literature have employed asymptotic formulas to estimate excess risk rates under source & capacity conditions, recovering classical results while also identifying new regimes in striking agreement with finite-size numerical experiments (Bordelon et al., 2020; Cui et al., 2021; Simon et al., 2023). The validity of these formulas beyond proportional asymptotics was subsequently established through non-asymptotic multiplicative bounds, thereby placing these rates on rigorous grounds (Cheng & Montanari, 2024; Misiakiewicz & Saeed, 2024; Defilippis et al., 2024). Motivated by this line of work, we derive our results under an analogous assumption: namely, that the state evolution equations for LASSO and matrix compressed sensing remain valid beyond proportional asymptotics. This assumption is supported both by extensive numerical evidence, depicted in Figure 3 and Appendices C, F, and by rigorous results on the convergence rates for the LASSO state evolution (Miolane & Montanari, 2021)[*]. Figure 3 also confirms the theoretical decay rates of the excess risk across all phases, with state evolution and simulations in excellent agreement (see Appendix F for details on the implementation). Nonetheless, establishing non-asymptotic multiplicative bounds for LASSO and matrix compressed sensing remains a challenging open problem. Our results provide both motivation and supporting evidence for this direction, which we leave for future work. For conciseness, we only present the conjecture regarding quadratic networks, and refer to Appendix C for the conjecture concerning diagonal networks.

**Conjecture 1.** *Let $\lambda > 0$, $\Delta \geq 0$ and consider $n, d \gg 1$ sufficiently large. Then with a probability at least $1 - o_n(1) - o_d(1)$, both the excess risk associated to the empirical risk minimizer eq. (7) and the Bayes-optimal risk (i.e. either $R_{\text{BO}}$ or $R(\hat{\boldsymbol{W}}, \hat{\boldsymbol{a}})$) satisfy $|R - \mathsf{R}_{n,d}| = \mathsf{R}_{n,d} \cdot o_{n,d}(1)$. More precisely, for the Bayes-optimal case we have $\mathsf{R}_{n,d}^{\text{BO}} = Q^\star - q$ with $Q^\star := \frac{1}{d} \operatorname{Tr}[(\boldsymbol{S}^\star)^2]$ and $q$ given by the fixed point of the following equation*

$$\hat{q} = \frac{4n/d^2}{\Delta + 2(Q^\star - q)}, \qquad 1 - 2\tilde{\alpha} + \frac{\Delta \hat{q}}{2} = \frac{4\pi^2}{3\hat{q}} \int \mu_{1/\sqrt{\hat{q}}}(x)^3 \mathrm{d}x, \qquad (19)$$

*where $\mu_{1/\sqrt{\hat{q}}}$ denotes the spectrum of $\boldsymbol{S}^\star + \frac{1}{\sqrt{\hat{q}}}\boldsymbol{Z}$ with $\boldsymbol{Z} \sim \text{GOE}(d)$. For the ERM, $\mathsf{R}_{n,d} = \frac{2n}{d^2}\delta^2 - \frac{\Delta}{2}$, where $\delta$ is given by the fixed point of the following equation*

$$\begin{cases} 4\frac{n}{d^2}\delta - \frac{\delta}{\epsilon} = \partial_1 J(\delta, \lambda\epsilon), \\ Q^\star + \frac{\Delta}{2} + 2\frac{n}{d^2}\delta^2 - \frac{\delta^2}{\epsilon} = (1 - \lambda\epsilon\partial_2)J(\delta, \lambda\epsilon), \end{cases} \qquad J(a, b) := \int_b^{+\infty} \mu_a(x)(x - b)^2 \mathrm{d}x. \quad (20)$$

*with $\mu_a$ denoting the spectrum of $\boldsymbol{S}^\star + a\boldsymbol{Z}$ with $\boldsymbol{Z} \sim \text{GOE}(d)$.*

---

[*]However, we need to extend their results to finite sample analysis (Rush & Venkataramanan, 2018)

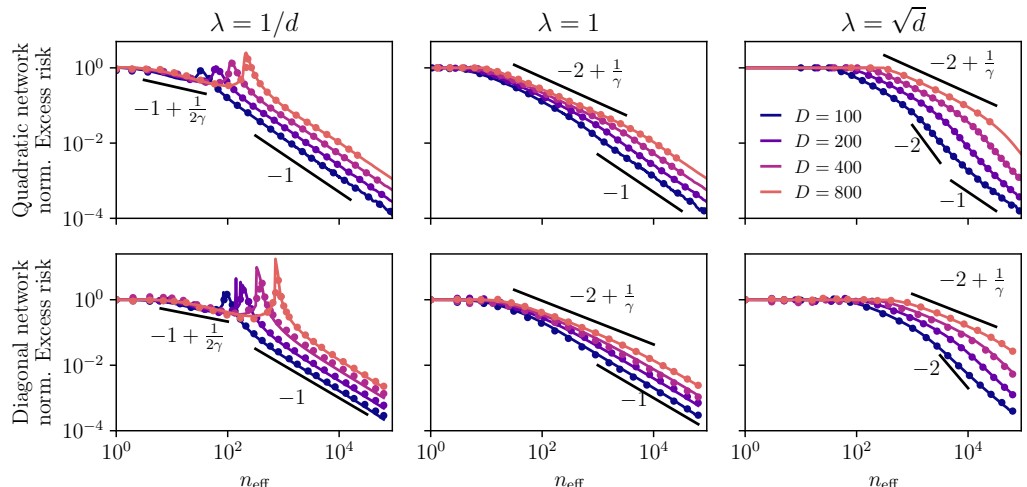

Figure 3: Excess risk in simulations (dots, $d = 100, 200, 400, 800$) versus non-asymptotic state evolution (solid lines) as a function of $n_{\text{eff}}$ ($n_{\text{eff}} = n$ for the diagonal case, $n_{\text{eff}} = n/d$ for the quadratic case) with $\lambda = 1/d, 1, \sqrt{d}$ and $\Delta = 0.5$. The risk is rescaled such that it starts at 1. We find excellent agreement, despite state evolution being rigorous only in the asymptotic limit $n_{\text{eff}}/d = \Theta(1)$ with $d \gg 1$. Black lines indicate the decay rates of the excess risk predicted by Result 1, again showing good agreement. The networks are trained in PyTorch using LBFGS as detailed in Appendix F.

## 3 CONCLUSION

We studied a theoretical framework for scaling laws in shallow networks with feature learning by mapping them to sparse vector and low-rank matrix estimation. This allowed us to derive a comprehensive phase diagram for the excess risk scaling laws, uncovering a universality between diagonal and quadratic networks. Our analysis provides a first-principles explanation of the weight spectra–generalization connection: underfitting, overfitting, and approximation errors correspond directly to distinct spectral features, yielding a firm foundation for empirical observations of heavy-tailed weight spectra and their link to generalization.

There are many natural extensions of this work, such as exploring additional structures present in the data (e.g., non-trivial covariances (Wortsman & Loureiro, 2025)), extending beyond two-layer networks and quadratic activations (Barbier et al., 2025), providing a rigorous proof of the state evolution conjecture following Miolane & Montanari (2021). Moreover, our current work only analyzes the global minimum, so we should also look at compute scaling laws of GD/SGD (Ben Arous et al., 2025) as well as the implicit biases of SGD towards heavy tails and its relation to generalization (Gurbuzbalaban et al., 2021; Simsekli et al., 2020; Hodgkinson et al., 2022). We hope these results will motivate further progress toward a systematic theory of neural scaling laws.

### ACKNOWLEDGEMENTS

We would like to thank Murat Erdogdu, Surya Ganguli, Antoine Maillard, Pierre Mergny and Denny Wu for insightful discussions. BL and LD were supported by the French government, managed by the National Research Agency (ANR), under the France 2030 program with the project references "ANR-23-IACL-0008" (PR[AI]RIE-PSAI) and "ANR-25-CE23-5660" (MAPLE), as well as the Choose France - CNRS AI Rising Talents program. We also acknowledge funding from the Swiss National Science Foundation grants SNSF SMArtNet (grant number 212049), OperaGOST (grant number 200021 200390) and DSGIANGO (grant number 225837). This work was supported by the Simons Collaboration on the Physics of Learning and Neural Computation via the Simons Foundation grant (#1257412 (FK) and #1257413 (LZ)).

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

## A  THE BRIDGE FROM SPARSE ESTIMATION TO NEURAL NETWORKS

### A.1  EQUIVALENCE BETWEEN DIAGONAL NETWORKS WITH $\ell_2$ WEIGHT DECAY AND LASSO

The first equivalence was discussed in a number of papers (Neyshabur et al., 2015; Soudry et al., 2018; Pesme & Flammarion, 2023). We consider the diagonal two–layer network with parameters $u, w \in \mathbb{R}^d$ and effective predictor

$$\theta = u \odot w, \qquad f(x) = \sum_{i=1}^{d} \theta_i x_i,$$

trained with squared loss and $\ell_2$ weight decay on both layers:

$$\min_{u,w \in \mathbb{R}^d} \quad \frac{1}{2}\|y - X(u \odot w)\|_2^2 + \frac{\lambda}{2}\big(\|u\|_2^2 + \|w\|_2^2\big). \tag{21}$$

Alternatively, one may also consider a diagonal two-layer ReLU network with two branches per coordinate:

$$f(x) \;=\; \sum_{i=1}^{d} \Big( u_i \, \sigma(w_i x_i) \;+\; u_i \, \sigma(-w_i x_i) \Big), \qquad \sigma(z) = \max\{z, 0\}.$$

Using $\sigma(z) - \sigma(-z) \;=\; z$, each pair of branches along coordinate $i$ induces an effective linear coefficient $\theta_i$ such that

$$f(x) \;=\; \sum_{i=1}^{d} \theta_i x_i.$$

We know show the reduction of this problem to the LASSO one:

**Step 1. Lower bound via AM–GM.** For each coordinate $i$ we have

$$u_i^2 + w_i^2 \;\geq\; 2|u_i w_i| = 2|\theta_i| \quad \text{(by AM–GM, with } a = u_i^2, b = w_i^2 \text{)}.$$

Summing over $i$ gives

$$\|u\|_2^2 + \|w\|_2^2 \;\geq\; 2\|\theta\|_1.$$

Therefore, for any factorization with $u \odot w = \theta$,

$$\frac{1}{2}\|y - X\theta\|_2^2 + \frac{\lambda}{2}\big(\|u\|_2^2 + \|w\|_2^2\big) \;\geq\; \frac{1}{2}\|y - X\theta\|_2^2 + \lambda\|\theta\|_1. \tag{22}$$

**Step 2. Tightness.** For any $\theta$, choose a factorization

$$u_i = \operatorname{sign}(\theta_i) \, |\theta_i|^{1/2}, \qquad w_i = |\theta_i|^{1/2}.$$

Then $u_i^2 = w_i^2 = |\theta_i|$, so that

$$u_i^2 + w_i^2 = 2|\theta_i|, \quad u_i w_i = \theta_i.$$

Hence equality holds in (22), and the regularizer becomes

$$\frac{\lambda}{2}(\|u\|_2^2 + \|w\|_2^2) = \lambda\|\theta\|_1.$$

Plugging back into (21), we obtain the exact equivalence

$$\min_{u,w} \; \frac{1}{2}\|y - X(u \odot w)\|_2^2 + \frac{\lambda}{2}\big(\|u\|_2^2 + \|w\|_2^2\big) \;\equiv\; \min_{\theta \in \mathbb{R}^d} \; \frac{1}{2}\|y - X\theta\|_2^2 + \lambda\|\theta\|_1,$$

which is precisely the *LASSO* loss.

### A.2 Equivalence between quadratic networks with $\ell_2$ weight decay and matrix compressed sensing

Again the equivalance has been discussed in a number of work (Gunasekar et al., 2017; Maillard & Kunisky, 2024; Erba et al., 2025; Bandeira & Maillard, 2025)

We consider the two-layer quadratic network with centered activations

$$f(\boldsymbol{x}; \boldsymbol{W}) \;=\; \frac{1}{\sqrt{p}} \sum_{j=1}^{p} \Big[ (\boldsymbol{w}_j^\top \boldsymbol{x})^2 - \mathbb{E}[(\boldsymbol{w}_j^\top \boldsymbol{x})^2] \Big].$$

Centering is equivalent to learning (and absorbing) the constant offset via a bias term, and can also be naturally implemented in practice by batch/layer normalization applied after squaring.

This network can be written as

$$f(\boldsymbol{x}) \;=\; \operatorname{Tr}[\boldsymbol{S}\boldsymbol{Z}],$$

where $\boldsymbol{S} := \frac{1}{\sqrt{p}} \boldsymbol{W}\boldsymbol{W}^\top \succeq 0$ and $\boldsymbol{Z} := \boldsymbol{x}\boldsymbol{x}^\top - \boldsymbol{\Sigma}_x, \boldsymbol{\Sigma}_x = \mathbb{E}[xx^\top]$. Thus the network corresponds exactly to a PSD matrix sensing model with centered measurements $\boldsymbol{Z}$. Centering removes only

a constant offset, which in practice would be absorbed by a bias term or handled automatically by batch/layer normalization. Moreover, weight decay on $\boldsymbol{W}$ induces a trace penalty on $\boldsymbol{S}$, since

$$\|\boldsymbol{W}\|_F^2 = \sqrt{p}\operatorname{tr}(\boldsymbol{S}),$$

so that training is equivalent to trace-regularized PSD matrix sensing.

Following the universality results for matrix sensing (see, e.g., Maillard & Kunisky (2024); Bandeira & Maillard (2025); Maillard et al. (2024); Xu et al. (2025); Erba et al. (2025)), the analysis can be simplified by replacing the empirical sensing operators $\boldsymbol{Z}$ by i.i.d. Gaussian symmetric matrices with matching covariance structure. In particular, for $x_i \sim \mathcal{N}(0, I_d)$, the centered measurements are distributed as rank-one Wishart fluctuations, which are asymptotically equivalent, in the sense of state evolution and AMP analysis, to Gaussian measurements with the same variance. Hence, without loss of generality, we may study the trace-regularized PSD matrix sensing problem with Gaussian measurement operators

$$y_\mu = \operatorname{Tr}[\boldsymbol{S}\boldsymbol{G}_\mu] + \xi_\mu, \qquad \boldsymbol{G}_\mu \sim \mathrm{GOE}(d).$$

## B  NOISELESS SETTING: ERROR RATES AND SPECTRAL PROPERTIES

In the noiseless case $\Delta = 0$, we have the following rates.

**Result 4** (Noiseless rates). *Under the setting of Section 1 for $\Delta = 0$ and $n_{\mathrm{eff}} \gg 1$, the excess risk satisfies*

$$R = \begin{cases} \Theta\left(n_{\mathrm{eff}}^{1-2\gamma}\right) & \text{if} \quad n_{\mathrm{eff}} \ll d \quad \text{and} \quad \lambda \ll \frac{1}{d^{1/2}n_{\mathrm{eff}}^{\gamma-1}} \\ \Theta\left(\lambda^2 d^2 / n_{\mathrm{eff}}^2\right) & \text{if} \quad n_{\mathrm{eff}} \gg d \quad \text{and} \quad \lambda \ll \frac{n_{\mathrm{eff}}}{d^{\gamma+1/2}} \\ \Theta\left((\lambda d^{1/2}/n_{\mathrm{eff}})^{2-1/\gamma}\right) & \text{if} \quad \max\left(\frac{1}{d^{1/2}n_{\mathrm{eff}}^{\gamma-1}}, \frac{n_{\mathrm{eff}}}{d^{\gamma+1/2}}\right) \ll \lambda \ll \frac{n_{\mathrm{eff}}}{d^{1/2}} \end{cases}, \qquad (23)$$

*and $R = \Theta(1)$ otherwise. In the diagonal network case, the first rate holds up to logarithmic factors that we specify in eq. (92).*

Note that as in the least-squares case, the noiseless rates are faster (Aubin et al., 2020; Berthier et al., 2020a; Cui et al., 2021). Perhaps more surprising, they are also universal, with no difference between the quadratic and diagonal networks. Moreover, the error rate $\Theta\left(n_{\mathrm{eff}}^{1-2\gamma}\right)$ agrees with Ben Arous et al. (2025) as in their setting there is no regularization.

**Corollary 2** (Optimal regularization and optimality of ERM, noiseless setting). *The optimal regularization satisfies*

$$\lambda_{\mathrm{opt}} = \begin{cases} \tilde{\Theta}(n_{\mathrm{eff}}^{-\gamma+1}d^{-1/2}), & \text{if} \quad \Delta = 0 \quad \text{and} \quad 1 \ll n_{\mathrm{eff}} \ll d \\ 0^+ & \text{if} \quad \Delta = 0 \quad \text{and} \quad n_{\mathrm{eff}} \gg d \end{cases}. \qquad (24)$$

*where $\tilde{\Theta}$ is up to logarithm factors in the argument. The excess risk rates in the optimally regularized case match Bayesian risks, given by*

$$\mathsf{R}_{n_{\mathrm{eff}},d}(\lambda_{\mathrm{opt}}) = \Theta(\mathsf{R}_{n_{\mathrm{eff}}}^{\mathrm{BO}}) = \begin{cases} \Theta(n_{\mathrm{eff}}^{1-2\gamma}) & \text{if} \quad \Delta = 0 \quad \text{and} \quad 1 \ll n_{\mathrm{eff}} \ll d \\ 0 & \text{if} \quad \Delta = 0 \quad \text{and} \quad n_{\mathrm{eff}} \gg d \end{cases}, \qquad (25)$$

*and $\mathsf{R}_{n_{\mathrm{eff}},d} = \Theta(1)$ otherwise. Again, in the diagonal network case, the first and third regimes the rate hold up to logarithmic factors that we specify in eq. (92).*

The phase diagram for the noiseless case is simpler, and exhibits just a subset of the phases. Specifically, phases I, II are the same as the noisy cases. For the other two phases, we have

- III (Heavy-tail): The spectrum is a perturbed and shifted version of the target's spectrum. The perturbation is of order $\mathcal{O}(\lambda d^3/n^{3/2})$, and the shift is of about $\frac{\lambda d^2}{4n}$. Thus the risk is mainly caused by the shift, which gives an error of $\Theta(\lambda^2 d^4/n^2)$. When $\lambda = 0$, the spectrum is the same as the target's, so the risk is zero.

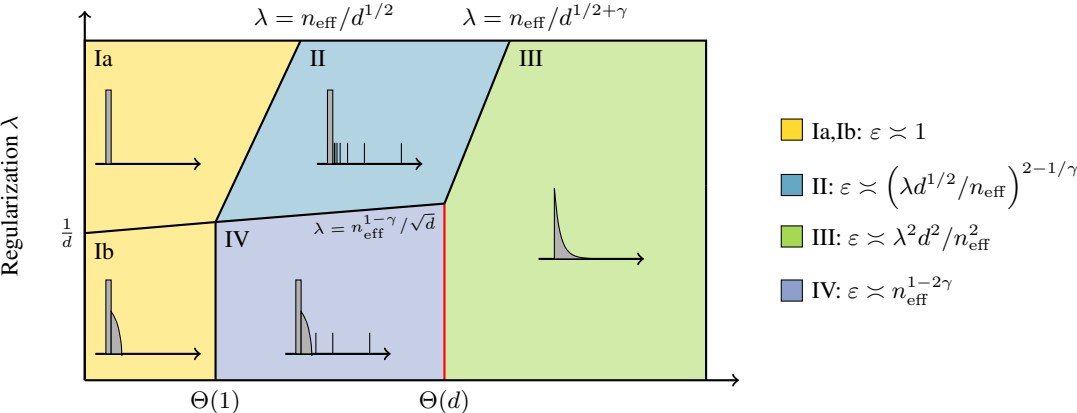

Effective number of samples $n_{\text{eff}}$ ($n$ for diagonal, $n/d$ for quadratic)

Figure 4: Excess risk rates of result 4 for the noiseless task ($\Delta = 0$), the corresponding spectral properties of neural networks.

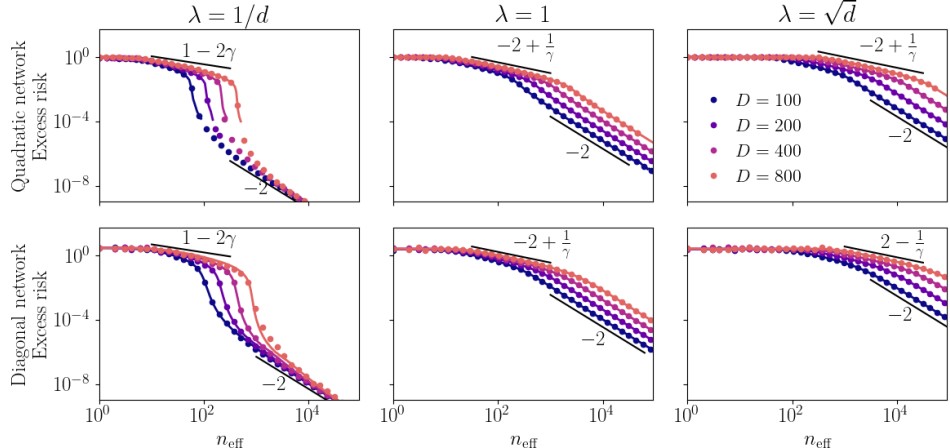

Figure 5: Excess risk in simulations versus non-asymptotic state evolution (solid lines) as a function of $n_{\text{eff}}$. The setting is the same as Figure 3 except that $\Delta = 0$. Black lines indicate the decay rates of the excess risk predicted by Result 4, again showing good agreement.

- IV (Bulk+Outliers): The bulk is of size $\Theta(d^{\gamma-3/10}/n^{\gamma-2/5})$, and there are approximately $\frac{n}{d}$ outliers. As $n$ increases, the bulk shrinks, all spikes come out of the bulk one by one, and the risk keeps decreasing. The contribution of the bulk is always negligible, so Result 3 implies that the risk is proportional to the total power of the unlearned spikes: $R = \frac{1}{d}\sum_{i \geq n/d}(\sqrt{d}i^{-\gamma})^2 = \Theta((d/n)^{2\gamma-1})$. Note that this is also information theoretically optimal, because with $n$ data points, we could learn at most $\frac{n}{d}$ uncorrelated directions.

The Result 4 are confirmed by Figure 5.

## C DERIVATION DETAILS - DIAGONAL LINEAR NETWORK

Considering the reparametrization defined in Section 1 and detailed in Appendix A, mapping empirical risk minimization with $L_2$ penalty on a two-layer diagonal linear network to LASSO regression,

in this section we study the supervised learning problem

$$\hat{\boldsymbol{\theta}} = \arg\min_{\boldsymbol{\theta} \in \mathbb{R}^d} \left\{ \frac{1}{2} \sum_{\mu=1}^n \left( y_\mu - \frac{\langle \boldsymbol{x}_\mu, \boldsymbol{\theta} \rangle}{\sqrt{d}} \right) + \lambda \|\boldsymbol{\theta}\|_1 \right\}, \tag{26}$$

with $\boldsymbol{x}_\mu \sim \mathcal{N}(\boldsymbol{0}, \boldsymbol{I}_d)$, and

$$y_\mu = \frac{\langle \boldsymbol{x}_\mu, \boldsymbol{\theta}^\star \rangle}{\sqrt{d}} + \sqrt{\Delta}\xi_\mu, \qquad \xi_\mu \sim \mathcal{N}(0,1), \qquad \mu = 1, \ldots, n \tag{27}$$

$$\boldsymbol{\theta}^\star \sim \mathcal{N}(\boldsymbol{0}, \boldsymbol{\Lambda}), \qquad \Lambda_{ij} = \delta_{ij}\Lambda_i, \qquad i = 1, \ldots, d \tag{28}$$

We also define the parameter $Q^\star = d^{-1}\operatorname{Tr}\boldsymbol{\Lambda} \xrightarrow{d\to\infty} \zeta(2\gamma)$. Without loss of generality we choose $\Lambda_1 \geq \Lambda_2 \geq \ldots \geq \Lambda_d$. The excess risk is defined as

$$R(\hat{\boldsymbol{\theta}}) = \frac{1}{d}\mathbb{E}[(\boldsymbol{x}^T\hat{\boldsymbol{\theta}} - \boldsymbol{x}^T\boldsymbol{\theta}^\star)^2]. \tag{29}$$

In sections C.1 and C.3 we derive the state evolution equations 48) for the excess risk of the ERM estimator eq. (26) and (107) for the Bayes-optimal estimator in the high-dimensional limit $n, d \to \infty$ with $n/d$ and $\lambda$ fixed. Then, in sections C.4 and C.2, assuming the excess risk equations holds for arbitrary scaling between dimensions and regularization, we derive the Results in Section 2.1.

## C.1 GENERALIZED APPROXIMATE MESSAGE PASSING AND STATE EVOLUTION

Our theory is built on the analysis of *Generalized Approximate Message Passing* (GAMP) algorithms, tailored for Bayes-optimal estimation and (convex) empirical risk minimization. In this section we provide an overview of the derivation of the expressions for $R$ and the LASSO $R$ in our setting from the GAMP framework.

Consider the matrix $\boldsymbol{X} \in \mathbb{R}^{n \times d}$, with i.i.d. Gaussian components $X_{ij} \sim \mathcal{N}(0,1)$, the vectors $\boldsymbol{b}^t \in \mathbb{R}^d$, $\boldsymbol{\omega}^t \in \mathbb{R}^n$, and the functions (known as *denoisers*) $f_t : \mathbb{R}^d \to \mathbb{R}^d$ and $g_t : \mathbb{R}^n \to \mathbb{R}^n$, with $t \geq 1$. The generic form of GAMP (Donoho et al., 2009; Rangan, 2011) is given by

$$\boldsymbol{\omega}^t = \boldsymbol{X} f_t(\boldsymbol{b}^t) - v_t g_{t-1}(\boldsymbol{\omega}^{t-1}), \tag{30}$$

$$\boldsymbol{b}^{t+1} = \boldsymbol{X}^T g_t(\boldsymbol{\omega}^t) + a_t f_t(\boldsymbol{b}^t). \tag{31}$$

The terms $a_t$ and $v_t$ are known in the statistical physics literature as *Onsager terms*, and they are defined as

$$a_t = -\frac{1}{d}\sum_{\mu=1}^n \frac{\partial}{\partial \omega_i} g_t(\boldsymbol{\omega}), \qquad v_t = \frac{1}{d}\sum_{i=1}^d \frac{\partial}{\partial b_i} f_t(\boldsymbol{b}). \tag{32}$$

For separable denoiser functions[*], one can track statistics of the iterated vectors $\boldsymbol{b}^t$, $\boldsymbol{\omega}^t$, leveraging well-known results from Bayati & Montanari (2011); Javanmard & Montanari (2013), through the so called *state evolution*.

### C.1.1 GAMP FOR CONVEX OPTIMIZATION

Consider the problem of minimizing the empirical risk with loss $\ell(y, z)$ convex in the second argument and convex penalty $r(\theta)$,

$$\arg\min_{\boldsymbol{\theta} \in \mathbb{R}^d} \sum_{\mu=1}^n \ell(y^\mu, \boldsymbol{\theta}^T \boldsymbol{x}^\mu) + \sum_{i=1}^d r(\theta_i), \tag{33}$$

It is possible to design a GAMP algorithm whose fixed points are solutions to the problem defined in (33). A detailed discussion on this approach can be found in Feng et al. (2022). Define the functions

$$\overline{g}(\omega, y, v) := \operatorname{prox}_{v\ell(y,\cdot)}(\omega), \qquad g(\omega, y, v) = \frac{\overline{g}(\omega, y, v) - \omega}{v} \tag{34}$$

$$f(b, a) := \operatorname{prox}_{\frac{1}{a}r}\left(\frac{b}{a}\right), \tag{35}$$

---

[*]$f : \mathbb{R}^d \to \mathbb{R}^d$ is separable if $\forall i \in \{1, \ldots, d\} : [f(\boldsymbol{b} \in \mathbb{R}^d)]_i = f_i(b_i)$, for some scalar function $f_i : \mathbb{R} \to \mathbb{R}$

where the *proximal operator* of a convex function $f$ is defined as

$$\text{prox}_f(x) = \arg\min_{z \in \mathbb{R}} \left\{ f(z) + \frac{1}{2}(z-x)^2 \right\}. \tag{36}$$

Then, Proposition 4.4 in Feng et al. (2022), guarantees that, given a fixed point $(\boldsymbol{\omega}, \boldsymbol{b})$ of the GAMP algorithm eq. (30,31) with denoiser functions $g_t(\omega) = g(\omega, y, v_t)$ and $f_t(b) = f(b, a_t)$, the vector $\hat{\boldsymbol{\theta}} := f_t(\boldsymbol{b})$ is the unique minimizer of (33).

As mentioned, in the high-dimensional limit $n, d \to \infty$, with $n/d$ fixed, we can track statistics of the iterated variables through a set of state evolution equations. We stress that the following result hold for the considered linear target function $y = \langle (\boldsymbol{\theta}^\star)^T, \boldsymbol{x} \rangle / \sqrt{d}$.

**Theorem 1** (Bayati & Montanari (2011); Javanmard & Montanari (2013), informal)**.** *Define*

$$\begin{cases} \hat{q}^t &= \frac{n}{d}\mathbb{E}_{(z,\omega_t)}[g(\omega_t, z, v_t)^2] \\ \hat{m}^t &= \frac{n}{d}\mathbb{E}_{(z,\omega_t)}[\partial_z g(\omega_t, z, v_t)] \\ \hat{v}^t &= -\frac{n}{d}\mathbb{E}_{(z,\omega_t)}[\partial_\omega g(\omega_t, z, v_t)] \end{cases} \quad \begin{cases} q^{t+1} &= \frac{1}{d}\mathbb{E}_{(\boldsymbol{\xi},\boldsymbol{\theta}^\star)}[\|f(\sqrt{\hat{q}^t}\boldsymbol{\xi} + \hat{m}^t\boldsymbol{\theta}^\star, \hat{v}^t)\|^2] \\ m^{t+1} &= \frac{1}{d}\mathbb{E}_{(\boldsymbol{\xi},\boldsymbol{\theta}^\star)}[\langle f(\sqrt{\hat{q}^t}\boldsymbol{\xi} + \hat{m}^t\boldsymbol{\theta}^\star, \hat{v}^t), \boldsymbol{\theta}^\star \rangle] \\ v^{t+1} &= \frac{1}{d}\mathbb{E}_{(\boldsymbol{\xi},\boldsymbol{\theta}^\star)}[\nabla_b \cdot f(\sqrt{\hat{q}^t}\boldsymbol{\xi} + \hat{m}^t\boldsymbol{\theta}^\star, \boldsymbol{\theta}^\star, \hat{v}^t)] \end{cases} \tag{37}$$

*where $\boldsymbol{\xi} \sim \mathcal{N}(\mathbf{0}, \boldsymbol{I}_d)$ and*

$$\begin{pmatrix} z \\ \omega^t \end{pmatrix} \sim \mathcal{N}\left( \begin{pmatrix} 0 \\ 0 \end{pmatrix}, \begin{pmatrix} Q^\star & m^t \\ m^t & q^t \end{pmatrix} \right). \tag{38}$$

*Then the iterated vectors $\boldsymbol{\omega}^t$ and $\boldsymbol{b}^t$ of the GAMP algorithm (30,31), with denoiser functions (34,35), respectively converge weakly to the Gaussian vectors $\boldsymbol{\Omega}^t = \sqrt{q^t - m^t}\boldsymbol{w} + m^t\boldsymbol{X}\boldsymbol{\theta}^\star$ (with $\boldsymbol{w} \sim \mathcal{N}(\mathbf{0}, \boldsymbol{I}_n)$) and $\boldsymbol{B}^t = \hat{m}^t\boldsymbol{\theta}^\star + \sqrt{\hat{q}^t}\boldsymbol{\xi}$, in the sense that, for any uniformly pseudo-Lipshitz of order $k$, deterministic $\phi : (\mathbb{R}^d \times \mathbb{R}^n)^t \times \mathbb{R}^d \to \mathbb{R}$,*

$$\phi(\boldsymbol{b}^0, \boldsymbol{\omega}^0, \boldsymbol{b}^1, \boldsymbol{\omega}^1, \ldots, \boldsymbol{\omega}^{t-1}, \boldsymbol{b}^t) \stackrel{\text{P}}{\simeq} \mathbb{E}\phi(\boldsymbol{B}^0, \boldsymbol{\Omega}^0, \boldsymbol{B}^1, \boldsymbol{\Omega}^1, \ldots, \boldsymbol{\Omega}^{t-1}, \boldsymbol{B}^t). \tag{39}$$

The previous theorem readily implies that, in the high-dimensional limit, $a_t \simeq \hat{v}^t$, and, given $\hat{\boldsymbol{\theta}}^t \simeq f(\boldsymbol{b}^t, a_t)$,

$$\frac{1}{d}\langle \hat{\boldsymbol{\theta}}^t, \boldsymbol{\theta}^\star \rangle \simeq m^t, \qquad \frac{1}{d}\|\hat{\boldsymbol{\theta}}\|^2 \simeq q^t \tag{40}$$

and the generalization error of the estimator $\hat{y}(\boldsymbol{x}) = f(\boldsymbol{x}, \hat{\boldsymbol{\theta}}^t) = (\langle \hat{\boldsymbol{\theta}}^t, \boldsymbol{x} \rangle / \sqrt{d}$ is

$$R(\hat{\boldsymbol{\theta}}^t) \simeq \mathbb{E}_{\boldsymbol{x}}\left( \frac{\langle \hat{\boldsymbol{\theta}}^t, \boldsymbol{x} \rangle}{\sqrt{d}} - \frac{\langle \hat{\boldsymbol{\theta}}^t, \boldsymbol{x} \rangle}{\sqrt{d}} \right)^2 = Q^\star - 2m^t + q^t. \tag{41}$$

**LASSO regression**   In the case of LASSO, with $\ell(y, z) = (y-z)^2/2$ and $r(\theta) = \lambda|\theta|$, we have that

$$g(\omega, y, v) = \frac{y - \omega}{1 + v}, \qquad f(b, a) = \frac{1}{a}\text{ST}_\lambda(b), \tag{42}$$

where $\text{ST}_\lambda(b) = \max(b - \lambda, 0) - \max(-b - \lambda, 0)$ denotes the soft-thresfolding function. The state evolution equations in this setting read

$$\begin{cases} \hat{q}^t &= \frac{n(\Delta + Q^\star - 2m^t + q^t)}{d(1+v^t)^2} \\ \hat{m}^t &= \frac{n}{d(1+v^t)} \\ \hat{v}^t &= \frac{n}{d(1+v^t)} \end{cases} \quad \begin{cases} m^{t+1} &= \frac{1}{d}\sum_{i=1}^d \Lambda_k \, \text{erfc}\left( \frac{\lambda}{\sqrt{2((\hat{m}^t)^2\Lambda_i + \hat{q}^t)}} \right) \\ v^{t+1} &= \frac{1}{d\hat{m}^t}\sum_{i=1}^d \text{erfc}\left( \frac{\lambda}{\sqrt{2((\hat{m}^t)^2\Lambda_i + \hat{q}^t)}} \right) \end{cases} \tag{43}$$

and

$$q^{t+1} = \frac{1}{d(\hat{m}^t)^2}\sum_{i=1}^d \left[ ((\hat{m}^t)^2\Lambda_i + \hat{q}^t + \lambda^2) \, \text{erfc}\left( \frac{\lambda}{\sqrt{2((\hat{m}^t)^2\Lambda_i + \hat{q}^t)}} \right) \right] \tag{44}$$

$$- \frac{1}{d(\hat{m}^t)^2}\sum_{i=1}^d \left[ \frac{2\lambda}{\sqrt{2\pi}}\sqrt{(\hat{m}^t)^2\Lambda_i + \hat{q}^t}\, e^{-\lambda^2/(2((\hat{m}^t)^2\Lambda_i + \hat{q}^t))} \right]. \tag{45}$$

At convergence, substituting the equation for $v$ into the equation for one for $\hat{m}$, introducing the parameter $\nu = \frac{\lambda}{\hat{m}}\sqrt{\frac{n}{2d}}$ and leveraging eq. (41), one obtains that the excess risk for LASSO regression in this setting is given by

$$
R(\hat{\boldsymbol{\theta}}) \simeq \mathsf{R}_{n,d}(\nu) = \frac{1}{n}\sum_{i=1}^{d}\left[\frac{n}{d}\Lambda_i\,\mathrm{erf}\left(\frac{\nu}{\sqrt{\frac{n}{d}\Lambda_i+\hat{\Delta}}}\right) + \left(\hat{\Delta}+2\nu^2\right)\mathrm{erfc}\left(\frac{\nu}{\sqrt{\frac{n}{d}\Lambda_i+\hat{\Delta}}}\right)\right]
$$
$$
- \frac{2\nu}{n\sqrt{\pi}}\sum_{i=1}^{d}\left[\sqrt{\frac{n}{d}\Lambda_i+\hat{\Delta}}\,e^{-\nu^2/(\frac{n}{d}\Lambda_i+\hat{\Delta})}\right], \tag{46}
$$

with $\hat{\Delta} = \Delta + \mathsf{R}_{n,d}(\nu)$ and

$$
\frac{\lambda}{\nu}\sqrt{\frac{n}{2d}} + \frac{1}{d}\sum_{i=1}^{d}\mathrm{erfc}\left(\frac{\nu}{\sqrt{\frac{n}{d}\Lambda_i+\hat{\Delta}}}\right) = \frac{n}{d}. \tag{47}
$$

For the specific choice of covariance $\Lambda_i = di^{-2\gamma}$, this becomes

$$
R(\hat{\boldsymbol{\theta}}) \simeq \mathsf{R}_{n,d}(\nu) = \frac{1}{n}\sum_{i=1}^{d}\left[ni^{-2\gamma}\,\mathrm{erf}\left(\frac{\nu}{\sqrt{ni^{-2\gamma}+\hat{\Delta}}}\right) + \left(\hat{\Delta}+2\nu^2\right)\mathrm{erfc}\left(\frac{\nu}{\sqrt{ni^{-2\gamma}+\hat{\Delta}}}\right)\right]
$$
$$
- \frac{2\nu}{n\sqrt{\pi}}\sum_{i=1}^{d}\left[\sqrt{ni^{-2\gamma}+\hat{\Delta}}\,e^{-\nu^2/(ni^{-2\gamma}+\hat{\Delta})}\right], \tag{48}
$$

and

$$
\frac{\lambda}{\nu}\sqrt{\frac{n}{2d}} + \frac{1}{d}\sum_{i=1}^{d}\mathrm{erfc}\left(\frac{\nu}{\sqrt{ni^{-2\gamma}+\hat{\Delta}}}\right) = \frac{n}{d}. \tag{49}
$$

**Conjecture 2.** *Define $R(\hat{\boldsymbol{\theta}},\lambda)$ the excess risk eq.(29) of the LASSO estimator $\hat{\boldsymbol{\theta}}$ eq. (26) with regularization strength $\lambda$. Then, there exists $C > 0$ such that, for any $n, d > C$, with probability $1 - o_n(1) - o_d(1)$,*

$$
|R(\hat{\boldsymbol{\theta}},\lambda) - \mathsf{R}_{n,d}(\nu(\lambda))| = \mathsf{R}_{n,d}(\nu(\lambda))\cdot o_{n,d}(1), \tag{50}
$$

*with $\mathsf{R}_{n,d}(\nu)$ defined in eq. (48) and $\nu(\lambda)$ solution of eq. (49).*

### C.1.2 SPECTRAL STRUCTURE OF THE ESTIMATOR

Theorem 1 readily implies Result 2. Given the unique fixed point $(\boldsymbol{\omega}, \boldsymbol{b})$ of GAMP, the minimizer of eq. (5) is given by $\hat{\theta} = \frac{1}{\hat{m}}\mathrm{ST}_\lambda(\boldsymbol{b})$, which satisfies, in distribution

$$
\hat{\theta}_i \sim \mathrm{ST}_{\epsilon_\mathrm{d}}(\theta_i^* + \delta_\mathrm{d}z_i), \tag{51}
$$

with $\epsilon_\mathrm{d} := \lambda/\hat{m}$, $\delta_\mathrm{d} := \sqrt{\hat{q}}/\hat{m}$ and $z_i \sim \mathcal{N}(0, 1)$. Note that $\epsilon_\mathrm{d} = \nu\sqrt{2d/n}$ and

$$
|\hat{\theta}_i| \sim \max\left(\left|\underbrace{\left\|\theta_i^\star + z_i\sqrt{\hat{\Delta}\frac{d}{n}}\right\|}_{=:u_i} - \nu\sqrt{\frac{2d}{n}}\right|, 0\right). \tag{52}
$$

The random variable variable $u_i$ satisfies

$$
u_i \sim \mathcal{N}\left(0, \Lambda_i + \hat{\Delta}\frac{d}{n}\right) \implies u_i = \begin{cases} \theta_i^\star + o_p(\sqrt{\Lambda_i}), & \text{if } n\Lambda_i \gg d\hat{\Delta}, \\ z_i\sqrt{\hat{\Delta}\frac{d}{n}}(1 + o_p(1)) & \text{if } n\Lambda_i \ll d\hat{\Delta}, \end{cases} \tag{53}
$$

where we say that $u_i$ corresponds to the signal component $\theta_{\star i}$ in probability when the signal variance $\Lambda_i$ dominates over the effective noise variance $\hat{\Delta}d/n$; conversely, the effective noise

dominates $u_i$.

Therefore, the ensemble $\{u_i\}_{i|n\Lambda_i < d\hat{\Delta}}$ constitutes a "bulk" of i.i.d. Gaussian variables, representing the combined effect of label noise and the limited number of samples. In fact, if the sample size is large enough, namely $n \gg d\hat{\Delta}\Lambda_d^{-1}$, the effect of the noise becomes undetectable. We refer to the remaining $\{u_i\}_{i|n\Lambda_i > d\hat{\Delta}}$ as "spikes", representing the components of the true signal $\boldsymbol{\theta}^\star$ that we want to learn. Therefore, the scale $i_{\hat{\Delta}}$ such that $n\Lambda_{i_{\hat{\Delta}}} \sim d\hat{\Delta}$ represents the number of "learnable" components. Note that in the diagonal linear network setting, the noise bulk has no sharp boundaries, and the signal components (or spikes) are themselves random variables. We adopt this terminology in parallel with the quadratic network case, but when $n\Lambda_i$ is comparable to $d\hat{\Delta}$, the signal and noise contribute equally to the variance of $u_i$, so a spike cannot be distinguished from the bulk. However, for clarity of exposition, we will nevertheless classify the indices according to whether $n\Lambda_i$ is larger or smaller than $d\hat{\Delta}$, acknowledging that this introduces an asymptotically negligible ambiguity near the crossover.

The $i^{\text{th}}$ components of the LASSO estimator are then obtained by soft-thresholding the variables $u_i$, with the parameter $\epsilon_{\text{d}}$ representing a cutoff that induces sparsity in the estimator, setting the smallest values to zero. Note that the cutoff depends on the regularization strength $\lambda$ only through $\nu$.

At this level, we can distinguish the following scenarios: in terms of number of data,

| spikes + bulk | only spikes |
|:---:|:---:|
| $n\Lambda_d \ll d\hat{\Delta}$ | $n\Lambda_d \gg d\hat{\Delta}$ |
| not all components can be learned | all components can be learned |

in terms of thresholding strength,

| weak | strong | extreme |
|:---:|:---:|:---:|
| $\nu^2 \ll \max(n\Lambda_d/d, \hat{\Delta})$ | $\max(n\Lambda_d/d, \hat{\Delta}) \ll \nu^2 \ll n\Lambda_1/d$ | $\nu^2 \gg n\Lambda_1/d$ |
| cutoff below spikes: | cutoff between spikes: | cutoff above all spikes: |
| all learnable signal is learned | signal is partially learned, noise is filtered | nothing is learned |

**Error decomposition** We can interpret the excess risk expression in eq. (46) in light of these considerations on the interplay between signal, noise bulk and regularization, by splitting the sum at the relevant scales identified in this section, namely $i_\nu$ such that $n\Lambda_{i_\nu} = d\nu^2$ and $i_{\hat{\Delta}}$ such that $n\Lambda_{i_{\hat{\Delta}}} = d\hat{\Delta}$. Note that the extreme regularization case $\nu^2 \gg \sqrt{n}$ is trivial, as the excess risk $\mathsf{R} \approx \sum_{i=1}^d i\Lambda_i/d$, *i.e.* it is dominated by the underfitting of all the signal components.

1. Weak thresholding: the excess risk can be decomposed as $\mathsf{R} \approx \mathsf{R}_{\text{under}} + \mathsf{R}_{\text{over}} + \mathsf{R}_{\text{app}}$

   • Underfitting term (signal components not learned)

$$\mathsf{R}_{\text{under}} = \frac{1}{d} \sum_{i_{\hat{\Delta}}+1}^d \Lambda_i \operatorname{erf}\left(\frac{\nu}{\sqrt{\hat{\Delta}}}\right) \approx \frac{1}{d} \sum_{i_{\hat{\Delta}}+1}^d \Lambda_i, \tag{54}$$

   • Overfitting term (learned noise)

$$\mathsf{R}_{\text{over}} = \frac{1}{n} \sum_{i=i_{\hat{\Delta}}+1}^d \left[\left(\hat{\Delta} + 2\nu^2\right) \operatorname{erfc}\left(\frac{\nu}{\sqrt{\hat{\Delta}}}\right) - \frac{2\nu}{\sqrt{\pi}}\sqrt{\hat{\Delta}}\, e^{-\nu^2/\hat{\Delta}}\right], \tag{55}$$

$$= \frac{d - i_{\hat{\Delta}} - 1}{n} \mathbb{E}\left[\left(\operatorname{ST}_{\varepsilon_{\text{d}}}\left(z_i\sqrt{\frac{\hat{\Delta}d}{n}}\right)\right)^2\right], \tag{56}$$

- Approximation term (learned signal)

$$\mathsf{R}_{\text{app}} = \frac{1}{n} \sum_{i=1}^{i_{\hat{\Delta}}} \left[ \frac{n}{d} \Lambda_i \operatorname{erf} \left( \frac{\nu}{\sqrt{\frac{n}{d}\Lambda_i + \hat{\Delta}}} \right) + \left( \hat{\Delta} + 2\nu^2 \right) \operatorname{erfc} \left( \frac{\nu}{\sqrt{\frac{n}{d}\Lambda_i + \hat{\Delta}}} \right) \right] \tag{57}$$

$$- \frac{2\nu}{n\sqrt{\pi}} \sum_{i=1}^{i_{\hat{\Delta}}} \left[ \sqrt{\frac{n}{d}\Lambda_i + \hat{\Delta}} \, e^{-\nu^2/(\frac{n}{d}\Lambda_i + \hat{\Delta})} \right] \tag{58}$$

2. Strong thresholding: the excess risk can be decomposed as $\mathsf{R} \approx \mathsf{R}_{\text{under}} + \mathsf{R}_{\text{app}}$

- Underfitting term (signal components not learned)

$$\mathsf{R}_{\text{under}} = \frac{1}{d} \sum_{i_\nu+1}^{d} \Lambda_i \operatorname{erf} \left( \frac{\nu}{\sqrt{n\Lambda_i}} \right) \approx \frac{1}{d} \sum_{i_\nu+1}^{d} \Lambda_i, \tag{59}$$

- Approximation term (learned signal)

$$\mathsf{R}_{\text{app}} = \frac{1}{n} \sum_{i=1}^{i_\nu} \left[ \frac{n}{d} \Lambda_i \operatorname{erf} \left( \frac{\nu}{\sqrt{\frac{n}{d}\Lambda_i + \hat{\Delta}}} \right) + \left( \hat{\Delta} + 2\nu^2 \right) \operatorname{erfc} \left( \frac{\nu}{\sqrt{\frac{n}{d}\Lambda_i + \hat{\Delta}}} \right) \right] \tag{60}$$

$$- \frac{2\nu}{n\sqrt{\pi}} \sum_{i=1}^{i_\nu} \left[ \sqrt{\frac{n}{d}\Lambda_i + \hat{\Delta}} \, e^{-\nu^2/(\frac{n}{d}\Lambda_i + \hat{\Delta})} \right] \tag{61}$$

In Section C.2 we observe that these are the relevant scales for computing the leading order terms of the excess risk and its scaling laws. Moreover, we estimate the values of $\nu$ and $\hat{\Delta}$ as functions of $n, d, \lambda$. These results, together with the interpretation provided in this section, lead to the identification of the phase diagram regions in fig. 1 and the phases descriptions in Section 2.2.

## C.2  LASSO SCALING LAWS

From eq. (48),

$$\mathsf{R} = \frac{1}{n} \sum_{i=1}^{d} [f_1(x_i) + f_2(x_i) - f_3(x_i)], \tag{62}$$

with $x_i := i n^{-1/(2\gamma)}$ and

$$f_1(x) = x^{-2\gamma} \operatorname{erf} \left( \frac{\nu}{\sqrt{x^{-2\gamma} + \hat{\Delta}}} \right) \tag{63}$$

$$f_2(x) = (\hat{\Delta} + 2\nu^2) \operatorname{erfc} \left( \frac{\nu}{\sqrt{x^{-2\gamma} + \hat{\Delta}}} \right) \tag{64}$$

$$f_3(x) = \frac{2}{\sqrt{\pi}} \nu \sqrt{x^{-2\gamma} + \hat{\Delta}} \exp \left( -\frac{\nu^2}{x^{-2\gamma} + \hat{\Delta}} \right) \tag{65}$$

We observe that the leading order of the functions changes scale around $x \sim \hat{\Delta}^{-1/(2\gamma)}$.
For $x^{-2\gamma} \gg \hat{\Delta}$

$$f_1(x) \approx x^{-2\gamma} \operatorname{erf} \left( \nu x^\gamma \right) \tag{66}$$

$$f_2(x) \approx (\hat{\Delta} + 2\nu^2) \operatorname{erfc} \left( \nu x^\gamma \right) \tag{67}$$

$$f_3(x) \approx \frac{2}{\sqrt{\pi}} \nu x^{-\gamma} \exp \left( -\nu^2 x^{2\gamma} \right) \tag{68}$$

For $x^{-2\gamma} \ll \hat{\Delta}$

$$f_1(x) \approx x^{-2\gamma} \operatorname{erf}\left(\nu\hat{\Delta}^{-1/2}\right) \tag{69}$$

$$f_2(x) \approx (\hat{\Delta} + 2\nu^2)\operatorname{erfc}\left(\nu\hat{\Delta}^{-1/2}\right) \tag{70}$$

$$f_3(x) \approx \frac{2}{\sqrt{\pi}}\nu\hat{\Delta}^{1/2}\exp\left(-\nu^2\hat{\Delta}^{-1}\right). \tag{71}$$

Note that the scale $x_i^{-2\gamma} \sim \hat{\Delta}$ corresponds precisely to the detectability threshold of signal components observed in Section C.1.2 for the LASSO estimator and in Section C.4 for the Bayes-optimal estimator.

We compute the excess risk $R$ from eq. (48) at leading order as a function of the parameter $\nu$, by splitting the sums at the crossover scales. Note that this procedure corresponds to the decomposition of the excess risk into approximation, overfitting and underfitting errors that we have identified in Section C.1.2. Afterwards, solving the self-consistent eq. (49) for $\nu$, we derive the Results in Section 2.1.

The three main regimes are

$$\begin{cases} \nu \gg x_1^\gamma \implies \nu^2 \gg n, \\ \sqrt{\max(x_d^{-2\gamma}, \hat{\Delta})} \ll \nu \ll x_1^\gamma \implies \max(nd^{-2\gamma}, \hat{\Delta}) \ll \nu^2 \ll n, \\ \nu \ll \sqrt{\max(x_d^{-2\gamma}, \hat{\Delta})} \implies \nu^2 \ll \max(nd^{-2\gamma}, \hat{\Delta}), \end{cases} \tag{72}$$

which are the extreme, strong and weak thresholding phases we have identified in Section C.1.2, for the specific choice of covariance $\Lambda_i = di^{-2\gamma}$.

In what follows we will often use the expansions

$$\operatorname{erf}(x \ll 1) = \frac{2}{\sqrt{\pi}}x + o(x), \tag{73}$$

$$\operatorname{erfc}(x \gg 1) = \frac{e^{-x^2}}{x\sqrt{\pi}}\left(1 - \frac{1}{2x^2} + \frac{3}{4x^4} + o(x^{-4})\right). \tag{74}$$

**Extreme thresholding**    For $\nu^2 \gg n$, the dominant term is given by underfitting

$$\frac{1}{n}\sum_{i=1}^d f_1(x_i) \approx \frac{1}{n}\sum_{i=1}^d (x_i)^{-2\gamma} \approx \zeta(2\gamma), \tag{75}$$

while the remaining terms are negligible

$$\frac{1}{n}\sum_{i=1}^d f_2(x_i) - f_3(x_i) \approx \frac{\hat{\Delta}}{\nu\sqrt{n\pi}} \sum_{i=1}^{\min(d,\lfloor(\hat{\Delta}/n)^{-1/(2\gamma)}\rfloor)} i^{-\gamma}e^{-i^{2\gamma}\nu^2/n} \tag{76}$$

$$\approx \frac{\hat{\Delta}}{\nu\sqrt{n\pi}}\exp(-\nu^2/n). \tag{77}$$

As expected, in this regime the large soft-thresholding parameter forces the components of the estimator to zero, and $R \approx \zeta(2\gamma)$.

**Strong thresholding**    If instead $\max(nd^{-2\gamma}, \hat{\Delta}) \ll \nu^2 \ll n$, defining $i_\nu := \lfloor(n/\nu^2)^{1/(2\gamma)}\rfloor$ and $i_{\hat{\Delta}} := \lfloor(n/\hat{\Delta})^{1/(2\gamma)}\rfloor$, we split the sums into four terms:

- approximation error:

$$\text{i)} \frac{1}{n} \sum_{i=1}^{i_\nu} f_1(x_i) - f_3(x_i) \approx \frac{2\nu}{\sqrt{n\pi}} \sum_{i=1}^{i_\nu} (1 - \exp(-i^{2\gamma}\nu^2/n))i^{-\gamma}$$

$$\approx \frac{2\nu^3}{n\sqrt{n\pi}} \sum_{i=1}^{i_\nu} i^\gamma$$

$$\approx \frac{2}{(1+\gamma)\sqrt{\pi}} \left(\frac{n}{\nu^2}\right)^{-1+1/(2\gamma)}$$

$$\text{ii)} \frac{1}{n} \sum_{i=1}^{i_\nu} f_2(x_i) \approx \frac{2\nu^2 + \hat{\Delta}}{n} \sum_{i=1}^{i_\nu} \left[1 - \text{erf}(i^\gamma\nu/\sqrt{n})\right]$$

- underfitting error:

$$\text{iii)} \frac{1}{n} \sum_{i=i_\nu+1}^{d} f_1(x_i) \approx \sum_{i=i_\nu+1}^{d} i^{-2\gamma} \approx \frac{1}{2\gamma-1} \left(\frac{n}{\nu^2}\right)^{-1+1/(2\gamma)}$$

$$\approx 2\left(\frac{n}{\nu^2}\right)^{-1+1/(2\gamma)} + \hat{\Delta}\nu^{-1/\gamma}n^{-1+1/(2\gamma)} + \Theta\left(\left(\frac{n}{\nu^2}\right)^{-1+1/(2\gamma)}\right)$$

$$\text{iv)} \frac{1}{n} \sum_{i=i_\nu+1}^{d} f_2(x_i) - f_3(x_i) \approx \frac{\hat{\Delta}}{\nu\sqrt{n\pi}} \sum_{i=i_\nu+1}^{\min(d,i_{\hat{\Delta}})} i^{-\gamma}e^{-i^{2\gamma}\nu^2/n} + \mathbf{1}_{[\hat{\Delta}>nd^{-2\gamma}]}\frac{d\hat{\Delta}^{5/2}}{n\nu^3}e^{-\nu^2/\hat{\Delta}}$$

$$\overset{\text{Laplace}}{\approx} \frac{\hat{\Delta}}{e\sqrt{\gamma(2\gamma-1)}}n^{-1+1/(2\gamma)}\nu^{-1/\gamma} + \mathbf{1}_{[\hat{\Delta}>nd^{-2\gamma}]}\frac{d\hat{\Delta}^{5/2}}{n\nu^3}\exp\left(-\frac{\nu^2}{\hat{\Delta}}\right)$$

where in the last step we have approximated the (Riemann) sum by an integral which we solved using the Laplace's method, that is (informally)

$$\int_a^b h(x)e^{Mg(x)}\mathrm{d}x \overset{M\gg 1}{\approx} \sqrt{\frac{2\pi}{M|g''(x_0)|}}h(x_0)e^{Mg(x_0)}, \qquad x_0 = \arg\max_{x\in[a,b]} g(x), \ \ g''(x) \le 0 \forall x \in [a,b]. \tag{78}$$

The dominant term is $\mathsf{R} \asymp (n/\nu^2)^{-1+1/(2\gamma)}$.

**Weak thresholding**    Finally , if $\nu^2 \ll \max(nd^{-2\gamma}, \hat{\Delta})$, we split the sums into the following four terms:

- approximation error:

$$\text{i)} \frac{1}{n} \sum_{i=1}^{\min(i_{\hat{\Delta}},d)} f_1(x_i) - f_3(x_i) \approx \frac{2\nu^3}{(1+\gamma)\sqrt{n\pi}}\left(\mathbf{1}_{[\hat{\Delta}>nd^{-2\gamma}]}\left(\frac{n}{\hat{\Delta}}\right)^{(1+\gamma)/(2\gamma)} + \mathbf{1}_{[\hat{\Delta}<nd^{-2\gamma}]}d^{1+\gamma}\right)$$

$$\text{ii)} \frac{1}{n} \sum_{i=1}^{\min(d,i_{\hat{\Delta}})} f_2(x_i) \approx (2\nu^2 + \hat{\Delta})\min\left(n^{-1+1/(2\gamma)}\hat{\Delta}^{-1/(2\gamma)}, \frac{d}{n}\right) \tag{79}$$

- overfitting + underfitting errors:

$$\text{iii)} \frac{1}{n} \sum_{i=\min(d,i_{\hat{\Delta}}+1)}^{d} f_1(x_i) \approx \mathbf{1}_{[\hat{\Delta}>nd^{-2\gamma}]}\frac{2\nu}{\sqrt{\hat{\Delta}\pi}} \sum_{i=i_{\hat{\Delta}}+1}^{d} i^{-2\gamma}$$

$$\approx \mathbf{1}_{[\hat{\Delta}>nd^{-2\gamma}]}\frac{2\nu}{\sqrt{\hat{\Delta}\pi}}\left(\frac{n}{\hat{\Delta}}\right)^{-1+1/(2\gamma)}$$

$$\text{iv)} \frac{1}{n} \sum_{i=\min(d,i_{\hat{\Delta}}+1)}^{d} f_2(x_i) - f_3(x_i) \approx \mathbf{1}_{[\hat{\Delta}>nd^{-2\gamma}]}\left[(2\nu^2+\hat{\Delta})\left(\frac{d}{n} - n^{-1+1/(2\gamma)}\hat{\Delta}^{-1/(2\gamma)}\right)\right]$$

$$- \mathbf{1}_{[\hat{\Delta}>nd^{-2\gamma}]}\left[\frac{2}{\sqrt{\pi}}\nu\sqrt{\hat{\Delta}}\exp(-\nu^2/\hat{\Delta})\frac{d}{n}\right] \tag{80}$$

The dominant term is therefore $R \asymp (2\nu^2 + \hat{\Delta})d/n$.

Note that, for $\nu = \tilde{\Theta}(\hat{\Delta})$, even if the cutoff is larger than the noise variance, it is not large enough to completely filter all bulk terms. We will verify that this is the case in Phases IV and V of Fig. 1, for $\lambda \ll \sqrt{n/d}$ and $n \ll d$. The leading order terms in the excess error decomposition are equivalent to the ones we have computed in the previous paragraph for the **Strong thresholding** case. However, we must take into account the overfitting contribution from the bulk components $\frac{d\hat{\Delta}^{5/2}}{n\nu^3} \exp\left(-\frac{\nu^2}{\hat{\Delta}}\right)$ (term iv), which is otherwise negligible for $\nu^2/\hat{\Delta}$ larger than any polylogarithmic function of $n, d$.

We can now proceed with the solution of the self-consistent equation (49), in order to derive the closed-form expressions for the excess risk scaling laws.

**Noisy setting**  Since $\Delta > 0$, then, provided $R = O(1)$, $\hat{\Delta} = \Theta(1)$. Let $n \gg d$. Eq. (49) readily implies that $\nu \asymp \lambda\sqrt{d/n}$, as the second term on the left-hand side is bounded by 1. Therefore, for $n \gg d$,

$$
R = \begin{cases}
\Theta(1), & \lambda \gg n/\sqrt{d}, \\
\Theta\left(\frac{n^2}{\lambda^2 d}\right)^{-1+1/(2\gamma)}, & \max\left(nd^{-\gamma-1/2}, \sqrt{n/d}\right) \ll \lambda \ll n/\sqrt{d} \\
\Theta\left(\frac{d}{n}\max\left(1, \frac{\lambda^2 d}{n}\right)\right), & \lambda \ll \max\left(nd^{-\gamma-1/2}, \sqrt{n/d}\right)
\end{cases}
\tag{81}
$$

Let instead $n \ll d$. The second term on the left-hand side of eq. (49) is

$$
\frac{1}{d}\sum_{i=1}^{d} \text{erfc}\left(\frac{\nu}{\sqrt{x^{-2\gamma} + \hat{\Delta}}}\right) \approx \begin{cases}
\sqrt{\frac{n}{\pi}}\nu^{-1}\exp\left(-\nu^2/n\right), & \nu^2 \gg n, \\
2\frac{n^{1/(2\gamma)}}{d}\left(\nu^{2-1/\gamma}\right)(1 + O(1)) + \text{erfc}\left(\frac{\nu}{\hat{\Delta}}\right), & 1 \ll \nu^2 \ll n, \\
1, & \nu \ll 1.
\end{cases}
\tag{82}
$$

For $\lambda \gg \sqrt{n/d}$, this term is subleading and $\nu \asymp \lambda\sqrt{d/n}$. One can observe that eq. (49) does not have a positive solution in this regime if $\nu \ll 1$, therefore we exclude this case. Hence, if $\lambda \ll \sqrt{n/d}$,

$$
\text{erfc}\left(\frac{\nu}{\hat{\Delta}}\right) \approx \frac{n}{d} \implies \sqrt{\frac{\hat{\Delta}}{\pi}}\nu^{-1}e^{-\nu^2/\hat{\Delta}} \approx n/d
\tag{83}
$$

$$
\implies \frac{\nu^2}{\hat{\Delta}} \approx \log\frac{d}{n},
\tag{84}
$$

where we used

$$
xe^{x^2} = a \implies 2x^2 e^{2x^2} = 2a^2 \implies x^2 = \frac{1}{2}W_0(2a^2) \overset{a\gg 1}{\approx} \frac{1}{2}\log(2a^2) \approx \log a,
\tag{85}
$$

with $W_0$ denoting the principal branch of the Lambert W function. Note that, in the underparametrized regime $n \ll d$, even a small regularization strength $\lambda$ leads to large effective regularization $\nu$ of order $\sqrt{\log d}$.

We can conclude that, for $n \ll d$

$$
R = \begin{cases}
\Theta(1), & \lambda \gg n/\sqrt{d}, \\
\Theta\left(\frac{n^2}{\lambda^2 d}\right)^{-1+1/(2\gamma)}, & \max\left(nd^{-\gamma-1/2}, \sqrt{n/d}\right) \ll \lambda \ll n/\sqrt{d} \\
\Theta\left(\left(\frac{n}{\log(d/n)}\right)^{-1+1/(2\gamma)} + \frac{\Delta}{\log(d/n)}\right), & \lambda \ll \sqrt{n/d}
\end{cases}
\tag{86}
$$

**Noiseless setting** If $\Delta = 0$, then $\hat{\Delta} = \mathsf{R}$. Let $n \gg d$. Similarly to the noisy setting, it is straightforward to show that $\nu \asymp \lambda\sqrt{d/n}$ in eq. (49), implying $\nu \approx \lambda\sqrt{d/n}$. Therefore, for $n \gg d$

$$\mathsf{R} = \begin{cases} \Theta(1), & \lambda^2 \gg \frac{n^2}{d} \\ \Theta\left(\left(\frac{n^2}{\lambda^2 d}\right)^{-1+1/(2\gamma)}\right), & \frac{n^2}{d^{1+2\gamma}} \ll \lambda^2 \ll \frac{n^2}{d} \\ \Theta\left(\frac{\lambda^2 d^2}{n^2}\right), & \lambda^2 \ll \frac{n^2}{d^{1+2\gamma}}. \end{cases} \tag{87}$$

Let instead $n \ll d$. In this regime, the Bayes optimal generalization error $\mathsf{R} \asymp n^{-2\gamma+1} \gg nd^{-2\gamma}$, which implies $\hat{\Delta} = \mathsf{R} \gg nd^{-2\gamma}$. Similarly to the noisy setting, the second term on the left-hand side of eq. (49) is

$$\frac{1}{d}\sum_{i=1}^{d}\operatorname{erfc}\left(\frac{\nu}{\sqrt{x^{-2\gamma}+R}}\right) \approx \begin{cases} \sqrt{\frac{n}{\pi}}\nu^{-1}\exp\left(-\nu^2/n\right), & \nu^2 \gg n, \\ 2\frac{n^{1/(2\gamma)}}{d}\left(\nu^{2-1/\gamma}\right)(1+\Theta(1)) + \operatorname{erfc}\left(\frac{\nu}{R}\right), & R \ll \nu^2 \ll n, \\ 1, & \nu^2 \ll R. \end{cases} \tag{88}$$

If $\nu^2 \gg n$, this term is negligible and $\nu \asymp \lambda\sqrt{d/n}$. For $\mathsf{R} \ll \nu^2 \ll n$, if this term is negligible with respect to $n/d$, and also in this case $\nu \asymp \lambda\sqrt{d/n}$ with $\mathsf{R} = \Theta(n^2/(\lambda d))^{-1+1/(2\gamma)}$ as in the $n \gg d$ regime. The condition $\nu^2 \gg \mathsf{R}$ becomes

$$\frac{\lambda^2 d}{n} \gg \frac{\lambda^{2-1/\gamma}d^{1-1/(2\gamma)}}{n^{2-1/\gamma}} \implies \lambda \gg \frac{1}{n^{\gamma-1}d^{1/2}}. \tag{89}$$

For consistency, we verify that the second term on the lhs of (49) is negligible with respect to $n/d$:

$$\frac{n^{1/(2\gamma)}}{d}\nu^{2-1/\gamma} \ll \frac{n}{d} \implies \nu \ll n. \tag{90}$$

Finally, if $\lambda \gg n^{-\gamma+1}d^{-1/2}$, repeating the same computations of the noisy setting, we find $\nu^2 \approx \mathsf{R}\log(d/n)$ and

$$\mathsf{R} \approx \left(\frac{n}{\mathsf{R}\log(d/n)}\right)^{-1+1/(2\gamma)} + \frac{\mathsf{R}}{\log(d/n)} \implies \left(1 - \frac{1}{\log(d/n)}\right)\mathsf{R}^{1/(2\gamma)} \approx \left(\frac{n}{\log(d/n)}\right)^{-1+1/(2\gamma)} \tag{91}$$

$$\implies \mathsf{R} \approx n^{-2\gamma+1}(\log(d/n))^{1-1/(2\gamma)}\left(1 - \frac{1}{\log(d/n)}\right)^{-2\gamma} \tag{92}$$

$$\implies \mathsf{R} = \tilde{\Theta}\left(n^{-2\gamma+1}\right). \tag{93}$$

In conclusion, for $n \ll d$,

$$\mathsf{R} = \begin{cases} \Theta(1), & \lambda^2 \gg \frac{n^2}{d} \\ \Theta\left(\left(\frac{n^2}{\lambda^2 d}\right)^{-1+1/(2\gamma)}\right), & n^{-2\gamma+2}d^{-1} \ll \lambda^2 \ll \frac{n^2}{d} \\ \tilde{\Theta}\left(n^{-2\gamma+1}\right), & \lambda^2 \ll n^{-2\gamma+2}d^{-1}. \end{cases} \tag{94}$$

**Interpolation peak** Around $n \sim d$ the excess risk exhibits an interpolation peak that diverges as $\lambda \to 0^+$. In this section we show that in this regime $R \asymp \lambda^{-2/3}$.
The self-consistent equation (49) becomes

$$\frac{\lambda}{\nu} \approx \frac{\sqrt{2}}{d}\sum_{i=1}^{d}\operatorname{erf}\left(\frac{\nu}{\sqrt{ni^{-2\gamma}+\hat{\Delta}}}\right). \tag{95}$$

As we have done in the previous paragraph, it is easy to verify that a strong effective regularization $\nu^2 \gg \hat{\Delta}$ results in the right-hand side being $\Theta(1)$, and to the inconsistent solution $\nu \sim \lambda$. Hence, taking $\nu^2 \ll \hat{\Delta}$ (weak thresholding regime),

$$\frac{\lambda}{\nu} \approx \frac{2\sqrt{2}\nu}{\sqrt{\pi}\hat{\Delta}} \implies \nu^2 \approx \frac{\sqrt{\pi}}{2\sqrt{2}}\frac{\lambda}{\hat{\Delta}} \tag{96}$$

$\nu^2 \asymp \lambda\sqrt{\widehat{\Delta}}$ and the excess risk becomes, asymptotically,

$$R \approx 2\nu^2 + R + \Delta - \frac{4}{\sqrt{\pi}}\nu\sqrt{R+\Delta} \tag{97}$$

$$\implies \nu\sqrt{R+\Delta} \approx \frac{\sqrt{\pi}}{4}\Delta \tag{98}$$

$$\implies \lambda^2(R+\Delta)^3 \approx \frac{\pi^2}{256}\Delta^4 \tag{99}$$

$$\implies R \asymp \frac{\Delta^{4/3}}{\lambda^{2/3}}. \tag{100}$$

### C.2.1 ADDITIONAL NUMERICAL SIMULATIONS

In this section, we include additional numerical experiments, visualizing the Results in Sections 2.1 (Fig. 8) and 2.2 (Figs. 6,7). The remarkable correspondence between simulations and all results derived from state evolution equations further supports our Conjecture 2.

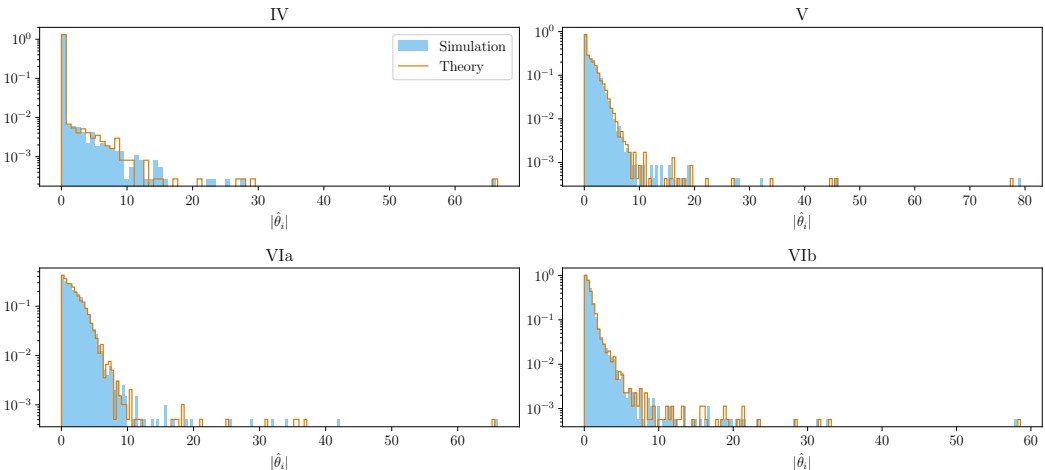

Figure 6: Comparison between spectra from simulations and theory across different training phases. Blue: LASSO estimator's components (in absolute value) histograms after training. Red: theoretical prediction eq. (51). All panels use $d=5000$ and $\lambda=d^{-1/2}$. The sample size is $n=200$ for Phase IV, $n=4000$ for Phase V, $n=6000$ for Phase VIa and $n=4\cdot10^4$ for Phase IVb. We discuss the phenomenology in Section 2.3.

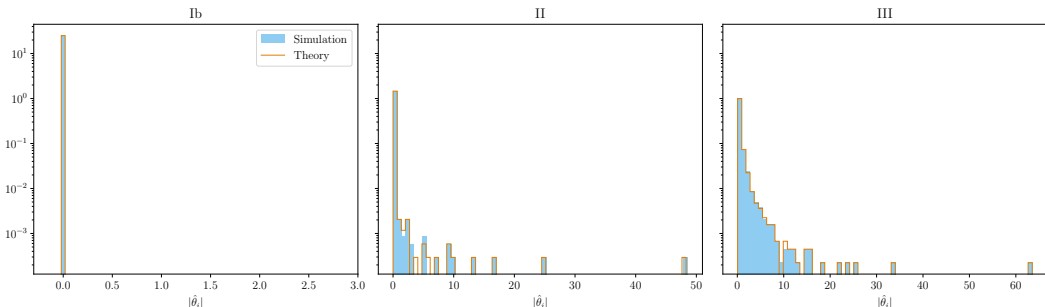

Figure 7: Comparison between spectra from simulations and theory across different training phases. Blue: LASSO estimator's components (in absolute value) histograms after training. Red: theoretical prediction eq. (51). All panels use $d=5000$; Phase Ib: $n=200$, $\lambda=2n/\sqrt{d}$; Phase II: $n=4000$, $\lambda=10$; for Phase III: $n=4\cdot10^4$. $\lambda=3$. We discuss the phenomenology in Section 2.3.

Figure 8 (left) shows the transitions between phases along the vertical lines of Fig. 1. For $n = 35$ the excess risk moves from Phase IV, which is independent of regularization, into Phase II and soon ($\lambda \approx 35/\sqrt{200}$) enters the plateau region of Phase Ib. For $n = 300$ and $n = 500$, the excess risk starts in the fast–decay region IVa, reaches its minimum at $\lambda \approx \sqrt{n/d}$ (when the soft-thresholding cutoff reaches the edge of the noise bulk, see Section 2.2), then crosses Phase II and enters the plateau Phase Ib. Finally, for $n = 3000$, the excess risk begins in Phase IVb, the fastest decay regime, since the noise bulk is negligible for $n \gg d^{2\gamma}$, then grows as it crosses Phase III and Phase II before reaching the plateau in Phase Ib. Notably, for $n = d = 200$, we observe the interpolation phenomenon when $\lambda < \sqrt{n/d} = 1$, with a peak that grows as the predicted $\lambda^{-2/3}$ in the limit $\lambda \to 0^+$.

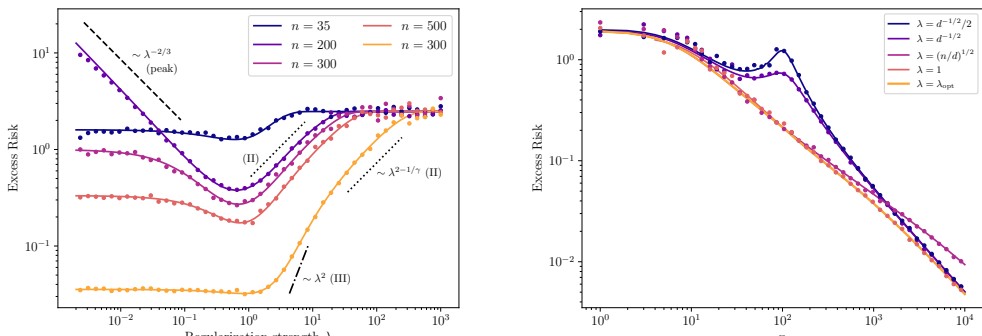

Figure 8: (**Left**) Excess risk of the LASSO estimator, as a function of the regularization strength $\lambda$, with $d = 200$, $\Delta = 0.5$, $\gamma = 0.75$. Dots represent numerical experiments, while lines the solution $\mathsf{R}_{n,d}$ of state evolution equations 48. The curves correspond to the crossovers between rates observed in Fig. 1. (**Right**) Excess risk of the LASSO estimator, as a function of the sample size $n$, with $d = 100$, $\Delta = 0.5$, $\gamma = 1$. The regularization $\lambda_{\mathrm{opt}}$ has been chosen as the minimizer of the theoretical excess risk $\mathsf{R}_{n,d}$ and its value is in accordance with Corollary 1. Dots represent numerical experiments, while lines the solution of state evolution equations 48.

## C.3 BAYES-OPTIMAL EXCESS RISK

The Bayesian predictor $\hat{y}_{\mathrm{BO}}$ is given by the expected value of the target function with respect to the posterior distribution $\mathsf{P}(\boldsymbol{\theta}|\mathcal{D})$. Applying Bayes' theorem, the posterior distribution in this setting reads.

$$\mathsf{P}(\boldsymbol{\theta}|\mathcal{D}) = \frac{1}{Z(\mathcal{D})} \prod_{i=1}^{d} \mathcal{N}(\theta_i; 0, \Lambda_i) \prod_{\mu=1}^{n} \mathcal{N}\left(y_\mu; \frac{1}{\sqrt{d}} \sum_{j=1}^{d} X_{\mu j} \theta_j, \Delta\right) \tag{101}$$

$$= \mathcal{N}(\boldsymbol{\theta}; \hat{\boldsymbol{\theta}}, \boldsymbol{V}), \tag{102}$$

where, recalling the notation $\boldsymbol{X} = (\boldsymbol{x}_1, \ldots, \boldsymbol{x}_n)^T \in \mathbb{R}^{n \times d}$ for the covariate matrix, $\boldsymbol{y} = (y_1, \ldots, y_n)^T$ for the label vector and $\boldsymbol{\Lambda} = \mathrm{diag}(\Lambda_1, ..., \lambda_d)$ for the weights' covariance,

$$\hat{\boldsymbol{\theta}} := \frac{1}{\sqrt{d}\Delta} \boldsymbol{V} \boldsymbol{X}^T \boldsymbol{y}, \qquad \boldsymbol{V} = \left(\boldsymbol{\Lambda}^{-1} + \frac{1}{d\Delta} \boldsymbol{X}^T \boldsymbol{X}\right)^{-1}. \tag{103}$$

Therefore, the Bayesian predictor is $\hat{y}^{\mathrm{BO}}(\boldsymbol{x}) = \hat{\boldsymbol{\theta}}^T \boldsymbol{x}$ and its excess risk is given by

$$R = \mathbb{E}[(\boldsymbol{x}^T \boldsymbol{\theta}^\star - \hat{y}^{\mathrm{BO}}(\boldsymbol{x}))^2] \tag{104}$$

$$= \mathbb{E}\|\boldsymbol{\theta} - \mathbb{E}_{\boldsymbol{\theta}|\mathcal{D}}[\boldsymbol{\theta}]\|^2 \tag{105}$$

$$= \mathrm{Tr}\, \boldsymbol{V}. \tag{106}$$

Leveraging a classical result from random matrix theory Silverstein & Bai (1995), we have that, in the high-dimensional limit $n/d \to \infty$, with fixed ratio $n/d$, the excess risk $R$ concentrates around

$R^{BO}$, solution of the fixed-point equations

$$R^{BO} = \frac{1}{d}\sum_{i=1}^{d}\frac{1}{\Lambda_i^{-1} + d^{-1}\hat{q}}, \qquad \hat{q} = \frac{n}{\Delta + R^{BO}} \tag{107}$$

$$= \sum_{i=1}^{d}\frac{1}{i^{2\gamma} + \hat{q}}, \tag{108}$$

where eq. (108) is the excess risk for the specific choice of covariance $\Lambda_i = di^{-2\gamma}$.
The same equations can be derived from the state evolution of Bayes-GAMP, *i.e.* the GAMP algorithm tailored to compute marginals of the posterior distribution and the Bayes-optimal predictor. The interested reader can find a more detailed discussion in Appendix D of Rangan (2011).

## C.4 BAYES-OPTIMAL SCALING LAWS

From eq. (107)

$$R^{BO} = \frac{1}{\hat{q}}\sum_{i=1}^{d}\frac{1}{1 + (\hat{q}^{-1/(2\gamma)}i)^{2\gamma}}, \qquad \hat{q} = \frac{n}{\Delta + R^{BO}} \tag{109}$$

Our goal is to derive the leading order of $R^{BO}$ in the asymptotic regime $n, d \gg 1$. The crossover scale at which the leading behavior of the sum's argument changes is given by $\hat{q}^{-1/(2\gamma)}i_{\hat{q}} \approx 1 \implies i_{\hat{q}} = \lfloor \hat{q}^{1/(2\gamma)} \rfloor$. If $i_{\hat{q}} \ll d$, we can split the sum at this relevant scale and retain the leading term for each part[*]

$$R^{BO} = \frac{1}{\hat{q}}\left(\sum_{i=1}^{\lfloor \hat{q}^{1/(2\gamma)} \rfloor}\frac{1}{1 + (\hat{q}^{-1/(2\gamma)}i)^{2\gamma}} + \sum_{\lfloor \hat{q}^{1/(2\gamma)} \rfloor + 1}^{d}\frac{1}{1 + (\hat{q}^{-1/(2\gamma)}i)^{2\gamma}}\right) \tag{110}$$

$$\approx \frac{1}{\hat{q}}\left(\sum_{i=1}^{\lfloor \hat{q}^{1/(2\gamma)} \rfloor}1 + \sum_{\lfloor \hat{q}^{1/(2\gamma)} \rfloor + 1}^{d}\hat{q}i^{-2\gamma}\right) \tag{111}$$

$$\approx \hat{q}^{-1+1/(2\gamma)} + \frac{1}{2\gamma - 1}\hat{q}^{-1+1/(2\gamma)} \tag{112}$$

$$= \frac{2\gamma}{2\gamma - 1}\hat{q}^{-1+1/(2\gamma)}, \tag{113}$$

where we approximate

$$\int_{i_{\hat{q}}+1}^{(d+1)}x^{-2\gamma}dx \le \sum_{i_{\hat{q}}+1}^{d}i^{-2\gamma} \le (i_{\hat{q}}+1)^{-2\gamma} + \int_{i_{\hat{q}}+1}^{d\hat{q}^{-1/(2\gamma)}}x^{-2\gamma}dx \tag{114}$$

$$\implies \left|\sum_{i_{\hat{q}}+1}^{d}i^{-2\gamma} - \frac{\hat{q}^{-1+1/(2\gamma)}}{2\gamma - 1}\right| = o_{\hat{q}}\left(q^{-1+1/(2\gamma)}\right). \tag{115}$$

If instead $i_{\hat{q}} \gg d$

$$R^{BO} \approx \frac{1}{\hat{q}}\sum_{i=1}^{d}1 = \frac{d}{\hat{q}} \tag{116}$$

**Noisy setting.** Since $\Delta > 0$, assuming $R^{BO} = O(1)$, the parameter $\hat{q} \asymp n$ and the Bayes-optimal generalization error

$$R^{BO} = \begin{cases} \Theta(n^{-1+1/(2\gamma)}), & n \ll d^{2\gamma} \\ \Theta(d/n), & n \gg d^{2\gamma}. \end{cases} \tag{117}$$

---

[*]We stress that, throughout the manuscript, the notation $\approx$ denotes equality up to terms that are asymptotically negligible.

**Noiseless setting** When $\Delta = 0$, the parameter $\hat{q} = n/R^{\text{BO}}$ and the Bayes-optimal generalization error

$$R^{\text{BO}} = \begin{cases} \Theta(n^{-2\gamma+1}), & n \ll d \\ 0, & n \gg d. \end{cases} \tag{118}$$

# D DERIVATION DETAILS - QUADRATIC NEURAL NETWORKS

## D.1 BBP SIMPLIFICATION

To derive our results, we make the following assumption, which for the moment we do not control rigorously. We assume that for the sake of simpliying the equations that $\boldsymbol{S}^\star$ has sub-extensive rank, i.e. it has eigenvalues $\{\sqrt{d}i^{-\gamma}\}_{i=1}^{cd}$ for $c \ll 1$, and zero otherwise. This technical assumption allows for a great simplification of the density $\mu_\delta$ (the spectrum of $\boldsymbol{S}^\star + \delta\boldsymbol{Z}$, where $\boldsymbol{Z} \sim \text{GOE}(d)$), which can be computed with BBP-like techniques as (Huang, 2018)

$$\mu_\delta(x) = \left(1 - \frac{K}{d}\right)(\mu_{\text{sc}} + o(1))(x/\delta)/\delta + \frac{1}{d}\sum_{i=1}^{K}\delta(x - f_\delta(\sqrt{d}i^{-\gamma})), \tag{119}$$

where $K \ll d$ is the largest integer such that $\sqrt{d}K^{-\gamma} \geq \delta$. $f_\delta(x) = x + \delta^2/x$ is the BBP-like shifting function. We then send $c \to 1$ *a posteriori*, after finding that the error does not depend explicitly on $c$. For the rest of the section, let us define $\tilde{\alpha} := n/d^2$.

## D.2 BAYESIAN ESTIMATOR

In this section we solve eq. (19) for $\Delta > 0$.

**Phase I: Large-samples** Phase I is $n \gg d^{2\gamma+1}$. The first equation of eq. (19) gives $\hat{q} = \Theta(n/d^2)$, and thus $d^{\frac{1}{2}-\gamma} \gg \frac{1}{\sqrt{\hat{q}}}$, so all the spikes are outside the bulk. We then have

$$\frac{4\pi^2}{3\hat{q}}\int \mu_{1/\sqrt{\hat{q}}}(x)^3 dx \approx \left(1 - \frac{cd}{d}\right)\frac{4\pi^2}{3}\int \mu_{\text{sc}}(x)^3 dx = 1 - c \tag{120}$$

by eq. (119), and thus eq. (19) gives

$$2\tilde{\alpha} - c = \frac{2\tilde{\alpha}\Delta}{\Delta + 2(Q^\star - q)}, \tag{121}$$

which gives $\text{R}^{\text{BO}} := Q^\star - q = \frac{\Delta c d^2}{4n} = \Theta(d^2/n)$.

**Phase II: Under-sampling** Phase II is $d \ll n \ll d^{2\gamma+1}$. We rewrite the integral in eq. (19) as

$$\frac{4\pi^2}{3\hat{q}}\int \mu_{1/\sqrt{\hat{q}}}(x)^3 dx = \frac{4\pi^2}{3}\int \nu(x)^3 dx = 4\pi^2 \int dx\nu(x)(H[\nu](x))^2, \tag{122}$$

where $\nu$ denotes the spectrum density of $Z + \sqrt{\hat{q}}\Theta^*$. The first equality is by change of variables and the second quality is from (Maillard et al., 2022, Lemma C.1). $H[\nu]$ denotes the Hilbert transform of $\nu$.

We further denote $\delta := \sqrt{\hat{q}d}$. The first equation of eq. (19) suggests that $\hat{q} = \Theta(n/d^2)$, and thus $\delta = \Theta(\sqrt{n/d})$. For $d \ll n \ll d^{2\gamma+1}$ we have $d^{-\gamma+\frac{1}{2}} \ll \delta \ll \sqrt{d}$, so according to eq. (119) $\nu$ is composed of a semicircle part (denoted as $\nu_0 := (1 - K/d)(\mu_{\text{sc}} + \tilde{\nu})$, where $\tilde{\nu}$ is calculated in Appendix D.5) and a discrete part ($\{x_i := f_1(\delta i^{-\gamma})\}_{i=1}^{K}$, where $K := \delta^{1/\gamma}$). Thus we have

$$\int dx\nu(x)(H[\nu](x))^2 = \int dx\nu_0(x)H[\nu_0](x)^2 + \frac{2}{d}\sum_{i=1}^{K}\int dx\nu_0(x)H[\nu_0](x)\frac{1}{\pi(x-x_i)}$$

$$+ \frac{1}{d^2}\sum_{i,j=1}^{K}\int dx\nu_0(x)\frac{1}{\pi^2(x-x_i)(x-x_j)} + \sum_{j=1}^{K}\frac{1}{d}\left(H[\nu_0](x_j) + \sum_{\substack{i=1 \\ i\neq j}}^{K}\frac{1}{\pi d(x_j-x_i)}\right)^2. \tag{123}$$

Denote the right side terms as $I_1$ to $I_4$. For the first term we have

$$I_1 \approx \frac{1}{3} \int \mathrm{d}x \mu_{\mathrm{sc}}^3(x) \left(1 - \Theta\left(\frac{1}{d} \sum_{i=K}^{cd} \delta^2 i^{-2\gamma}\right)\right)(1 - 3K/d) \approx \frac{1}{4\pi^2}\left(1 - \Theta(\delta^{1/\gamma} d^{-1})\right), \quad (124)$$

where we use eq. (196). Then we can estimate the leading orders of the other three terms. For the second term, we have

$$I_2 \approx -\frac{2}{d} \int \mathrm{d}x \mu_{\mathrm{sc}}(x) \frac{x}{2\pi} \sum_{i=1}^{K} \frac{1}{x_i} = o\left(d^{-1} \sum_{i=1}^{K} \frac{1}{x_i}\right) = o(\delta^{1/\gamma} d^{-1}), \quad (125)$$

where we use $H[\mu_{\mathrm{sc}}] = \frac{x}{2\pi}$ for $x \in [-2,2]$ and the fact that $x\mu_{\mathrm{sc}}$ is an odd function. For the third term we have

$$I_3 = \Theta\left(d^{-2} \sum_{i,j=1}^{K} \frac{1}{x_i x_j}\right) = \Theta\left(\left(d^{-1} \sum_{i=1}^{K} \frac{1}{x_i}\right)^2\right) = \Theta(\delta^{2/\gamma} d^{-2}). \quad (126)$$

$I_4$ is composed of three terms. The first term is

$$\frac{1}{d} \sum_{j=1}^{K} (H[\nu_0](x_j))^2 \approx \frac{1}{d} \sum_{j=1}^{K} x_j^{-2} \approx \frac{1}{\delta^2 d} K^{2\gamma+1} \int_0^1 \frac{x^{2\gamma}}{(1+x^{2\gamma})^2} \mathrm{d}x = \Theta(\delta^{1/\gamma} d^{-1}), \quad (127)$$

where we use the fact that $H[\mu_{\mathrm{sc}}](x) \approx \frac{1}{x}$ for $x \gg 1$. The second term of $I_4$ is

$$\sum_{\substack{i,j=1 \\ i \neq j}}^{K} \frac{2}{d^2} H[\nu_0](x_j) \frac{1}{\pi(x_j - x_i)} \approx \sum_{\substack{i,j=1 \\ i \neq j}}^{K} \frac{2}{\pi d^2} \frac{1}{x_j(x_j - x_i)}$$

$$= \sum_{\substack{i,j=1 \\ i \neq j}}^{K} \frac{1}{\pi d^2}\left(\frac{1}{x_j(x_j - x_i)} + \frac{1}{x_i(x_i - x_j)}\right) \quad (128)$$

$$= \sum_{\substack{i,j=1 \\ i \neq j}}^{K} \frac{1}{\pi d^2 x_i x_j} = \frac{1}{\pi d^2}\left(\sum_{i=1}^{K} \frac{1}{x_i}\right)^2 = \Theta(\delta^{2/\gamma} d^{-2}).$$

The last term of $I_4$ can be written as

$$\frac{1}{d^3} \sum_{\substack{i,j,k=1 \\ i \neq j,k}}^{K} \frac{1}{(x_j - x_i)(x_k - x_i)} \approx \frac{1}{\delta^2 d^3} \sum_{\substack{i,j,k=1 \\ i \neq j,k}}^{K} \frac{1}{(j^{-\gamma} - i^{-\gamma})(k^{-\gamma} - i^{-\gamma})} \approx \frac{1}{\delta^2 d^3} \sum_{i=1}^{K} \sum_{m,n} \frac{i^{2\gamma+2}}{mn\gamma^2} \tag{129}$$

where we assuming $m, n \ll i$ to obtain the leading term. For a fixed $i$, we sum over $m, n = -(i-1), \cdots, -1, 1, \cdots, K - i$, which gives

$$\sum_{i=1}^{K} \sum_{p,q} \frac{i^{2\gamma+2}}{pq} = \sum_{i=1}^{K} i^{2\gamma+2}(H_{K-i} - H_{i-1}^2) \approx K^{2\gamma+3} \int_0^1 x^{2\gamma}\left(\log \frac{1-x}{x}\right)^2 dx, \quad (130)$$

where we denote $H_i = \sum_{p=1}^{i} \frac{1}{p} = \log i + \Theta(1)$. Thus we have

$$\frac{1}{d^3} \sum_{\substack{i,j,k=1 \\ i \neq j,k}}^{K} \frac{1}{(x_j - x_i)(x_k - x_i)} = \Theta(\delta^{-2} d^{-3} K^{2\gamma+3}) = \Theta(\delta^{3/\gamma} d^{-3}). \quad (131)$$

For $d \ll n \ll d^{2\gamma+1}$ and $\gamma > \frac{1}{2}$ we have $\delta^{1/\gamma} d^{-1} \ll 1$, and thus

$$\frac{4\pi^2}{3} \int \nu(y)^3 dy = 1 + \Theta(d^{-1}\delta^{1/\gamma}). \quad (132)$$

Taking it back to (19), we have

$$\frac{2\tilde{\alpha}\Delta}{\Delta + 2(Q^\star - q)} - 2\tilde{\alpha} = \Theta\left(d^{-1}\left(\frac{4n}{d\Delta}\right)^{\frac{1}{2\gamma}}\right), \tag{133}$$

which gives

$$R^{BO} := Q^\star - q = \Theta\left(\left(\frac{d\Delta}{4n}\right)^{1-\frac{1}{2\gamma}}\right). \tag{134}$$

**Phase III: Not enough data**  Phase III is $n \ll d$, and thus $\delta \ll 1$. By eq. (119) and Appendix D.5, there are no outliers and the first-order correction reads

$$\frac{4\pi^2}{3}\int \nu(x)^3 dx = 1 - \frac{1}{d}\sum_{i=1}^{cd}(\delta i^{-\gamma})^2 = 1 - \zeta(2\gamma)\hat{q}. \tag{135}$$

Taking it back to eq. (19), we have

$$\frac{2\tilde{\alpha}\Delta}{\Delta + 2(Q^\star - q)} - 2\tilde{\alpha} = -\frac{4\tilde{\alpha}Q^\star}{\Delta + 2(Q^\star - q)}, \tag{136}$$

and thus $R^{BO} := Q^\star - q = Q^\star$, where we use $Q^\star \approx \zeta(2\gamma)$.

### D.3  ERM

In this section we solve eq. (20) for $\Delta > 0$.

**Phase I: Trivial phase**  The first case of phase I is $\delta > \sqrt{d}$ and $0 < 2 - \frac{\lambda\epsilon}{\delta} \ll 1$. In this case the spikes are covered by the bulk and the cutoff is close to the boundary of the bulk. Thus we have

$$J(\delta, \lambda\epsilon) \approx \int_{\lambda\epsilon}^{2\delta} \mu_{sc}(x/\delta)/\delta(x - \lambda\epsilon)^2 dx \approx \delta^2\frac{16t^{7/2}}{105\pi}, \tag{137}$$

where $t := 2 - \frac{\lambda\epsilon}{\delta}$. Then eq. (20) reduces to

$$\begin{cases} 4\tilde{\alpha}\delta^2 - \frac{\delta^2}{\epsilon} = \delta^2\frac{16t^{5/2}}{15\pi} \\ Q^\star + \frac{\Delta}{2} + 2\tilde{\alpha}\delta^2 - \frac{\delta^2}{\epsilon} = \delta^2\frac{16t^{5/2}}{15\pi}, \end{cases} \tag{138}$$

where we use $t \ll 1$ and keep only the leading term. The difference between the two equations of (138) gives

$$R := 2\tilde{\alpha}\delta^2 - \frac{\Delta}{2} = Q^\star \tag{139}$$

and $\lambda\epsilon \approx 2\delta = \sqrt{\frac{1}{2\tilde{\alpha}}(Q^\star + \Delta/2)}$. Then the condition $\delta > \sqrt{d}$ gives

$$n < \frac{1}{16}(2Q^\star + \Delta)d. \tag{140}$$

The condition $t > 0$ gives $\frac{\lambda}{2} \approx \frac{\delta}{\epsilon} < 4\tilde{\alpha}\delta$, and thus

$$\lambda < 8\sqrt{\frac{2Q^\star + \Delta}{4}}\sqrt{\frac{n}{d^2}}. \tag{141}$$

The condition $t \ll 1$ is naturally satisfied because $\frac{16t^{5/2}}{15\pi} \le \frac{4\tilde{\alpha}}{\delta} \ll 1$ for $n < d$ and $d \gg 1$.

The second case of phase I is $\lambda\epsilon > 2\max(\delta, \sqrt{d})$. In this case both the spikes and the bulk are below the cutoff, and thus $J(\delta, \lambda\epsilon) = 0$. Then eq. (20) reduces to

$$\begin{cases} 4\tilde{\alpha}\delta - \frac{\delta}{\epsilon} = 0 \\ Q^\star + \frac{\Delta}{2} + 2\tilde{\alpha}\delta^2 - \frac{\delta^2}{\epsilon} = 0. \end{cases} \tag{142}$$

The first equation of (142) gives $\epsilon = \frac{1}{4\tilde{\alpha}}$. The difference of the two equations in (142) gives

$$\mathsf{R} := 2\tilde{\alpha}\delta^2 - \frac{\Delta}{2} = Q^\star, \tag{143}$$

and consequently $\delta = \sqrt{\frac{2Q^\star + \Delta}{4\tilde{\alpha}}}$. The condition $\lambda\epsilon > 2\max(\delta, \sqrt{d})$ reduces to

$$\lambda > \max\left(8\sqrt{\frac{2Q^\star + \Delta}{4}}\sqrt{\frac{n}{d^2}}, \frac{4n}{d^{3/2}}\right). \tag{144}$$

To conclude, Phase I is

$$\mathsf{R} = Q^\star, \quad \text{if} \quad n < \frac{1}{16}(2Q^\star + \Delta)d \quad \text{or} \quad \lambda > \frac{4n}{d^{3/2}}. \tag{145}$$

**Phase II: Over-regularization phase** The second phase is $\max(\delta, d^{-\gamma+1/2}) \ll \lambda\epsilon \ll \sqrt{d}$. In this case we only need to consider the spikes outside the cutoff, so we have

$$J(\delta, \lambda\epsilon) = \frac{1}{d}\sum_{i=1}^{K}(\sqrt{d}i^{-\gamma} + \frac{\delta^2}{\sqrt{d}i^{-\gamma}} - \lambda\epsilon)^2, \tag{146}$$

where $K$ is given by $\sqrt{d}K^{-\gamma} + \frac{\delta^2}{\sqrt{d}K^{-\gamma}} - \lambda\epsilon = 0$. Thus we have $K \approx (\sqrt{d}/\lambda\epsilon)^{1/\gamma}$ satisfying $1 \ll K < d$. By keeping only the leading terms, we have

$$J(\delta, \lambda\epsilon) \approx Q^\star + \left(\frac{\gamma+1}{\gamma-1} - \frac{1}{2\gamma-1}\right)\left(\frac{\lambda\epsilon}{\sqrt{d}}\right)^{\frac{2\gamma-1}{\gamma}} - \frac{2\lambda\epsilon}{\sqrt{d}}\zeta(\gamma)\mathbf{1}_{\gamma>1}, \tag{147}$$

where we use

$$\sum_{i=1}^{K}i^{-\gamma} \approx \frac{K^{1-\gamma}}{1-\gamma} + \zeta(\gamma)\mathbf{1}_{\gamma>1} \tag{148}$$

and thus $Q^\star := \sum_{i=1}^{cd}i^{-2\gamma} \approx \zeta(2\gamma)$. Taking it into eq. (20), we have

$$\begin{cases} 4\tilde{\alpha}\delta - \frac{\delta}{\epsilon} = 0 \\ Q^\star + \frac{\Delta}{2} + 2\tilde{\alpha}\delta^2 - \frac{\delta^2}{\epsilon} = Q^\star - \frac{2\gamma}{2\gamma-1}\left(\frac{\lambda\epsilon}{\sqrt{d}}\right)^{\frac{2\gamma-1}{\gamma}}, \end{cases} \tag{149}$$

which gives $\epsilon = \frac{1}{4\tilde{\alpha}}$ and

$$\mathsf{R} := 2\tilde{\alpha}\delta^2 - \frac{\Delta}{2} = \frac{2\gamma}{2\gamma-1}\left(\frac{\lambda\epsilon}{\sqrt{d}}\right)^{\frac{2\gamma-1}{\gamma}} = \frac{2\gamma}{2\gamma-1}\left(\lambda\frac{d^{3/2}}{4n}\right)^{\frac{2\gamma-1}{\gamma}}. \tag{150}$$

The condition $\max(\delta, d^{-\gamma+1/2}) \ll \lambda\epsilon \ll \sqrt{d}$ gives

$$\max\left(\sqrt{\frac{n}{d^2}}, \frac{n}{d^{\gamma+\frac{3}{2}}}\right) \ll \lambda \ll \frac{d^{3/2}}{n}. \tag{151}$$

**Phase III: Intermediate over-regularization phase** Phase III is $\delta \ll \lambda\epsilon \ll d^{-\gamma+\frac{1}{2}}$. In this case all the spikes are above the cutoff, and all the bulk is below the cutoff. Thus we have

$$J(\delta, \lambda\epsilon) = \frac{1}{d}\sum_{i=1}^{cd}(\sqrt{d}i^{-\gamma} + \frac{\delta^2}{\sqrt{d}i^{-\gamma}} - \lambda\epsilon)^2 \approx Q^\star - \lambda\epsilon c(d)^{\min(\gamma,1)-1} + \lambda^2\epsilon^2 \tag{152}$$

by eq. (148). Then eq. (20) simplifies to

$$\begin{cases} 4\tilde{\alpha}\delta - \frac{\delta}{\epsilon} = 0 \\ Q^\star + \frac{\Delta}{2} + 2\tilde{\alpha}\delta^2 - \frac{\delta^2}{\epsilon} = Q^\star - \lambda^2\epsilon^2, \end{cases} \tag{153}$$

which gives $\epsilon = \frac{1}{4\tilde{\alpha}}$ and

$$\mathsf{R} := 2\tilde{\alpha}\delta^2 - \frac{\Delta}{2} = \frac{\lambda^2 d^4}{16n^2}. \tag{154}$$

The condition $\delta \ll \lambda\epsilon \ll d^{-\gamma+\frac{1}{2}}$ reduces to

$$\sqrt{\frac{n}{d^2}} \ll \lambda \ll \frac{n}{d^{\gamma+3/2}}, \tag{155}$$

which further requires $n \gg d^{2\gamma+1}$.

**Phase IV and V: Benign and harmful overfitting phase**   Phase IV and V are $d \ll n \ll d^2$ and $0 < 2 - \frac{\lambda\epsilon}{\delta} \ll 1$, $d^{-\gamma+\frac{1}{2}} \ll \delta \ll \sqrt{d}$. In this case the cutoff is close to the boundary of the bulk and a part of the spikes are outside the bulk. Thus we have $J(\delta, \lambda\epsilon) \approx J_1(\delta, \lambda\epsilon) + J_2(\delta, \lambda\epsilon)$, where

$$J_2(\delta, \lambda\epsilon) := \int_{\lambda\epsilon}^{2\delta} \mu_{\text{sc}}(x/\delta)/\delta(x - \lambda\epsilon)^2 dx \approx \delta^2 \frac{16t^{7/2}}{105\pi} + A\delta^2 t^{9/2} \tag{156}$$

up to the sub-leading order, where $A$ is a constant, $t := 2 - \frac{\lambda\epsilon}{\delta}$ and

$$J_1(\delta, \lambda\epsilon) := \frac{1}{d} \sum_{i=1}^{(\delta/\sqrt{d})^{-1/\gamma}} (\sqrt{d}i^{-\gamma} + \frac{\delta^2}{\sqrt{d}i^{-\gamma}} - \lambda\epsilon)^2$$

$$\approx Q^\star + \left(\frac{\delta}{\sqrt{d}}\right)^{2-\frac{1}{\gamma}} \left(-\frac{1}{2\gamma-1} + (\lambda\epsilon/\delta)^2 + 2 - 2\frac{\lambda\epsilon}{\delta}\frac{1}{1-\gamma} - \frac{2}{1+\gamma}\frac{\lambda\epsilon}{\delta} + \frac{1}{1+2\gamma}\right)$$

$$- \mathbf{1}_{\gamma>1}\zeta(\gamma)\frac{2\lambda\epsilon}{\sqrt{d}}, \tag{157}$$

where we use $\delta \gg d^{-\gamma+\frac{1}{2}}$ to obtain the first line and use eq. (148) for the second line. Then eq. (20) simplifies to

$$\begin{cases} 4\tilde{\alpha}\delta^2 - \frac{\delta^2}{\epsilon} = \delta^2 \frac{16t^{5/2}}{15\pi} + \delta^2 t^{7/2}(\frac{32}{105\pi} + 9A) \\ \qquad + \left(\left(-\frac{1}{2\gamma-1} + 4 + 2 - \frac{4}{1-\gamma} - \frac{4}{1+\gamma} + \frac{1}{1+2\gamma}\right)(2 - 1/\gamma) + C'(\gamma)\right)\left(\frac{\delta}{\sqrt{d}}\right)^{2-\frac{1}{\gamma}} \\ Q^\star + \frac{\Delta}{2} + 2\tilde{\alpha}\delta^2 - \frac{\delta^2}{\epsilon} = Q^\star + \delta^2 \frac{16t^{5/2}}{15\pi} + \delta^2 t^{7/2}(\frac{16}{105\pi} + 9A) \\ \qquad + \left(-\frac{1}{2\gamma-1} + 4 + 2 - \frac{4}{1-\gamma} - \frac{4}{1+\gamma} + \frac{1}{1+2\gamma} + C'(\gamma)\right)\left(\frac{\delta}{\sqrt{d}}\right)^{2-\frac{1}{\gamma}}, \end{cases} \tag{158}$$

where we use $t \ll 1$ to drop the smaller terms. We also use the shorthand $C'(\gamma) := -4 + \frac{2}{1-\gamma} + \frac{4}{1+\gamma}$. The second equation of (158) subtracted by the first gives

$$\frac{\Delta}{2} - 2\tilde{\alpha}\delta^2 = \left(\left(6 - \frac{1}{2\gamma-1} - \frac{4}{1+\gamma} + \frac{1}{1+2\gamma}\right)\frac{1-\gamma}{\gamma} - \frac{4}{\gamma}\right)\left(\frac{\delta}{\sqrt{d}}\right)^{2-\frac{1}{\gamma}} - \delta^2 \frac{16t^{7/2}}{105\pi}. \tag{159}$$

Therefore, at the leading order we have $\delta \approx \sqrt{\frac{\Delta}{4\tilde{\alpha}}}$. Further assuming that $\lambda\delta \ll 1$, the first equation of (158) gives

$$\delta^2 \frac{16t^{5/2}}{15\pi} \approx 4\tilde{\alpha}\delta^2 - \frac{\delta^2}{\epsilon} \approx \Delta. \tag{160}$$

Then we have

$$\mathsf{R} := 2\tilde{\alpha}\delta^2 - \frac{\Delta}{2} = \frac{24\gamma^3}{4\gamma^3 + 4\gamma^2 - \gamma - 1}\left(\frac{d\Delta}{4n}\right)^{1-\frac{1}{2\gamma}} + \frac{\Delta}{7}\left(\frac{15\pi}{4}\right)^{2/5}\left(\frac{n}{d^2}\right)^{2/5}. \tag{161}$$

The condition $d \ll n \ll d^2$ and $\lambda\delta \ll 1$, $0 < t \ll 1$, $d^{-\gamma+\frac{1}{2}} \ll \delta \ll \sqrt{d}$ reduces to

$$\lambda \ll \sqrt{\frac{n}{d^2}} \quad \text{and} \quad d \ll n \ll d^2, \tag{162}$$

where we use $\delta^2 t^{5/2} \approx \Delta - \frac{\lambda\delta}{2}$. Note that under this condition $t \ll 1$ is satisfied because $\delta \gg 1$.

**Interpolation peak**   The interpolation peak is at $\tilde{\alpha} = \frac{1}{4}$ and $\lambda \ll 1$. The first case of the interpolation peak is $\max(\lambda\epsilon, d^{-\gamma+1/2}) \ll \delta \ll \sqrt{d}$. Then $J(\delta, \lambda\epsilon) \approx J_1(\delta) + J_2(\delta, \lambda\epsilon)$, where

$$J_1(\delta) := \frac{1}{d} \sum_{i=1}^{cd} \text{ReLU}(\sqrt{d}i^{-\gamma} + \frac{\delta^2}{\sqrt{d}i^{-\gamma}})^2 \approx Q^\star + C(\gamma)\left(\frac{\delta}{\sqrt{d}}\right)^{2-\frac{1}{\gamma}} \tag{163}$$

by eq. (177) and

$$J_2(\delta, \lambda\epsilon) := \delta^2 \int_{\lambda\epsilon}^2 \mu_{\mathrm{sc}}(x)(x - \lambda\epsilon/\delta)^2 \approx \frac{\delta^2}{2} - \frac{8}{3\pi}\lambda\epsilon\delta. \tag{164}$$

Then eq. (20) reduces to

$$\begin{cases} 4\tilde{\alpha}\delta^2 - \frac{\delta^2}{\epsilon} = \delta^2 - \frac{8}{3\pi}\lambda\epsilon\delta + C(\gamma)(2 - 1/\gamma)\left(\frac{\delta}{\sqrt{d}}\right)^{2-\frac{1}{\gamma}} \\ Q^\star + \frac{\Delta}{2} + 2\tilde{\alpha}\delta^2 - \frac{\delta^2}{\epsilon} = Q^\star + \frac{1}{2}\delta^2 + C(\gamma)\left(\frac{\delta}{\sqrt{d}}\right)^{2-\frac{1}{\gamma}}. \end{cases} \tag{165}$$

By using $\tilde{\alpha} = \frac{1}{4}$, we obtain

$$\epsilon = \frac{\Delta}{2}\lambda^{-2/3}\left(\frac{3\pi}{8}\right)^2, \ \delta = \left(\frac{3\pi\Delta^2}{32}\right)^{1/3}\lambda^{-1/3} \tag{166}$$

as the leading order solution, where the $\left(\frac{\delta}{\sqrt{d}}\right)^{2-\frac{1}{\gamma}}$ term is ignored because $\frac{\delta}{\sqrt{d}} \ll 1$. Then we have

$$\mathsf{R} := 2\tilde{\alpha}\delta^2 - \frac{\Delta}{2} \approx 2\left(\frac{3\pi\Delta^2}{32}\right)^{2/3}\lambda^{-2/3}. \tag{167}$$

The condition $\max(\lambda\epsilon, d^{-\gamma+1/2}) \ll \delta \ll \sqrt{d}$ reduces to

$$d^{-3/2} \ll \lambda \ll 1 \tag{168}$$

The second case of the interpolation peak is $\max(\lambda\epsilon, \sqrt{d}) \ll \delta$, which gives $J(\delta, \lambda\epsilon) \approx \frac{\delta^2}{2} - \frac{8}{3\pi}\lambda\epsilon\delta$. Similarly we can obtain the solution

$$\epsilon = \left(Q^\star + \frac{\Delta}{2}\right)\lambda^{-2/3}\left(\frac{3\pi}{8}\right)^2, \ \delta = \left(\frac{3\pi}{8}\right)^{1/3}\left(Q^\star + \frac{\Delta}{2}\right)^{2/3}\lambda^{-1/3} \tag{169}$$

and thus

$$\mathsf{R} := 2\tilde{\alpha}\delta^2 - \frac{\Delta}{2} \approx 2\left(\frac{3\pi}{8}\right)^{2/3}\left(Q^\star + \frac{\Delta}{2}\right)^{4/3}\lambda^{-2/3}. \tag{170}$$

The condition $\max(\lambda\epsilon, \sqrt{d}) \ll \delta$ reduces to

$$\lambda \ll d^{-3/2}. \tag{171}$$

**Phase VI: Large-sample phase** The first case of Phase VI is $n \gg d^2$ and $\lambda\epsilon \ll \delta \ll d^{-\gamma+\frac{1}{2}}$. In this case the cutoff is almost 0 and all spikes are outside the bulk. Then we have

$$J(\delta, \lambda\epsilon) \approx \frac{1}{2}\delta^2 + \frac{1}{d}\sum_{i=1}^{cd}(\sqrt{d}i^{-\gamma} + \frac{\delta^2}{\sqrt{d}i^{-\gamma}}) \approx Q^\star + \frac{1}{2}\delta^2(1 + 4c), \tag{172}$$

where we recall that the rank of $\boldsymbol{S}^\star$ is $cd$. Thus eq. (20) simplifies to

$$\begin{cases} 4\tilde{\alpha}\delta - \frac{\delta}{\epsilon} = \delta(1 + 4c) \\ Q^\star + \frac{\Delta}{2} + 2\tilde{\alpha}\delta^2 - \frac{\delta^2}{\epsilon} = Q^\star + \frac{1}{2}\delta^2(1 + 4c), \end{cases} \tag{173}$$

which has a solution

$$\delta^2 = \frac{\Delta}{4\tilde{\alpha} - 1 - 4c}, \ \epsilon = \frac{1}{4\tilde{\alpha} - 1 - 4c}. \tag{174}$$

Then we have

$$\mathsf{R} := 2\tilde{\alpha}\delta^2 - \frac{\Delta}{2} \approx \frac{\Delta}{8\tilde{\alpha}}(1 + 4c) \tag{175}$$

for $\tilde{\alpha} \gg 1$ and the condition $\lambda\epsilon \ll \delta \ll d^{-\gamma+\frac{1}{2}}$ reduces to

$$\lambda \ll \sqrt{\frac{n}{d^2}} \quad \text{and} \quad n \gg d^{2\gamma+1}. \tag{176}$$

The second case of Phase VI is $d^2 \ll n \ll d^{2\gamma+1}$ and $\max(\lambda\epsilon, d^{-\gamma+\frac{1}{2}}) \ll \delta \ll 1$. In this case the cutoff is almost 0 but only a part of the spikes are outside the bulk. Thus we have $J(\delta, \lambda\epsilon) = J_1(\delta) + \frac{1}{2}\delta^2$, where

$$\begin{aligned}
J_1(\delta) :&= \frac{1}{d}\sum_{i=1}^{cd} \text{ReLU}(\sqrt{d}i^{-\gamma} + \frac{\delta^2}{\sqrt{d}i^{-\gamma}})^2 \\
&\approx \frac{1}{d}\sum_{i=1}^{(\delta/\sqrt{d})^{-1/\gamma}} (\sqrt{d}i^{-\gamma} + \frac{\delta^2}{\sqrt{d}i^{-\gamma}})^2 \\
&= Q^\star + C(\gamma)\left(\frac{\delta}{\sqrt{d}}\right)^{2-\frac{1}{\gamma}},
\end{aligned} \tag{177}$$

where $C(\gamma) := \frac{4-4\gamma^2}{1-4\gamma^2}$ is a constant. In the second line we use $\delta \gg d^{-\gamma+1/2}$. In the third line we use eq. (148) and only keep the leading term. Then eq. (20) reduces to

$$\begin{cases}
4\tilde{\alpha}\delta^2 - \frac{\delta^2}{\epsilon} = \delta^2 + C(\gamma)(2 - 1/\gamma)\left(\frac{\delta}{\sqrt{d}}\right)^{2-\frac{1}{\gamma}} \\
Q^\star + \frac{\Delta}{2} + 2\tilde{\alpha}\delta^2 - \frac{\delta^2}{\epsilon} = Q^\star + \frac{1}{2}\delta^2 + C(\gamma)\left(\frac{\delta}{\sqrt{d}}\right)^{2-\frac{1}{\gamma}}
\end{cases} \tag{178}$$

The second equation subtracted by the first equation gives

$$\frac{\Delta}{2} - 2\tilde{\alpha}\delta^2 = -\frac{1}{2}\delta^2 - C(\gamma)(1 - 1/\gamma)\left(\frac{\delta}{\sqrt{d}}\right)^{2-\frac{1}{\gamma}}. \tag{179}$$

As $n \gg d^2$, at the leading order we have $\delta \approx \sqrt{\frac{\Delta}{4\tilde{\alpha}}}$. Then one can verify that if $n \ll d^{2\gamma+1}$ we have $\left(\frac{\delta}{\sqrt{d}}\right)^{2-\frac{1}{\gamma}} \ll \delta^2$, which suggests that

$$\mathsf{R} := 2\tilde{\alpha}\delta^2 - \frac{\Delta}{2} \approx \frac{1}{2}\delta^2 \approx \frac{\Delta}{8\tilde{\alpha}}. \tag{180}$$

In this case we also have $\epsilon \approx \frac{1}{4\tilde{\alpha}}$, and thus the condition $\max(\lambda\epsilon, d^{-\gamma+\frac{1}{2}}) \ll \delta \ll 1$ reduces to

$$\lambda \ll \sqrt{\frac{n}{d^2}} \quad \text{and} \quad d^2 \ll n \ll d^{2\gamma+1}. \tag{181}$$

To conclude, Phase VI is

$$\mathsf{R} \approx \frac{\Delta d^2}{8n}, \quad \text{if} \quad n \gg d^2 \quad \text{and} \quad \lambda \ll \sqrt{\frac{n}{d^2}}. \tag{182}$$

### D.4 Universal error decomposition of feature learning

In this section we derive Result 3.

(i) When a part of the spikes are outside the bulk and a part of the spikes are inside, and the regularization does not cut off the bulk (e.g., in phases IV and V), we can rewrite the SE as

$$\begin{cases}
4\tilde{\alpha}\delta^2 - \frac{\delta^2}{\epsilon} = \delta\partial_\delta(J_1(\delta, \lambda\epsilon) + J_2(\delta, \lambda\epsilon)) \\
Q^\star + \frac{\Delta}{2} + 2\tilde{\alpha}\delta^2 - \frac{\delta^2}{\epsilon} = (1 - \lambda\epsilon\partial_{\lambda\epsilon})(J_1(\delta, \lambda\epsilon) + J_2(\delta, \lambda\epsilon)),
\end{cases} \tag{183}$$

where the auxiliary functions are defined as

$$J_1(\delta, \lambda\epsilon) := \frac{1}{d}\sum_{i=1}^{K(\delta)}(s_i + \frac{\delta^2}{s_i} - \lambda\epsilon)^2, \tag{184}$$

and

$$J_2(\delta, \lambda\epsilon) := \delta^2 \int_{\lambda\epsilon/\delta}^{2} \mu_{\mathrm{sc}}(x)(x - \lambda\epsilon/\delta)^2. \tag{185}$$

Recall we are considering a general model with $s_i$ denoting the $i-$th eigenvalue in a descending order and $K(\delta) \ll d$ satisfying $s_{K(\delta)} = \delta$. The excess risk is given by

$$\mathsf{R} := 2\tilde{\alpha}\delta^2 - \frac{\Delta}{2} = Q^\star + (\delta\partial_\delta + \lambda\epsilon\partial_\epsilon - 1)(J_1(\delta, \lambda\epsilon) + J_2(\delta, \lambda\epsilon)). \tag{186}$$

Then we have

$$(\delta\partial_\delta + \lambda\epsilon\partial_{\lambda\epsilon} - 1)J_1(\delta, \lambda\epsilon) = \frac{1}{d}\delta K'(\delta)(2\delta - \lambda\epsilon)^2 + \frac{2}{d}\sum_{i=1}^{K(\delta)}(s_i + \frac{\delta^2}{s_i} - \lambda\epsilon)(\frac{2\delta^2}{s_i} - \lambda\epsilon) - \frac{1}{d}\sum_{i=1}^{K(\delta)}(s_i + \frac{\delta^2}{s_i} - \lambda\epsilon)^2$$

$$= \frac{1}{d}\delta K'(\delta)(2\delta - \lambda\epsilon)^2 + \frac{1}{d}\sum_{i=1}^{K(\delta)}\left[(\frac{\delta^2}{s_i} - \lambda\epsilon)^2 + \frac{\delta^2}{s_i}(s_i + \frac{\delta^2}{s_i} - \lambda\epsilon)\right] - \frac{1}{d}\sum_{i=1}^{K(\delta)}s_i^2. \tag{187}$$

and

$$(\delta\partial_\delta + \lambda\epsilon\partial_{\lambda\epsilon} - 1)J_2(\delta, \lambda\epsilon) = \delta^2 \int_{\lambda\epsilon/\delta}^{2} \mu_{\mathrm{sc}}(x)(x - \lambda\epsilon/\delta)^2. \tag{188}$$

Now we obtain eq. (17) by using $Q^\star := \frac{1}{d}\sum_{i=1}^{d}s_i^2$.

(ii) When the regularization cuts off the bulk and a part of the spikes (e.g., in phase III), we can rewrite the SE as

$$\begin{cases} 4\tilde{\alpha}\delta^2 - \frac{\delta^2}{\epsilon} = \delta\partial_\delta J_1(\delta, \lambda\epsilon) \\ Q^\star + \frac{\Delta}{2} + 2\tilde{\alpha}\delta^2 - \frac{\delta^2}{\epsilon} = (1 - \lambda\epsilon\partial_{\lambda\epsilon})J_1(\delta, \lambda\epsilon), \end{cases} \tag{189}$$

where the auxiliary function is still defined as

$$J_1(\delta, \lambda\epsilon) := \frac{1}{d}\sum_{i=1}^{K(\delta, \lambda\epsilon)}(s_i + \frac{\delta^2}{s_i} - \lambda\epsilon)^2, \tag{190}$$

but the cutoff becomes $s_{K(\delta, \lambda\epsilon)} + \frac{\delta^2}{s_{K(\delta, \lambda\epsilon)}} - \lambda\epsilon = 0$. The excess risk is given by

$$\mathsf{R} := 2\tilde{\alpha}\delta^2 - \frac{\Delta}{2} = Q^\star + (\delta\partial_\delta + \lambda\epsilon\partial_{\lambda\epsilon} - 1)J_1(\delta, \lambda\epsilon)$$

$$= Q^\star + \frac{2}{d}\sum_{i=1}^{K(\delta, \lambda\epsilon)}(s_i + \frac{\delta^2}{s_i} - \lambda\epsilon)(\frac{2\delta^2}{s_i} - \lambda\epsilon) - \frac{1}{d}\sum_{i=1}^{K(\delta, \lambda\epsilon)}(s_i + \frac{\delta^2}{s_i} - \lambda\epsilon)^2 \tag{191}$$

$$= \frac{1}{d}\sum_{i=1}^{K(\delta, \lambda\epsilon)}\left[(\frac{\delta^2}{s_i} - \lambda\epsilon)^2 + \frac{\delta^2}{s_i}(s_i + \frac{\delta^2}{s_i} - \lambda\epsilon)\right] + \frac{1}{d}\sum_{i=K(\delta, \lambda\epsilon)+1}^{d}s_i^2,$$

which gives eq. (18).

## D.5 PERTURBATIVE EXPANSION OF THE BULK

In this session we discuss how to obtain the correction of the bulk in eq. (119). Consider $H := Z + \sum_{i=1}^{k}\lambda_i v_i v_i^T$ with $\{\lambda_i\}_{i=1}^{k}$ smaller than 1, where $Z \sim \mathrm{GOE}(d)$ and $\{v_i\}_{i=1}^{k}$ are uniformly sampled from the unit sphere. Its resolvent can be expanded as

$$m_H(z) := \frac{1}{d}\mathrm{Tr}(z - H)^{-1}$$

$$\approx m_Z(z) + \frac{1}{d}\sum_{i=1}^{k}\lambda_i v_i^T(z - Z)^{-2}v_i + \frac{1}{d}\sum_{i,j=1}^{k}\mathrm{Tr}(z - Z)^{-1}v_i v_i^T(z - Z)^{-1}v_j v_J^T(z - Z)^{-1}. \tag{192}$$

For the first-order correction we have $\frac{1}{d}\sum_{i=1}^{k}\lambda_i v_i^T(z-Z)^{-2}v_i \approx \frac{\sum_{i=1}^{k}\lambda_i}{d}m_Z'(z)$. For the second-order correction we have

$$\frac{1}{d}\sum_{i,j=1}^{k}\mathrm{Tr}(z-Z)^{-1}v_i v_i^T(z-Z)^{-1}v_j v_j^T(z-Z)^{-1}$$

$$\approx \frac{\mathrm{Tr}(z-Z)^{-1}\,\mathrm{Tr}(z-Z)^{-2} + \mathrm{Tr}(z-Z)^{-3}}{d^2(d+2)}\sum_{i=1}^{k}\lambda_i^2 \qquad (193)$$

$$\approx -\frac{1}{d}\sum_{i=1}^{k}\lambda_i^2 m_Z(z)m_Z'(z).$$

This gives a correction on the spectrum $\tilde{\nu}(x) = \frac{\sum_{i=1}^{k}\lambda_i}{d}\mu_{\mathrm{sc}}'(x) - \frac{\sum_{i=1}^{k}\lambda_i^2}{d}\mathrm{im}(m_Z(x+i0)m_Z'(x+i0))$. Note that the first term is an odd function and the second term is an even function, and the resolvant of GOE is given by

$$m_Z(x+i0) = \frac{x}{2} + i\mu_{\mathrm{sc}}(x), \qquad (194)$$

so we have

$$\frac{4\pi^2}{3}\int \nu_H(x)^3 dx \approx \frac{4\pi^2}{3}\int \mu_{\mathrm{sc}}(x)^3 dx + 4\pi^2\int \mu_{\mathrm{sc}}(x)^2\tilde{\nu}(x)dx$$

$$= 1 - \left(\frac{1}{d}\sum_{i=1}^{k}\lambda_i^2\right)2\pi^2\int \mu_{\mathrm{sc}}(x)^2(x\mu_{\mathrm{sc}}'(x) + \mu_{\mathrm{sc}}(x))dx$$

$$= 1 - \left(\frac{1}{d}\sum_{i=1}^{k}\lambda_i^2\right)\frac{4\pi^2}{3}\int \mu_{\mathrm{sc}}(x)^3 dx \qquad (195)$$

$$= 1 - \frac{1}{d}\sum_{i=1}^{k}\lambda_i^2.$$

This correction is the leading term only if $\sum_{i=1}^{k}\lambda_i^2 \ll \sum_{i=1}^{k}\lambda_i^2$. However, if $\lambda_i = i^{-\gamma}$, all the higher-order terms are of the same order, but their sum converges for $|z| > 3$, which gives

$$\frac{4\pi^2}{3}\int \nu_H(x)^3 dx = 1 - \Theta\left(\frac{1}{d}\sum_{i=1}^{k}\lambda_i^2\right) \qquad (196)$$

instead.

### D.6 NOISELESS SETTING

#### D.6.1 BAYESIAN ESTIMATOR

In this section we solve eq. (19) for $\Delta = 0$. For $\Delta = 0$, the first equation of eq. (19) gives $\hat{q} = \frac{2\tilde{\alpha}}{Q^\star - q} = \frac{\tilde{\alpha}}{\mathsf{R}^{\mathrm{BO}}}$.

**Phase I: Perfect recovery**  We can verify that $\tilde{\alpha} = \frac{c}{2}$ and $\mathsf{R}^{\mathrm{BO}} = 0$ is a solution of eq. (19) for $\Delta = 0$ because

$$\lim_{\hat{q}\to\infty}\frac{4\pi^2}{3\hat{q}}\int \mu_{1/\sqrt{\hat{q}}}(x)^3 \mathrm{d}x = 1 - c. \qquad (197)$$

Then $\alpha = \frac{c}{2}$ is the perfect recovery threshold, which suggests that

$$\mathsf{R}^{\mathrm{BO}} = 0 \quad \text{if} \quad n > \frac{c}{2}d^2. \qquad (198)$$

**Phase II: Under-sampling** if $d \ll n \ll d^2$, eq. (132) combined with eq. (19) gives

$$-2\tilde{\alpha} = \Theta\left(d^{-1}\left(\frac{\tilde{\alpha}d}{MMSE}\right)^{1/2\gamma}\right),$$
(199)

and thus

$$\mathsf{R}^{\mathrm{BO}} = \Theta\left(\left(\frac{n}{d}\right)^{2\gamma-1}\right).$$
(200)

Note that for this case we still have $\delta^{1/\gamma}d^{-1} \ll 1$, so the derivation of eq. (132) is valid.

**Phase III: Not enough data** If $n \ll d$, eq. (135) combined with eq. (19) gives

$$2\tilde{\alpha} = Q^\star \frac{2\tilde{\alpha}}{Q^\star - q},$$
(201)

and thus $\mathsf{R}^{\mathrm{BO}} := Q^\star - q = Q^\star$.

### D.6.2 ERM

In this section we solve the equations in Appendix D.3 for $\Delta = 0$.

**Phase I: Trivial Phase** The results in Phase I of Appendix D.3 holds also for $\Delta = 0$, which gives Phase I:

$$\mathsf{R} = Q^\star, \quad \text{if} \quad n < \frac{1}{8}Q^\star d \quad \text{or} \quad \lambda > \frac{4n}{d^{3/2}}.$$
(202)

**Phase II: Over-regularization phase** The analysis of Phase II in Appendix D.3 also holds for $\Delta = 0$, which gives

$$\begin{cases} 4\tilde{\alpha}\delta - \frac{\delta}{\epsilon} = 0 \\ Q^\star + 2\tilde{\alpha}\delta^2 - \frac{\delta^2}{\epsilon} = Q^\star - \frac{2\gamma}{2\gamma-1}\left(\frac{\lambda\epsilon}{\sqrt{d}}\right)^{\frac{2\gamma-1}{\gamma}}. \end{cases}$$
(203)

The leading order solution is $\epsilon = \frac{1}{4\alpha}$, $\delta^2 = \frac{\gamma}{(2\gamma-1)\tilde{\alpha}}\left(\lambda\frac{d^{3/2}}{4n}\right)^{\frac{2\gamma-1}{\gamma}}$ and

$$\mathsf{R} := 2\tilde{\alpha}\delta^2 = \frac{2\gamma}{2\gamma-1}\left(\lambda\frac{d^{3/2}}{4n}\right)^{\frac{2\gamma-1}{\gamma}}.$$
(204)

The condition $\max(\delta, d^{-\gamma+1/2}) \ll \lambda\epsilon \ll \sqrt{d}$ gives

$$\max\left(\frac{d^{\gamma-3/2}}{n^{\gamma-1}}, \frac{n}{d^{\gamma+\frac{3}{2}}}\right) \ll \lambda \ll \frac{d^{3/2}}{n}.$$
(205)

**Phase III: Intermediate over-regularization phase** Phase III is given by the analysis of Phase III in Appendix D.3. Eq. (153) solves

$$\epsilon = \frac{1}{4\tilde{\alpha}}, \quad \delta^2 = \frac{\lambda^2 d^6}{32n^3}$$
(206)

for $\Delta = 0$. Thus we have

$$\mathsf{R} := 2\tilde{\alpha}\delta^2 = \frac{\lambda^2 d^4}{16n^2}.$$
(207)

The condition $\delta \ll \lambda\epsilon \ll d^{-\gamma+1/2}$ gives

$$\lambda \ll \frac{n}{d^{\gamma+3/2}} \quad \text{and} \quad n \gg d^2.$$
(208)

**Phase IV: Under-sampling phase**  Phase IV is given by the analysis of Phases IV and V in Appendix D.3. Eq. (159) reduces to

$$2\tilde{\alpha}\delta^2 = \left(\left(6 - \frac{1}{2\gamma} - \frac{4}{1+\gamma} + \frac{1}{1+2\gamma}\right)\frac{\gamma-1}{\gamma} + \frac{4}{\gamma}\right)\left(\frac{\delta}{\sqrt{d}}\right)^{2-\frac{1}{\gamma}} + \delta^2\frac{16t^{7/2}}{105\pi} \tag{209}$$

for $\Delta = 0$ and $\frac{16}{15\pi}t^{5/2} \approx 4\tilde{\alpha}$, where we assume $\lambda \ll \tilde{\alpha}\delta$. Then the second term on the right side is negligible since $t^{7/2} \ll \tilde{\alpha}$. (209) gives

$$\delta = 2^{-\gamma}\left(\left(6 - \frac{1}{2\gamma} - \frac{4}{1+\gamma} + \frac{1}{1+2\gamma}\right)\frac{\gamma-1}{\gamma} + \frac{4}{\gamma}\right)^{\gamma}\frac{d^{\gamma+1/2}}{n^{\gamma}}, \tag{210}$$

and then we further have

$$\mathsf{R} := 2\tilde{\alpha}\delta^2 = 2^{-2\gamma+1}\left(\left(6 - \frac{1}{2\gamma} - \frac{4}{1+\gamma} + \frac{1}{1+2\gamma}\right)\frac{\gamma-1}{\gamma} + \frac{4}{\gamma}\right)^{2\gamma}\left(\frac{d}{n}\right)^{2\gamma-1}. \tag{211}$$

The condition $d \ll n \ll d^2$ and $\lambda \ll \alpha\delta, 0 < 2 - \frac{\lambda\epsilon}{\delta} \ll 1, d^{-\gamma+\frac{1}{2}} \ll \delta \ll \sqrt{d}$ reduces to

$$\lambda \ll \frac{d^{\gamma-3/2}}{n^{\gamma-1}} \quad \text{and} \quad d \ll n \ll d^2, \tag{212}$$

where we use $\delta^2 t^{5/2} \sim 4\tilde{\alpha}\delta^2 - \frac{\lambda\delta}{2} > 0$. Note that we further have $t^{5/2} \sim 4\tilde{\alpha} - \frac{\lambda}{2\delta} \ll 1$ satisfied.

## E  COMPARISON WITH $L_2$ REGULARIZATION

In this section we compare the scaling laws we have obtained for ERM to the ones of ridge regression for a linear model, proving in particular their sub-optimality. The ridge estimator is defined as

$$\hat{\boldsymbol{\theta}}_{\text{ridge}} = \arg\min_{\boldsymbol{\theta}\in\mathbb{R}^d}\frac{1}{n}\sum_{\mu=1}^{n}(y_\mu - \langle\boldsymbol{\theta}, \boldsymbol{x}_\mu\rangle)^2 + \lambda\|\boldsymbol{\theta}\|_2^2. \tag{213}$$

This can be mapped to both the diagonal network and quadratic network case, depending on the choice of $\boldsymbol{x}$. For simplicity, we assume $\boldsymbol{x} \sim \mathcal{N}(0, I_d)$.

Cheng & Montanari (2024) readily implies the following.

**Theorem 2** (Excess risk rates for ridge regression). *. Assume that $y = \langle\boldsymbol{\theta}^\star, \boldsymbol{x}\rangle + \sqrt{\Delta}\zeta$ with $\zeta \sim \mathcal{N}(0, 1)$ and $\mathbb{E}[\|\boldsymbol{\theta}^\star\|_2^2] = \Theta(1)$. For $n, d \gg 1$, the excess risk associated to the estimator defined in (213) satisfies*

$$\mathsf{R}_{n,d} = \Theta\begin{cases} 1, & \text{if} \quad n \ll d \quad \text{or} \quad \lambda \gg 1, \\ \lambda^2, & \text{if} \quad n \gg d \quad \text{and} \quad \sqrt{d/n} \ll \lambda \ll 1 \\ \Delta d/n, & \text{if} \quad n \gg d \quad \text{and} \quad \lambda \ll \sqrt{d/n} \\ \Delta\lambda^{-1/2}, & \text{if} \quad n = d \quad \text{and} \quad \lambda \ll 1. \end{cases} \tag{214}$$

*Proof.* The excess risk concentrates with high probability, for $n, d \gg 1$, around the following deterministic expression (Cheng & Montanari, 2024)

$$\mathsf{R}_{n,d} = \frac{n\nu^2}{n(1+\nu)^2 - d}\mathbb{E}[\|\boldsymbol{\theta}^\star\|_2^2] + \frac{d\Delta}{n(1+\nu)^2 - d} \tag{215}$$

with $\nu$ the unique non-negative solution of

$$\frac{n}{d}\left(1 - \frac{\lambda}{\nu}\right) = \frac{1}{1+\nu}. \tag{216}$$

Therefore

$$\nu = \Theta\begin{cases} d/n + \lambda, & \text{if} \quad n \ll d, \\ \lambda, & \text{if} \quad n \gg d, \\ \sqrt{\lambda}, & \text{if} \quad n = d \quad \text{and} \quad \lambda \ll 1. \end{cases} \tag{217}$$

Substituting into (215), the result follows. $\square$

Therefore, we cannot obtain a non-trivial risk for $n \ll d$ with $L_2$ regularization.

# F    NUMERICAL DETAILS

The state equations (20) can be iterated in a more numerically convenient form that depends on an extended set of parameters $(q, m, \Sigma, \hat{q}, \hat{m}, \hat{\Sigma})$ as presented in Erba et al. (2025), Appendix A.4.4. Then, we can compute $\delta$ and $\epsilon$ as $\delta = \sqrt{\hat{q}}/\hat{m}$ and $\epsilon = 2/\hat{m}$. The limiting distribution of $S^*$ is not easy to compute exactly as $d \to \infty$, and for finite $d$ the contribution of the spikes to the state evolution is the sum $J_1(\delta, \lambda\epsilon)$, whose upper limit depends on the width of the bulk $\delta$. The state equations include derivatives of this sum, where delta spikes are neglected due to the numerical difficulty of representing them. Having obtained results that do not match the experiments with these approximations, we resorted to computing the integral $J$ by a Monte-Carlo procedure. The overlaps $m, q, \Sigma$ are computed using finite size samples of matrices and their eigen-decomposition at each step $t$ of the state evolution

$$M = \sqrt{\hat{q}^t} Z + \hat{m}^t S^* = O \operatorname{diag}(\nu_1, \ldots, \nu_d) O^T \tag{218}$$

where $Z \sim \operatorname{GOE}(d)$ and $S^* = \frac{\sqrt{d}}{\sum_i i^{-2\gamma}} \operatorname{diag}(1, 2^{-\gamma}, \cdots, d^{-\gamma})$ can be taken as a diagonal matrix, since this amounts to a rotation of $M$, which does not affect the distribution of $Z$ by rotational invariance. One can then apply the spectral denoiser described in Erba et al. (2025) and compute the overlaps using the reconstructed matrix $\tilde{M} = O \operatorname{diag}(\tilde{\nu}_1, \ldots, \tilde{\nu}_d) O^T$, where $\tilde{\nu}_i = \frac{1}{\hat{\Sigma}} \operatorname{ReLu}(\nu_i - 2\lambda)$ are the denoised eigenvalues. Finally, the order parameters can be computed as

$$\begin{cases} m^{t+1} = \frac{1}{d} \mathbb{E}_M \operatorname{Tr}[(S^*)^T \tilde{M}] \\ q^{t+1} = \frac{1}{d} \mathbb{E}_M \operatorname{Tr}[\tilde{M}^T \tilde{M}] \\ \Sigma^{t+1} = \frac{2}{d} \mathbb{E}_M \left[ \sum_{i=1}^d \frac{\Theta(\nu_i - 2\lambda)}{\hat{\Sigma}} + \sum_{i<j} \frac{\tilde{\nu}_i - \tilde{\nu}_j}{\nu_i - \nu_j} \right] \end{cases} \tag{219}$$

The expectation is taken over $n_{samples} \sim 10$ samples for $d \sim 10^2$, independently for each order parameter, for a total of $3n_{samples}$ sampled matrices per iteration of the state evolution.

Since the ERM problem is convex we resort to using LBFGS with Wolfe line search. We used the PyTorch implementation of the optimiser, taking care of evaluating the network efficiently at each pass. For the specifics of the implementation we refer to the code repository https://github.com/SPOC-group/QuadraticNetPowerlaw. Convergence is typically achieved with a precision of at least $10^{-8}$ in a few hundred iterations. The main challenge is in storing the dataset in memory. For each run we used up to $1800$ gigabytes of RAM on nodes with 2 Intel Xeon 8360Y CPUs. Our total computing cost (including intial explorations) is around $250000$ CPU hours.

