# OpenReview forum: "Scaling Laws and Spectra of Shallow Neural Networks in the Feature Learning Regime"
_ICLR.cc/2026/Conference — ICLR 2026 Oral_

### Official Review · Reviewer_XduY · 2025-10-29

**Soundness:** 4
**Presentation:** 4
**Contribution:** 4
**Rating:** 8
**Confidence:** 4

**Summary:**

The paper obtains precise limits of the test loss and weight spectrum for two models of shallow (linear/quadratic) networks. A complete phase diagram is given as a function of the regularization intensity and effective sample size.

**Strengths:**

The paper gives a very complete answer for the scaling laws of the regularized global minimum in these two simplified models. These networks implicitely add a L1-type of bias, comparison to previous work on random feature models/kernel methods which are in a L2 setting. The tools for the L2 setting are now well established (Random Matrix Theory etc), but the L1 case was to my knowledge less studied.

The paper is very clear, and the authors have done a great job of explaining the different phases and relate them to behavior of the weight spectrum, which helps a lot to not be lost in the many big formulas.

**Weaknesses:**

The paper only studies diagonal linear networks and quadratic networks with fixed second layer weights, which is basically a shallow fully-connected linear network with symmetric weights. These are very simple models, far from anything used in practice, but one has to start somewhere.

**Questions:**

- Do the L1 models that you consider outperform their L2 counterparts in general or in specific settings? And if so by how much? Since you seem to consider true functions that are not exactly sparse but only have a certain decay, I would imagine that the advantage of the L1 approach would appear for sufficiently steep decay of the true function spectrum?

---

> ### Author Response · Authors · 2025-11-19
>
> We thank the reviewer for the XduY of our work and for the feedback which will help improving the manuscript. All suggestion are implemented in the updated version.
>
> 1. *Weakness 1: The paper only studies diagonal linear networks and quadratic networks with fixed second layer weights, which is basically a shallow fully-connected linear network with symmetric weights. These are very simple models, far from anything used in practice, but one has to start somewhere.*
>
> Please see our summary rebuttal.
>
> 2. *Question 1: Do the L1 models that you consider outperform their L2 counterparts in general or in specific settings? And if so by how much? Since you seem to consider true functions that are not exactly sparse but only have a certain decay, I would imagine that the advantage of the L1 approach would appear for sufficiently steep decay of the true function spectrum?*
>
> In our setting, one can show that with $L_2$ regularization, the generalization error remains constant when $n_{\rm eff} \ll d$ and decays as $1/n_{\rm eff}$ when $n_{\rm eff} \gg d$. Therefore, $L_1$ regularization is necessary to achieve nontrivial performance. Please see Appendix D for details.

---

### Official Review · Reviewer_jRAX · 2025-10-31

**Soundness:** 4
**Presentation:** 3
**Contribution:** 3
**Rating:** 8
**Confidence:** 4

**Summary:**

This paper provides a theoretical framework for neural scaling laws in shallow neural networks that go beyond the random features (lazy training) regime by explicitly modeling feature learning. It focuses on two analytically tractable architectures:

1) Diagonal networks — equivalent to LASSO (l1-regularized linear regression).

2) Quadratic networks — equivalent to matrix compressed sensing (low-rank estimation with nuclear norm regularization).

Through these mappings, the authors derive:

A complete phase diagram (Fig. 1, p. 6) of generalization excess risk as a function of sample size, model dimension and weight decay.

A universal scaling law showing crossovers between plateaus, fast- and slow-decay regimes, and double-descent–like interpolation peaks.

A connection between scaling phases and spectral properties of the learned weights, yielding a theoretical explanation for the empirically observed power-law weight spectra in modern deep networks.

The analysis combines approximate message passing (AMP), state evolution, and random matrix theory, yielding results that match both Bayes-optimal rates and extensive numerical simulations (Figs. 2–6, pp. 7–25).

**Strengths:**

Novel analytical unification:
The paper builds an elegant bridge between feature-learning neural networks, LASSO, and matrix compressed sensing, allowing exact asymptotic characterization of generalization in previously intractable regimes.

Comprehensive phase diagram: Figure 1 neatly summarizes all scaling regimes and their transitions. It captures empirical phenomena (plateaus, overfitting peaks, fast decays) reported in large-scale scaling-law studies (Kaplan et al. 2020; Paquette et al. 2024).

Spectral–generalization link: The “universal error decomposition” (Result 3) expresses the generalization error as an explicit functional of the weight eigenvalue spectrum, providing the first first-principles explanation of heavy-tailed spectra linked to generalization quality.

**Weaknesses:**

Restricted architecture scope: Results hold for shallow (two-layer) diagonal or quadratic networks; extensions to multilayer or non-polynomial activations remain speculative. It’s unclear how much insight transfers to modern deep transformers or CNNs.

Dependence on heuristic AMP validity: The central results (e.g., Eq. 19) assume non-asymptotic validity of state evolution, supported empirically but not proven. The authors acknowledge this (Sec. 2.4) as a conjecture, so the paper’s main theorems are partially non-rigorous.

Target-model assumptions: The teacher–student setup with Gaussian inputs and power-law eigenvalues is mathematically convenient but idealized. Real data distributions or nonlinear activations could alter scaling transitions.

**Questions:**

1) Can the spectral–generalization decomposition (Eq. 17) be extended to multi-layer or nonlinear activations?

2) How sensitive are your phase boundaries to non-Gaussian inputs or correlated features?

3) Is there a principled path toward proving the non-asymptotic AMP conjecture (Eq. 19)?

4) Does the observed \lambda^{-2/3} interpolation scaling have a connection to known double-descent exponents in overparameterized deep models?

---

> ### Author Response · Authors · 2025-11-19
>
> We thank the reviewer for the appreciation of our work and for the feedback which will help improving the manuscript. All suggestion are implemented in the updated version.
>
> 1. *Weakness 1: Restricted architecture scope: Results hold for shallow (two-layer) diagonal or quadratic networks; extensions to multilayer or non-polynomial activations remain speculative. It’s unclear how much insight transfers to modern deep transformers or CNNs.*
>
>     *Question 1: Can the spectral–generalization decomposition (Eq. 17) be extended to multi-layer or nonlinear activations?*
>
> Please see our summary rebuttal.
>
> 2. *Weakness 2: Dependence on heuristic AMP validity: The central results (e.g., Eq. 19) assume non-asymptotic validity of state evolution, supported empirically but not proven. The authors acknowledge this (Sec. 2.4) as a conjecture, so the paper’s main theorems are partially non-rigorous.*
>
>     *Question 3: Is there a principled path toward proving the non-asymptotic AMP conjecture (Eq. 19)?*
>
> We acknowledge that our current results rely on a non-asymptotic characterization of the AMP state evolution which we conjecture (Conjecture 1), supported by extensive numerical checks. Proving such a conjecture is an important open problem, which has been partially addressed in related but different settings. The closer is Cheng and Montanari (2024), who proved a result analogous to our conjecture but for ridge regression. In the context of LASSO, Miolane and Montanari (2021) derived rigorous convergence rates for state evolution assuming $n_{\rm eff}/d$ remains bounded. Similar convergence rates also appear in Rush and Venkataramanan (2018) for more general i.i.d. priors. Extending the ideas of Chen and Lam (2021) to the LASSO setting present substantial technical challenge, and proving Conjecture 1 would require an additional layer of random matrix theory on the top of such a result. We have clarified this discussion in the revised manuscript.
>
> 3. *Weakness 3: Target-model assumptions: The teacher–student setup with Gaussian inputs and power-law eigenvalues is mathematically convenient but idealized. Real data distributions or nonlinear activations could alter scaling transitions.*
>
> The assumption of power-law target coefficients is common in the kernel literature, where it is known as a *source condition*, c.f. Caponnetto and Vito (2007). We acknowledge that the choice of i.i.d. Gaussian covariate is due to technical simplicity, and plan to extend it to power-law covariance in future work. Regarding nonlinear activations, as noted in our reply to Weakness 1 and Question 1, we expect similar behavior to persist if the universality conjecture of Barbier et al. (2025) holds.
>
> 4. *Question 2: How sensitive are your phase boundaries to non-Gaussian inputs or correlated features?*
>
> Due to Gaussian universality (Chem and Lam 2021), we expect our results to remain valid for i.i.d. non-Gaussian data. However, when the input features exhibit strong correlations, e.g., following a power-law covariance, the phase boundaries will depend on the interplay of the target and covariance eigenvalues decay. Analyzing this effect is an important direction for future research.
>
> 5. *Question 4: Does the observed $\lambda^{-2/3}$ interpolation scaling have a connection to known double-descent exponents in overparameterized deep models?*
>
> To the best of our knowledge, no prior work has reported such an interpolation scaling in the context of double descent. We would greatly appreciate any pointers to related literature from the reviewers.
>
> **References**:
>
> *Chen Cheng and Andrea Montanari. Dimension free ridge regression. The Annals of Statistics,
> 52(6):2879–2912, 2024.*
>
> *Leo Miolane and Andrea Montanari. The distribution of the lasso. The Annals of Statistics, 49(4):2313–
> 2335, 2021.*
>
> *Cynthia Rush and Ramji Venkataramanan. Finite sample analysis of approximate message passing
> algorithms. IEEE Transactions on Information Theory, 64(11):7264–7286, 2018.*
>
> *Wei Kuo Chen and Wai-Kit Lam. Universality of approximate message passing algorithms. Electronic
> Journal of Probability, 26:36, 2021.*
>
> *Andrea Caponnetto and Ernesto De Vito. Optimal rates for the regularized least-squares algorithm.
> Foundations of Computational Mathematics, 7(3):331–368, 2007.*
>
> *Jean Barbier, Francesco Camilli, Minh-Toan Nguyen, Mauro Pastore, and Rudy Skerk. Statistical
> physics of deep learning: Optimal learning of a multi-layer perceptron near interpolation. arXiv preprint
> arXiv:2510.24616, 2025.*

---

### Official Review · Reviewer_d5QU · 2025-11-01

**Soundness:** 4
**Presentation:** 2
**Contribution:** 4
**Rating:** 6
**Confidence:** 4

**Summary:**

This work extends neural scaling law theory  by analyzing quadratic and diagonal neural networks in the feature learning regime, using tools from matrix compressed sensing and LASSO to map out how excess risk scales with data, model size, and weight decay. The authors identify distinct scaling regimes, plateaus, and spectral signatures (like power-law tails in the weight distribution) and show how these phenomena predict generalization, providing a first-principles explanation for trends observed empirically in large neural networks and the correlation between power-law tails in the weight distribution and generalization in neural networks.

**Strengths:**

1. The paper gives a unified phase diagram of generalization error across sample size, width, and regularization for diagonal and quadratic networks (true feature learning, not just kernels).

2. Exploring the theoretical relationship between the weight spectrum and generalization is an important open problem, and this paper provides an explanation for it in certain special neural network architectures. It analytically characterizes the trained weight spectra (bulk, spikes, heavy tails) in every regime and links these spectral features directly to underfitting, overfitting, and approximation error.

3. It explains benign vs harmful overfitting and double-descent–like peaks from first principles, and shows how optimal regularization suppresses noisy bulk without destroying learned signal.

**Weaknesses:**

**If you address the issues in the Weaknesses and Question sections, I will increase the score.**

1. The writing of the paper could be improved. Some of the notation is not explained very clearly (see  Questions sections), and also it seems the paper does not include a conclusion section.
2. The current setting is limited to special two-layer neural networks, which allows the study of their excess risk minimizer to be reduced to matrix compressed sensing and Lasso problem. I would like to know whether, for more general two-layer (or even multi-layer) neural networks, there is a similar way to reduce the problem. If this is difficult, do you think it would be possible instead to study the training dynamics of the weight spectrum under (stochastic) optimizers in order to explore the relationship between heavy-tailed weight spectra and training dynamics?
3. In the paper, you assume that the spectrum of the target network weights follows a power-law distribution (which is  heavy-tailed ). Is this assumption necessary[1]? and how much does this heavy-tailed structure affect the final weight spectrum of the student network?
4. Minor: It looks like in Figure 3, the third plot in the first row should be labeled with the exponent $-2 + \frac{1}{\gamma}$.


[1] Gurbuzbalaban, Mert, Umut Simsekli, and Lingjiong Zhu. "The heavy-tail phenomenon in SGD." International Conference on Machine Learning. PMLR, 2021.

**Questions:**

In addition to the issues mentioned in the Weaknesses section, I may also have the following questions.

1. In line 473, what does the first  $R$ represent? In addition, what is the relationship between the $R_{n,d}$ of ERM mentioned here (line 479) and the $R$ that appears in equations (11), (14), and (17)?

2. There are also some papers [2][3][4][5][6][7][8][9][10] that discuss the relationship between the spectrum of the weights / features / data and generalization or neural scaling law. Could you provide some comments and discussion on these papers?




[2] Bartlett, Peter L., et al. "Benign overfitting in linear regression." Proceedings of the National Academy of Sciences 117.48 (2020): 30063-30070.

[3] Simsekli, Umut, et al. "Hausdorff dimension, heavy tails, and generalization in neural networks." Advances in Neural Information Processing Systems 33 (2020): 5138-5151.

[4] Hodgkinson, Liam, et al. "Generalization bounds using lower tail exponents in stochastic optimizers." International Conference on Machine Learning. PMLR, 2022.

[5] Wang, Yutong, Rishi Sonthalia, and Wei Hu. "Near-interpolators: Rapid norm growth and the trade-off between interpolation and generalization." International Conference on Artificial Intelligence and Statistics. PMLR, 2024.

[6] Wang, Zhichao, et al. "Spectral evolution and invariance in linear-width neural networks." Advances in neural information processing systems 36 (2023): 20695-20728.

[7] Dandi, Yatin, et al. "A random matrix theory perspective on the spectrum of learned features and asymptotic generalization capabilities." arXiv preprint arXiv:2410.18938 (2024).

[8] Worschech, Roman, and Bernd Rosenow. "Analyzing Neural Scaling Laws in Two-Layer Networks with Power-Law Data Spectra." arXiv preprint arXiv:2410.09005 (2024).

[9]  Kothapalli, Vignesh, et al. "From spikes to heavy tails: Unveiling the spectral evolution of neural networks." Transactions on Machine Learning Research (2025).

[10] Arous, Gérard Ben, et al. "Learning quadratic neural networks in high dimensions: SGD dynamics and scaling laws." arXiv preprint arXiv:2508.03688 (2025).

---

> ### Author Response · Authors · 2025-11-19
>
> We thank the reviewer for the positive evaluation of our work and for the feedback which will help improving the manuscript. All suggestion are implemented in the updated version.
>
> 1. *Weakness 1: The writing of the paper could be improved. Some of the notation is not explained very clearly (see Questions sections), and also it seems the paper does not include a conclusion section.*
>
> We have carefully revised the manuscript to ensure that all notations are clearly defined and consistently used. We have also added a conclusion section summarizing our key findings and outlining future research directions.
>
> 2. *Weakness 2: The current setting is limited to special two-layer neural networks, which allows the study of their excess risk minimizer to be reduced to matrix compressed sensing and Lasso problem. I would like to know whether, for more general two-layer (or even multi-layer) neural networks, there is a similar way to reduce the problem. If this is difficult, do you think it would be possible instead to study the training dynamics of the weight spectrum under (stochastic) optimizers in order to explore the relationship between heavy-tailed weight spectra and training dynamics?*
>
> Please see our summary rebuttal.
>
> 3. *Weakness 3: In the paper, you assume that the spectrum of the target network weights follows a power-law distribution (which is heavy-tailed). Is this assumption necessary[1]? and how much does this heavy-tailed structure affect the final weight spectrum of the student network?*
>
> Yes, this assumption is crucial for the rates reported in Result 1 and Figure 1, and is motivated by the widespread *source conditions* from the kernel literature, e.g. Caponnetto et al. (2007). Isotropic target weights were studied by Erba et al. (2025), who showed that the network asymptotic spectral distribution are described by a noisy version of the target spectrum (free convolution with a semi-circle law).
>
> The key difference with respect to [1] is that they study SGD, showing that the SGD noise can induce a heavy-tailed stationary distribution for the weights. To our knowledge, [1] does not discuss the excess risk rates, and it is unclear if this heavy-tailed behavior translates to a scaling law for the risk. Understanding if this is the case, as well as relating these two different mechanisms for heavy-tailed weights is an interesting research direction.
>
> 4. *Weakness 4: Minor: It looks like in Figure 3, the third plot in the first row should be labeled with the exponent $2 - 1/\gamma$.*
>
> Thank you for pointing this out, we have corrected this in the revised version.
>
> 5. *Question 1: In line 473, what does the first $R$ represent? In addition, what is the relationship between the $R_{n,d}$ of ERM mentioned here (line 479) and the $R$ that appears in equations (11), (14), and (17)?*
>
> Thank you for highlighting this ambiguity. In line 473, the first $R$ represent either $R_{\rm BO}$ (as defined in line 268) for the Bayes optimal case or $R(\hat{\boldsymbol{W}},\hat{\boldsymbol{a}})$ in the ERM case. The $R_{n,d}$ mentioned in line 479 refers to the same quantity as the $R_{n_{\rm eff},d}$ appearing in Eqs. (11), (14), and (17), i.e. the asymptotics of the excess risk. We used $R_{n,d}$ instead of $R_{n_{\rm eff},d}$ in line 479 because the conjecture there specifically concerns the quadratic network case. In Eq. (17), $R$ should also be replaced by $R_{n,d}$. We have revised the manuscript accordingly for consistency and clarity.
>
> **References**:
>
> *Andrea Caponnetto and Ernesto De Vito. Optimal rates for the regularized least-squares algorithm.
> Foundations of Computational Mathematics, 7(3):331–368, 2007.*
>
> *Vittorio Erba, Emanuele Troiani, Lenka Zdeborova, and Florent Krzakala. The nuclear route: Sharp
> asymptotics of ERM in overparameterized quadratic networks. NeurIPS 2025, 2025. arXiv preprint
> arXiv:2505.17958.*

---

> > ### Author Response · Authors · 2025-11-19
> >
> > 6. *Question 2: There are also some papers [2][3][4][5][6][7][8][9][10] that discuss the relationship between the spectrum of the weights / features / data and generalization or neural scaling law. Could you provide some comments and discussion on these papers?*
> >
> > Thank you for pointing out these related works. We appreciate the opportunity to discuss them and have included all of them in the revised manuscript.
> >
> > Our results extend the phenomenon of benign overfitting from linear models [2,5] to nonlinear settings. Specifically, when $\lambda=0^+$ and $1\ll n_{\rm eff}\ll d$, we find that the training error vanishes while the generalization error remains small. Moreover, we identify two distinct sub-phases within this region, where the error first decreases and then increases with $n_{\rm eff}$. While [2,5] primarily analyze the role of data covariance, our current model assumes i.i.d. Gaussian data; extending our framework to incorporate structured data covariance is an interesting direction for future work.
> >
> > References [3,4] study the relationship between the heavy-tail exponents induced by SGD and generalization. Our model does not explicitly capture this effect, since the student’s heavy-tail exponent at the global minimum coincides with that of the teacher. As discussed in our reply to Weakness 3, this is likely due to the convex nature of our model, and it motivates extending our analysis to non-convex dynamics and to SGD.
> >
> > Many empirical findings reported in [6] are consistent with our theoretical results. For instance, [6] argues that “it is the alignments between the features learned from the heavy-tailed part and the features in the teacher model that finally determine the generalization error of NNs”, which is precisely quantified in our Result 3.
> >
> > Reference [7] shows that after a single gradient step, the spectrum becomes a “noise + one spike” structure, suggesting that one feature is learned. Our analysis generalizes this observation by quantifying how many features are learned and how well they are recovered at convergence—where these learned features also correspond to spectral spikes. [9] suggests that spectra can become heavy-tailed after a few steps under sufficiently large learning rates. This early-phase dynamics is not captured by our current theory but represents a natural direction for future investigation.
> >
> > Reference [8] is also closely related: it studies a similar power-law teacher–student setting but solves the dynamics explicitly only for linear activations, providing semi-quantitative results for nonlinear ones. Our phase diagram, in contrast, is more comprehensive but static—we do not yet model the training dynamics, which we plan to explore in follow-up work.
> >
> > Finally, [10] is directly related to our setting. It studies an almost identical teacher–student model, except with a noiseless teacher and without L2 regularization, and derives the error scaling of online SGD. Interestingly, this scaling matches that of our Phase IV. Extending [10]’s framework to more general settings is a promising future direction.

---

> > > ### Comment · Reviewer_d5QU · 2025-11-21
> > > **Respond to Authors**
> > >
> > > Thank you for your response. I think most of my concerns have been reasonably addressed. I suggest that the authors include all the corresponding discussion and clarifications in the camera-ready version. Given the importance of studying the relationship between the properties of the weight spectrum and generalization, as well as neural scaling laws, I will raise my score.

---

### Official Review · Reviewer_Bf9w · 2025-11-11

**Soundness:** 3
**Presentation:** 3
**Contribution:** 3
**Rating:** 6
**Confidence:** 4

**Summary:**

This paper studied the neural scaling law and analyzed the spectra of the trained weight matrix of two-layer diagonal networks and quadratic neural networks. Their teacher network has the same architecture as the student model, but with certain noise. By transferring diagonal networks to the LASSO problem and transferring quadratic neural networks to matrix compressed sensing, the authors can utilize  Generalized Approximate Message Passing (GAMP)  and its state evolution to compute the final excess risks and the spectral distributions of trained weight matrices. This heuristic computation works well for a power law target, a general ridge parameter, input dimension, and effective sample size. As a result, this paper showed how the scaling of the sample size and regularization affect the generalization, and how different scalings lead to different spectral behavior of the trained weight matrix (e.g., rank collapse, bulk+spikes, heavy-tailed distribution).

**Strengths:**

The paper is well-written, and its results are of interest to the deep learning theory community. To the best of my knowledge, this is the first paper that theoretically explains the relationship between the spectral behavior of learned weights and generalization errors, as well as the 5+1 phases for trained weights observed by Mahoney and Martin in 2019.

**Weaknesses:**

The neural network architectures are too specific. I understand that for general two-layer neural networks, the AMP is too complicated to analyze. However, it would be better to explain the difficulty, if there is any other simple architecture that can still work beyond quadratic or diagonal networks, and what the expected results are for general cases. It would be insightful to provide such information or discussion.
Additionally, there is a lack of training in dynamic analysis. This paper only used the state evolution to analyze the solution of the LASSO and matrix compressed sensing, but did not study the scaling of the training time or computational scaling.

**Questions:**

1. What is $\varepsilon$ in Figure 1? Figure 1 is nice and very important for the presentation. However, it is a little bit messy in the x and y axes. There is a line $\lambda = \sqrt{n_{\text{eff}}/d}$, but when $n_{\text{eff}} = 0$, why do you have $\lambda = 1/d$? You should explain the red lines in the caption of the figure.

2. For VIa and VIb, both risks decay fast, but the spectra of weights behave differently. One is bulk+spikes, and the other is heavy-tailed. Does this mean that we do not need heavy-tailed spectra to get a good generalization error? Martin & Mahoney, 2021a, and Martin et al., 2021, argued that networks with heavy-tailed spectra exhibit optimal generalization capabilities.

3. In results 2, does the trained weight still have independent entries? $\hat{\theta}_i$ are independent for different $i$?

---

> ### Author Response · Authors · 2025-11-19
>
> We thank the reviewer for the positive evaluation of our work and for the feedback which will help improving the manuscript. All suggestion are implemented in the updated version.
>
> 1. *Weakness: The neural network architectures are too specific. I understand that for general two-layer neural networks, the AMP is too complicated to analyze. However, it would be better to explain the difficulty, if there is any other simple architecture that can still work beyond quadratic or diagonal networks, and what the expected results are for general cases. It would be insightful to provide such information or discussion. Additionally, there is a lack of training in dynamic analysis. This paper only used the state evolution to analyze the solution of the LASSO and matrix compressed sensing, but did not study the scaling of the training time or computational scaling.*
>
> Please see our summary rebuttal.
>
> 2. *Question 1: What is $\epsilon$ in Figure 1? Figure 1 is nice and very important for the presentation. However, it is a little bit messy in the x and y axes. There is a line $\lambda=\sqrt{n_{\rm eff}/d}$, but when $n_{\rm eff}=0$, why do you have $\lambda=1/d$? You should explain the red lines in the caption of the figure.*
>
> We have replaced $\epsilon$ with $\mathsf{R}$ (the excess risk) in the updated manuscript. We have also clarified that the left endpoint of the x-axis corresponds to $n_{\rm eff}=1/d$, which is the smallest possible value for the quadratic network (i.e., only one sample). This gives $\lambda=\sqrt{n_{\rm eff}/d}=1/d$. In addition, we have updated the caption to explicitly state that the red lines represent discontinuous phase boundaries.
>
> 3. *Question 2: For VIa and VIb, both risks decay fast, but the spectra of weights behave differently. One is bulk+spikes, and the other is heavy-tailed. Does this mean that we do not need heavy-tailed spectra to get a good generalization error? Martin and Mahoney, 2021a, and Martin et al., 2021, argued that networks with heavy-tailed spectra exhibit optimal generalization capabilities.*
>
> Although VIa and VIb exhibit similar risk decay rates, only VIb achieves the Bayes-optimal rate. In the regime $d \ll n_{\rm eff} \ll d^2$, the optimal performance is reached in phase II with $\lambda=\sqrt{n_{\rm eff}/d}$. The corresponding spectral density shows a transition from outlier-dominated (zero eigenvalues+spikes) to heavy-tailed behavior. This supports the argument of Martin et al. (2021) that heavy-tailed spectra are associated with superior generalization. We have added this discussion in our manuscript.
>
> 4. *Question 3: In results 2, does the trained weight still have independent entries? $\hat{\theta}_i$ are independent for different $i$?*
>
> This is a subtle and important point. For finite-size systems, the trained weights are not strictly independent. However, we expect them to become asymptotically independent in the large-dimensional limit, in a sense similar to Eq. (3.19) of Gerbelot et al. (2023). As Result 2 is presented in an informal manner, a fully rigorous treatment and proof of this property are left for future work.
>
> **Reference**: *Cedric Gerbelot and Raphael Berthier. Graph-based approximate message passing iterations. Information and Inference: A Journal of the IMA, 12(4):2562–2628, 2023.*

---

### Author Response · Authors · 2025-11-19
**Summary rebuttal**

We thank all reviewers for their detailed and constructive feedback. We find the overall positive assessment of our work encouraging. To the best of our knowledge, our work is the first to establish a direct and interpretable mathematical connection between spectral properties and generalization scaling laws, and we believe this perspective may inspire substantial follow-up work.

Below, we address two of the common concerns raised by the reviewers. The other points are addressed in specific answers to the reviewers, and we warmly welcome any further question.

**Specific architecture**: Several reviewers noted that our analysis focuses on specific architectures and characterizes only the global optimum rather than the optimization dynamics. We acknowledge this as both a natural shortcoming but also as a strength: it captures relevant phenomena, yet it is *tractable*.

The technical cornerstone of the analysis is the mapping between the quadratic network and matrix compressive sensing. Such a mapping is specific to the quadratic case, and does not directly extend to deeper architectures or other activations. Extending these results to multilayer networks or non-polynomial activations is a major open challenge requiring new ideas, rather than a technical refinement of our current methods. We note that Barbier et al. (2025) recently proposed a non-rigorous extension to general MLPs; if their conjectures hold, similar error scalings should emerge in the universal phase they identify.

**Training dynamics**: Regarding training dynamics, we agree that this is an important and complementary direction. For quadratic networks, the scaling of online SGD risk in the noiseless case was analyzed by Arous et al. (2025). More broadly, GD dynamics can be studied through dynamic mean-field theory (DMFT); for example, Fan et al. (2025) established convergence rates for related models. We expect this framework could eventually yield training-time scaling laws as well, though such an analysis is beyond the scope of our current work.

**References**:

*Jean Barbier, Francesco Camilli, Minh-Toan Nguyen, Mauro Pastore, and Rudy Skerk. Statistical
physics of deep learning: Optimal learning of a multi-layer perceptron near interpolation. arXiv preprint
arXiv:2510.24616, 2025.*

*Gerard Ben Arous, Murat A Erdogdu, N Mert Vural, and Denny Wu. Learning quadratic neural
networks in high dimensions: SGD dynamics and scaling laws. arXiv preprint arXiv:2508.03688, 2025.*

*Zhou Fan, Justin Ko, Bruno Loureiro, Yue M Lu, and Yandi Shen. Dynamical mean-field analysis
of adaptive langevin diffusions: Replica-symmetric fixed point and empirical bayes. arXiv preprint
arXiv:2504.15558, 2025*

---

### Author Response · Authors · 2025-12-04
**Brief Summary for the Area Chair**

Dear Area Chair,

We are writing to provide a concise, factual overview of the discussion for submission 11227, “Scaling Laws and Spectra of Shallow Neural Networks in the Feature Learning Regime”, in case it is useful as you form your decision.
We engaged with the reviewers, who expressed strong appreciation for the novelty of our work, the importance of the results, and the principled resolution of several open theoretical questions—such as the mechanisms behind benign vs harmful overfitting, the structure of the scaling-law phase diagram, and the quantitative link between weight spectra and generalization.
 We have addressed the main criticisms and updated the manuscript accordingly (changes in red):

**Architectural scope and dynamics.**
Several reviewers noted that our analysis focuses on diagonal and quadratic two-layer networks and characterizes the global optimum rather than the full optimization dynamics. In the summary rebuttal, we clarified that this specific setting is what makes the mapping to LASSO and matrix compressed sensing tractable, while still capturing key feature-learning phenomena (double descent, benign overfitting, spectral phases). We also pointed to recent and closely related work on dynamics (e.g., Ben Arous et al., Fan et al.) and explained that extending our phase diagram to multi-layer/non-polynomial activations or full SGD dynamics is a substantial, non-trivial open direction rather than a straightforward extension.

**Status of Conjecture 1 (non-asymptotic AMP).**
We made explicit that our central results rely on Conjecture 1, a non-asymptotic AMP/state-evolution statement. This conjecture is standard and widely believed in the community: it is the natural analogue, in our setting, of assumptions that were used in ridge regression and only recently proved rigorously by Cheng & Montanari (2024). We stress that the conjecture is a technical, not conceptual, open problem, and that all our scaling predictions are extensively validated by numerical simulations. We also discussed related rigorous results (Miolane & Montanari, Rush & Venkataramanan, Chen & Lam) and what would be needed to turn the conjecture into a theorem.

**Teacher spectrum and source condition.**
Reviewer d5QU asked about the necessity of the power-law teacher spectrum and its impact. We clarified that this assumption is crucial for the rates in Result 1 and Figure 1 and is directly motivated by the standard source condition from the kernel literature (Caponnetto & De Vito, 2007). We also connected our setting to recent works on heavy-tailed spectra and SGD-induced heavy tails, and explained how our results complement those by giving a static, Bayes-optimal viewpoint.

**Reviewer-specific follow-up.**
- Reviewer Bf9w: we have addressed the concerns regarding the analysis of training dynamics in our summary rebuttal.
- Reviewer d5QU: we have clarified the relationship between teacher and student spectra, specifically justifying the necessity of a heavy-tailed teacher spectrum. Furthermore, we have expanded on how our work relates to prior literature connecting generalization to weight spectra (please refer to our reply to Question 2).
- Reviewer jRAX: we have addressed the concerns regarding our assumptions on idealized data and weight distributions. In particular, we emphasized that the weight-distribution assumption is standard in the scaling-laws literature, known as the source condition.

Reviewer d5QU acknowledged our rebuttal, stating that their concerns were “reasonably addressed.” They expressed the intention to raise their score, given the importance of studying the relationship between weight spectra and generalization.

We hope this brief summary of the post-rebuttal landscape is helpful.

With best regards,

The authors

---

### Meta-Review · Area_Chair_56MH · 2026-01-02

**Summary:**

This paper provides a nice analysis of the spectrum of two simple neural networks and the connection to neural scaling laws. It also provides a theoretical connection to the 5+1 phase model of weight spectra by Martin & Mahoney. Most reviewers think highly of the paper. All reviewers questioned the relatively limited scope of the two simple models and the lack of dynamic analysis. Reviewer d5QU, in addition, asked about the assumption of a heavy-tailed distribution. Reviewer JRAX also mentioned that the results are based on non-asymptotic validity of state evolution, which makes the results somewhat partially rigorous.

**Reviewer Concerns:**

I believe the authors' summary rebuttal correctly points out that the two simple models are mainly for tractability (by mapping them to LASSO and matrix compressed sensing). The extension to analyzing the training dynamics also sounds like it would be difficult to address during the rebuttal period.

**Reviewer Scores:**

I anticipate that the reviewers will mostly accept this rebuttal and will not reduce their scores. The average score is 7 before the rebuttal, and reviewer d5QU mentioned they would raise the score. Therefore, I believe the final score of this paper should be somewhat above the oral threshold.

---

### Decision · Program_Chairs · 2026-01-26

Accept (Oral)